# A mechanism for *hunchback* promoters to readout morphogenetic positional information in less than a minute

Jonathan Desponds[1], Massimo Vergassola[1]*, Aleksandra M Walczak[2]

[1]Physics Department, University of California, San Diego, La Jolla, United States; [2]Laboratoire de Physique, Ecole Normale Supérieure, PSL Research University, CNRS, Sorbonne Université, Paris, France

**Abstract** Cell fate decisions in the fly embryo are rapid: hunchback genes decide in minutes whether nuclei follow the anterior/posterior developmental blueprint by reading out positional information in the Bicoid morphogen. This developmental system is a prototype of regulatory decision processes that combine speed and accuracy. Traditional arguments based on fixed-time sampling of Bicoid concentration indicate that an accurate readout is impossible within the experimental times. This raises the general issue of how speed-accuracy tradeoffs are achieved. Here, we compare fixed-time to on-the-fly decisions, based on comparing the likelihoods of anterior/posterior locations. We found that these more efficient schemes complete reliable cell fate decisions within the short embryological timescales. We discuss the influence of promoter architectures on decision times and error rates, present concrete examples that rapidly readout the morphogen, and predictions for new experiments. Lastly, we suggest a simple mechanism for RNA production and degradation that approximates the log-likelihood function.

*For correspondence:
massimo@physics.ucsd.edu

## Introduction

From development to chemotaxis and immune response, living organisms make precise decisions based on limited information cues and intrinsically noisy molecular processes, such as the readout of ligand concentrations by specialized genes or receptors (*Houchmandzadeh et al., 2002*; *Perry et al., 2012*; *Takeda et al., 2012*; *Marcelletti and Katz, 1992*; *Bowsher and Swain, 2014*). Selective pressure in biological decision-making is often strong, for reasons that range from predator evasion to growth maximization or fast immune clearance. In development, early embryogenesis of insects and amphibians unfolds outside of the mother, which arguably imposes selective pressure for speed to limit the risks of predation and infection by parasitoids (*O'Farrell, 2015*). In *Drosophila* embryos, the first 13 cycles of DNA replication and mitosis occur without cytokinesis, resulting in a multinucleated syncytium containing about 6000 nuclei (*O'Farrell et al., 2004*). Speed is witnessed both by the rapid and synchronous cleavage divisions observed over the cycles, and the successive fast decisions on the choice of differentiation blueprints, which are made in less than 3 min (*Lucas et al., 2018*).

In the early fly embryo, the map of the future body structures is set by the segmentation gene hierarchy (*Nüsslein-Volhard et al., 1984*; *Houchmandzadeh et al., 2002*; *Jaeger, 2011*). The definition of the positional map starts by the emergence of two (anterior and posterior) regions of distinct *hunchback* (*hb*) expression, which are driven by the readout of the maternal Bicoid (Bcd) morphogen gradient (*Houchmandzadeh et al., 2002*, *Figure 1a*). *hunchback* spatial profiles are sharp and the variance in *hunchback* expression of nuclei at similar positions along the AP axis is small (*Desponds et al., 2016*; *Lucas et al., 2018*). Taken together, these observations imply that the short-time readout is accurate and has a low error. Accuracy ensures spatial resolution and the

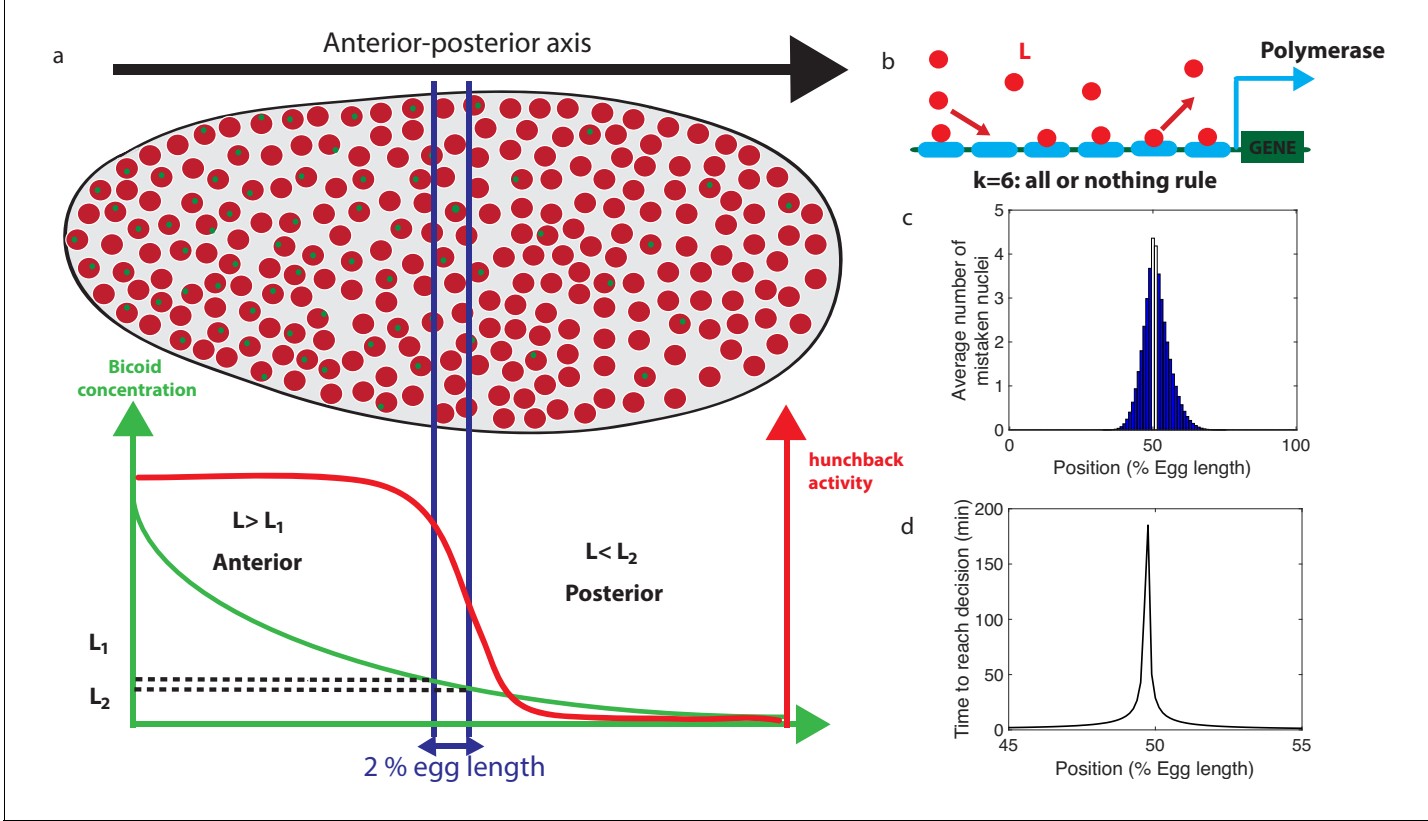

**Figure 1.** Decision between anterior and posterior developmental blueprints. (a) The early *Drosophila* embryo and the Bicoid morphogen gradient. The cartoon shows a projection on one plane of the embryo at nuclear cycles 10–13, when nuclei (red dots) have migrated to the surface of the embryo (*O'Farrell, 2015*). The activity of the *hunchback* gene decreases along the Anterior-Posterior (AP) axis. The green dots represent active transcription loci. The average concentration $L(x)$ of maternal Bicoid decreases exponentially along the AP axis by about a factor five from the anterior (left) to the posterior (right) ends. Between the blue lines lies the boundary region. Its width $\delta_x$ is 2% of the egg length. *hunchback* activity decreases along the AP axis and undergoes a sharp drop around the boundary region. The half-maximum Bcd concentration position in WT embryos is shifted by about 5% of the egg-length toward the anterior with respect to the mid-embryo position (*Struhl et al., 1992*). We consider this *hunchback* half maximum expression as a reference point when describing the AP axis of the embryo. The *hunchback* readout defines the cell fate decision whether each nucleus will follow an anterior or the posterior gene expression program. (b) A typical promoter structure contains six binding sites for Bicoid molecules present at concentration $L$. $k = 6$ indicates that the gene is active only when all binding sites are occupied, defining an all-or-nothing promoter architecture. Other forms and details of the promoter structure will be discussed in *Figure 2* and *Figure 3*. (c) The average number of nuclei making a mistake in the decision process as a function of the egg length position at cycle 11. For a fixed-time decision process completed within $T = 270$ seconds (and $k = 6$, that is, all-or-nothing activation scheme), a large number of nuclei make the wrong decision (full blue bars). $T = 270$ seconds is the duration of the interphase of nuclear cycle 11 (*Tran et al., 2018*). For nuclei located in the boundary region either answer is correct so that we leave these bars unfilled. Most errors happen close to the boundary, as intuitively expected. See Appendix 1 for a detailed description of how the error is computed for the fixed-time decision strategy. (d) The time needed to reach the standard error probability of 32% (*Gregor et al., 2007a*) for the same process as in panel c (see also the subsection 'How many nuclei make a mistake?') as a function of egg length position. Decisions are easy away from the center but the time required for an accurate decision soars close to the boundary up to 50 min – much longer than the embryological times. Parameters for panels (c) and (d) are six binding sites that bind and unbind Bicoid without cooperaivity and a diffusion limited on rate per site $\mu_{max}L = 0.124s^{-1}$, and an unbinding rate per site $\nu_1 = 0.0154s^{-1}$ that lead to half activation in the boundary region.

correct positioning of future organs and body structures, while low errors ensure reproducibility and homogeneity among spatially close nuclei. The amount of positional information available at the transcriptional locus is close to the minimal amount necessary to achieve the required precision (*Gregor et al., 2007b*; *Porcher et al., 2010*; *Garcia et al., 2013*; *Petkova et al., 2019*). Furthermore, part of the morphogenetic information is not accessible for reading by downstream mechanisms (*Tikhonov et al., 2015*), as information is channeled and lost through subsequent cascades of gene activity. In spite of that, by the end of nuclear cycle 14 the positional information encoded in the gap gene readouts is sufficient to correctly predict the position of each nucleus within 2% of the egg length (*Petkova et al., 2019*). Adding to the time constraints, mitosis resets the binding of

transcription factors (TF) during the phase of synchronous divisions (*Lucas et al., 2018*), suggesting that the decision about the nuclei's position is made by using information accessible within one nuclear cycle. Experiments additionally show that during the nuclear cycles 10–13 the positional information encoded by the Bicoid gradient is read out by *hunchback* promoters precisely and within 3 min (*Lucas et al., 2018*).

Effective speed-accuracy tradeoffs are not specific to developmental processes, but are shared by a large number of sensing processes (*Rinberg et al., 2006*; *Heitz and Schall, 2012*; *Chittka et al., 2009*). This generality has triggered interest in quantitative limits and mechanisms for accuracy. Berg and Purcell derived the seminal tradeoff between integration time and readout accuracy for a receptor evaluating the concentration of a ligand (*Berg and Purcell, 1977*) based on its average binding occupancy. Later studies showed that this limit takes more complex forms when rebinding events of detached ligands (*Bialek and Setayeshgar, 2005*; *Kaizu et al., 2014*) or spatial gradients (*Endres and Wingreen, 2008*) are accounted for. The accuracy of the averaging method in *Berg and Purcell, 1977* can be improved by computing the maximum likelihood estimate of the

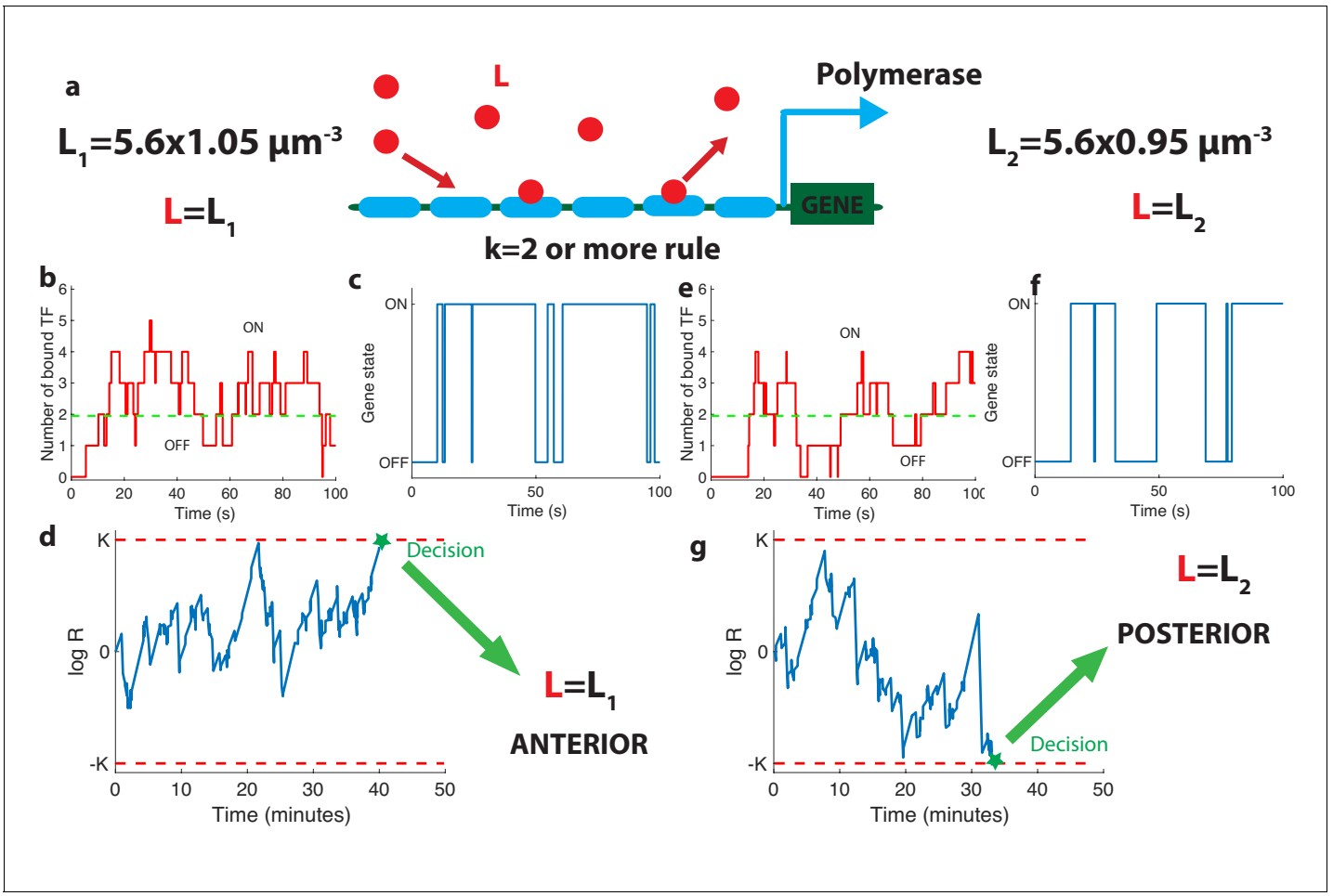

**Figure 2.** The relation between promoter structure and on-the-fly decision-making. (a). Using six Bicoid binding sites, the promoter decides between two hypothetical Bcd concentrations $L = L_1$ and $L = L_2$, given the actual (unknown) concentration $L$ in the nucleus. The number of occupied Bicoid sites fluctuates with time (b) and we assume the gene is expressed (c) when the number of occupied Bicoid binding sites on the promoter is $\geq k$ (green dashed line, here $k = 2$). The gene activity defines a non-Markovian telegraph process. The ratio of the likelihoods that the time trace of this telegraph process is generated by $L = L_1$ vs $L = L_2$ is the log-likelihood ratio used for decision-making (d). The log-likelihood ratio undergoes random excursions until it reaches one of the two decision boundaries ($K, -K$). In d. the actual concentration is $L = L_1$ and the log-likelihood ratio hits the upper barrier and makes the right decision. When $L = L_2$, less Bicoid-binding sites are occupied (e) and the gene is less likely to be expressed (f), resulting in a negative drift in the log-likelihood ratio, which directs the random walk to the lower boundary $-K$ and the $L = L_2$ decision (g). We consider that all six binding sites bind Bicoid independently and are identical with binding rate per site $\mu_{\max}L = 0.07s^{-1}$ and unbinding rate per site $\nu_1 = 0.08s^{-1}$, $e = 0.2$, $k = 2$, $L = L_1 = 5.88\mu m^{-3}$ for panels (b, c, d) and $L = L_2 = 5.32\mu m^{-3}$ (e, f, g).

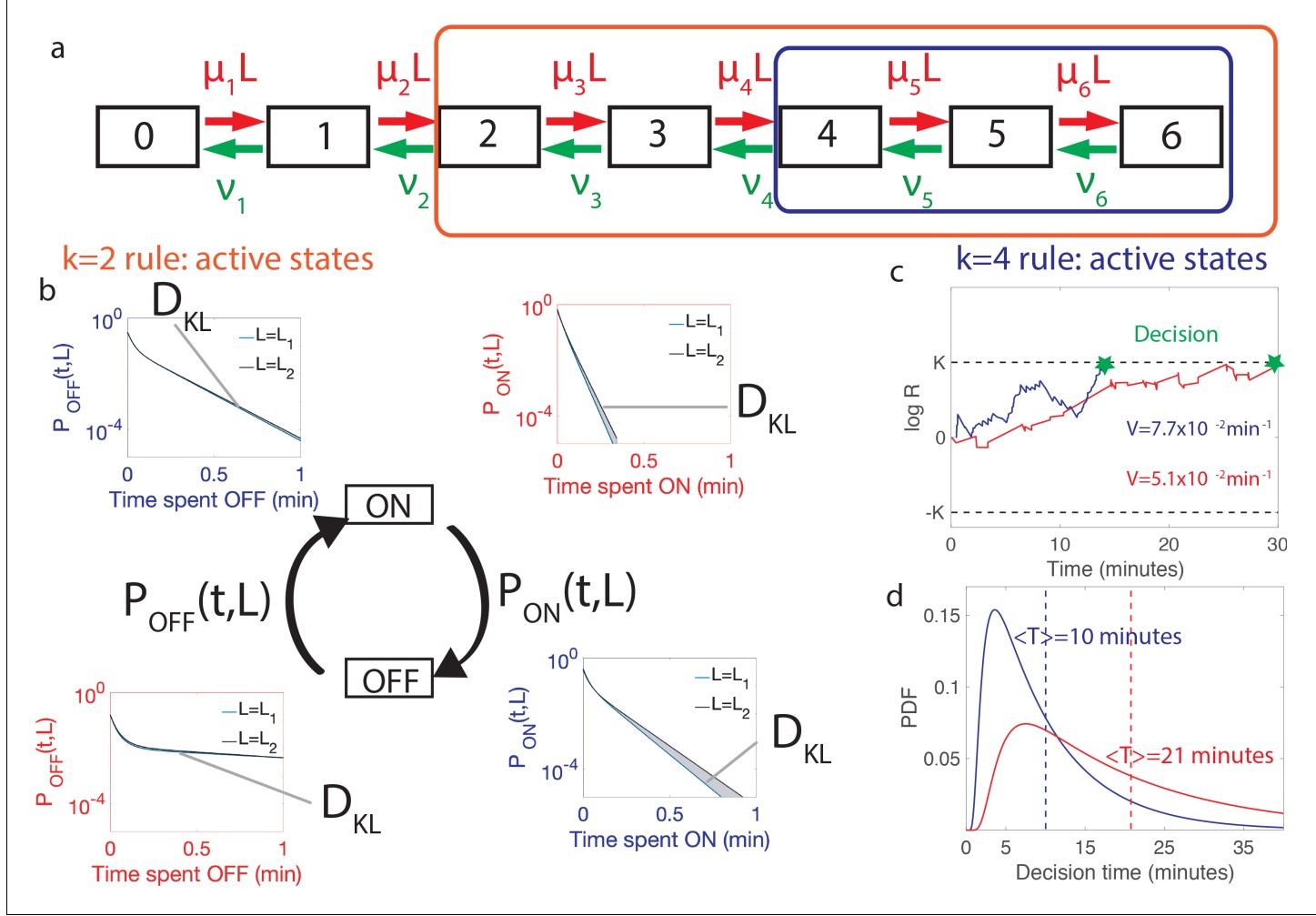

**Figure 3.** Comparing the performance of two promoter activation rules. (a) The dynamics of the six Bcd binding site promoter is represented by a seven state Markov chain where the state number indicates the number of occupied Bicoid-binding sites. The boxes indicate the states in which the gene is expressed for the 2-or-more activation rule (red box and red in panels **b-d**) where the gene is active when 2-or-more TF are bound and the 4-or-more activation rule (blue box and blue in panels **b-d**) where the gene is active when 4-or-more TF are bound. (b) The dynamics of TF binding translates into bursting and inactive periods of gene activity. The OFF and ON time distributions are different for the two hypothetical concentrations (blue boxes for $k = 4$ and red boxes for $k = 2$). The Kullback-Leibler divergence between the distributions for the two hypothetical concentrations ($D_{KL}$) sets the decision time and is related to the difference in the area below the two distributions. For the $k = 4$ activation rule, the OFF time distributions are similar for the two hypothetical concentrations but the ON times distributions are very different. The ON times are more informative for the $k = 4$ activation rule than the $k = 2$ activation rule (c) The drift $V$ of the log-likelihood ratio characterizes the deterministic bias in the decision process. The differences in (b) translate into larger drift for $k = 4$ for the same binding/unbinding dynamics. (d) The distribution of decision times (calculated as the first-passage time of the log-likelihood random walk) decays exponentially for long times. Higher drift leads to on average faster decisions than for the $k = 4$ activation rule (mean decision times are shown in dashed lines). For all panels the six binding sites are independent and identical with $L = L_1 = 5.88 \ \mu m^{-3}$, $L_2 = 5.32 \ \mu m^{-3}$, $e = 0.1$, $\mu_{\max} L = 0.14 \ s^{-1}$ and $\nu = 0.08 \ s^{-1}$ for all binding sites.

time series of receptor occupancy for a given model (*Endres and Wingreen, 2009*; *Mora and Wingreen, 2010*). However, none of these approaches can result in a precise anterior-posterior (AP) decision for the *hunchback* promoter in the short time of the early nuclear cycles, which has led to the conclusion that there is not enough time to apply the fixed-time Berg-Purcell strategy with the desired accuracy (*Gregor et al., 2007a*). Additional mechanisms to increase precision (including internuclear diffusion) do yield a speed-up (*Erdmann et al., 2009*; *Aquino et al., 2016*), yet they are not sufficient to meet the 3-min challenge. The issue of the embryological speed-accuracy tradeoff remains open.

The approaches described above are fixed-time strategies of decision-making, that is, they assume that decisions are made at a pre-defined deterministic time $T$ that is set long enough to achieve the desired level of error and accuracy. As a matter of fact, fixing the decision time is not optimal because the amount of available information depends on the specific statistical realization of the noisy signal that is being sensed. The time of decision should vary accordingly and therefore depend on the realization. This intuition was formalized by A. Wald by his Sequential Probability Ratio Test (SPRT) (*Wald, 1945a*). SPRT achieves optimality in the sense that it ensures the fastest decision strategy for a given level of expected error. The adaption of the method to biological sensing posits that the cell discriminates between two hypothetical concentrations by accumulating information through binding events, and by computing on the fly the ratio of the likelihoods (or appropriate proxies) for the two concentrations to be discriminated (*Siggia and Vergassola, 2013*). When the ratio 'strongly' favors one of the two hypotheses, a decision is triggered. The strength of the evidence required for decision-making depends on the desired level of error. For a given level of precision, the average decision time can be substantially shorter than for traditional averaging algorithms (*Siggia and Vergassola, 2013*). SPRT has also been proposed as an efficient model of decision-making in the fields of social interactions and neuroscience (*Gold and Shadlen, 2007*; *Marshall et al., 2009*) and its connections with non-equilibrium thermodynamics are discussed in *Roldán et al., 2015*.

Wald's approach is particularly appealing for biological concentration readouts since many of them, including the anterior-posterior decision faced by the *hunchback* promoter, appear to be binary decisions. Our first goal here is to specifically consider the paradigmatic example of the *hunchback* promoter and elucidate the degree of speed-up that can be achieved by decisions on the fly. Second, we investigate specific implementations of the decision strategy in the form of possible *hunchback* promoter architectures. We specifically ask how cooperative TF binding affects the sensing limits. Our results have implications beyond fly development and are generally relevant to regulatory processes. We identify promoter architectures that, by approximating Wald's strategy, do satisfy several key experimental constraints and reach the experimentally observed level of accuracy of *hunchback* expression within the (apparently) very stringent time limits of the early nuclear cycles.

## Results

### Methodological setup

#### The decision process of the anterior vs posterior *hunchback* expression

The problem faced by nuclei in their decision of anterior *vs* posterior developmental fate is sketched in *Figure 1a*. By decision we mean that nuclei commit to a cell fate through a process that is mainly irreversible leading to one of two classes of cell states that correspond to either the anterior or the posterior regions of the embryo, based on positional information acquired through gene activity. We limit our investigation of promoter architectures to the six experimentally identified Bicoid-binding sites (*Figure 1b*). We do not consider the known Hunchback-binding sites because before nuclear cycle 13 there is little time to produce sufficient concentrations of zygotic proteins for a significant feedback effect and the measured maternal *hunchback* profile has not been shown to alter anterior-posterior decision-making. Following the observation that Bcd readout is the leading factor in nuclei fate determination (*Ochoa-Espinosa et al., 2009*), we also neglect the role of other maternal gradients, for example Caudal, Zelda or Capicua (*Jiménez et al., 2000*; *Sokolowski et al., 2012*; *Tran et al., 2018*; *Lucas et al., 2018*), since the readout of these morphogens can only contribute additional information and decrease the decision time. We focus on the proximal promoter since no active enhancers have been identified for the *hunchback* locus in nuclear cycles 11–13 (*Perry et al., 2011*). Our results can be generalized to enhancers (*Hannon et al., 2017*), the addition of which only further improves the speed-accuracy efficacy, as we explicitly show for a simple model of Bicoid activated enhancers in the section 'Joint dynamics of Bicoid enhancer and promoter'. Since our goal is to show that accurate decisions can be made rapidly, we focus on the *worst case decision-making scenario*: the positional information (*Wolpert et al., 2015*) is gathered through a readout of the Bicoid concentration only, and the decision is assumed to be made independently in each nucleus. Having additional information available and/or coupling among nuclei can only strengthen our conclusion.

The profile of the average concentration of the maternal morphogen Bicoid $L(x)$ is well represented by an exponential function that decreases from the anterior toward the posterior of the embryo : $L(x) = L_0 e^{-5(x-x_0)/100}$, where $x$ is the position along the anterior-posterior axis measured in terms of percentage egg-length (EL), and $x_0$ is the position of half maximum $hb$ expression corresponding to $L_0$ Bcd concentration. The decay length $\lambda = 5$ corresponds to 20% EL (*Gregor et al., 2007a*). Nuclei convert the graded Bicoid gradient into a sharp border of *hunchback* expression (*Figure 1a*), with high and low expressions of the *hunchback* promoter at the left and the right of the border respectively (*Driever and Nüsslein-Volhard, 1988*; *Struhl et al., 1992*; *Crauk and Dostatni, 2005*; *Gregor et al., 2007b*; *Houchmandzadeh et al., 2002*; *Porcher et al., 2010*; *Garcia et al., 2013*; *Lucas et al., 2013*; *Tran et al., 2018*; *Lucas et al., 2018*). We define the *border region* of width $\delta_x$ symmetrically around $x_0$ by the dashed lines in *Figure 1a*. $\delta_x$ is related to the positional resolution (*Erdmann et al., 2009*; *Tran et al., 2018*) of the anterior-posterior decision: it is the minimal distance measured in percentages of egg-length between two nuclei's positions, at which the nuclei can distinguish the Bcd concentrations. Although this value is not known exactly, a lower bound is estimated as $\delta_x \sim 2\%$ EL (*Gregor et al., 2007a*), which corresponds to the width of one nucleus.

We denote the Bcd concentration at the anterior (respectively posterior) boundary of the border region by $L_1$ (respectively $L_2$) (see *Figure 1a*). At each position $x$, nuclei compare the probability that the local concentration $L(x)$ is greater than $L_1$ (anterior) or smaller than $L_2$ (posterior). By using current best estimates of the parameters (see Appendix 1), a classic fixed-time-decision integration process and an integration time of 270 s (the duration of the interphase in nuclear cycle 11 [*Tran et al., 2018*]), we compute in *Figure 1c* the probability of error per nucleus for each position in the embryo (see Appendix 2 for details). As expected, errors occur overwhelmingly in the vicinity of the border region, where the decision is the hardest (*Figure 1c*). For nuclei located within the border region, both anterior and posterior decisions are correct since the nuclei lie close to both regions. It follows that, although the error rate can formally be computed in this region, the errors do not describe positional resolution mistakes and do not contribute to the total error (white zone in *Figure 1c*).

In view of *Figure 1c* and to simplify further analysis we shall focus on the boundaries of the border region : each nucleus discriminates between hypothesis 1 – the Bcd concentration is $L = L_1$, and hypothesis 2 – the Bcd concentration is $L = L_2$. To achieve a positional resolution of $\delta_x = 2\%$ EL, nuclei need to be able to discriminate between differences in Bcd concentrations on the order of 10%. In addition to the variation in Bcd concentration estimates that are due to biological precision, concentration estimated using many trials follows a statistical distribution. The central limit theorem suggests that this distribution is approximately Gaussian. This assumption means that the probability that the Bcd concentration estimate deviates from the actual concentration by more than the prescribed 10% positional resolution is 32% (see the subsection 'How many nuclei make a mistake?' for variations on the value and arguments on the error rate). In *Figure 1d*, we show that the time required under a fixed-time-decision strategy for a promoter with six binding sites to estimate the Bcd concentration within the 32% Gaussian error rate (*Gregor et al., 2007a*) close to the boundary is much longer than 270 s, $\simeq 40$ minutes (see Appendix 2 for details of the calculation). The activation rule for the promoter architecture in the figure is that all binding sites need to be bound for transcription initiation.

## Identifying fast decision promoter architectures
### The promoter model

We model the $hb$ promoter as six Bcd binding sites (*Schröder et al., 1988*; *Driever et al., 1989*; *Struhl et al., 1989*; *Ochoa-Espinosa et al., 2005*) that determine the activity of the gene (*Figure 2a*). Bcd molecules bind to and unbind from each of the $i = 1, ..., 6$ sites with rates $\mu_i$ and $\nu_i$, which are allowed to be different for each site. For simplicity, the gene can only take two states : either it is silenced (OFF) and mRNA is not produced, or the gene is activated (ON) and mRNA is produced at full speed. While models that involve different levels of polymerase loading are biologically relevant and interesting, the simplified model allows us to gain more intuition and follows the worst-case scenario logic that we discussed in the previous subsection 'The decision process of the anterior *vs* posterior *hunchback* expression'. The same remark applies for the wide variety of promoter architectures considered in previous works (*Estrada et al., 2016*; *Tran et al., 2018*). In

particular, we assume that only the number of sites that are bound matters for gene activation (and not the specific identity of the sites). Such architectures are again a subset of the range of architectures considered in *Estrada et al., 2016*; *Tran et al., 2018*.

The dynamics of our model is a Markov chain with seven states with probability $P_i$ corresponding to the number of sites occupied: from all sites unoccupied (probability $P_0$) to all six sites bound by Bcd molecules (probability $P_6$). The minimum number $k$ of bound Bicoid sites required to activate the gene divides this chain into the two disjoint and complementary subsets of active states ($P_{\mathrm{ON}} = \sum_{i=k}^{6} P_i$, for which the gene is activated) and inactive states ($P_{\mathrm{OFF}} = \sum_{i=0}^{k-1} P_i$, for which the gene is silenced) as illustrated in *Figure 2b and d*.

As Bicoid ligands bind and unbind the promoter (*Figure 2b*), the gene is successively activated and silenced (*Figure 2c*). This binding/unbinding dynamics results in a series of OFF and ON activation times that constitute all the information about the Bcd concentration available to downstream processes to make a decision. We note that the idea of translating the statistics of binding-unbinding times into a decision remains the same as in the Berg-Purcell approach, where *only* the activation times are translated into a decision (and not the deactivation times). The promoter architecture determines the relationship between Bcd concentration and the statistics of the ON-OFF activation time series, which makes it a key feature of the positional information decision process. Following (*Siggia and Vergassola, 2013*), we model the decision process as a Sequential Probability Ratio Test (SPRT) based on the time series of gene activation. At each point in time, SPRT computes the likelihood $P$ of the observed time series under both hypotheses $P(L_1)$ and $P(L_2)$ and takes their ratio : $R(t) = P(L_1)/P(L_2)$. The logarithm of $R(t)$ undergoes stochastic changes until it reaches one of the two decision threshold boundaries $K$ or $-K$ (symmetric boundaries are used here for simplicity) (*Figure 2d*). The decision threshold boundaries $K$ are set by the error rate $e$ for making the wrong decision between the hypothetical concentrations : $K = \log((1-e)/e)$ (see *Siggia and Vergassola, 2013* and Appendix 3). The choice of $K$ or $e$ depends on the level of reproducibility desired for the decision process. We set $K \simeq 0.75$, corresponding to the widely used error rate $e \simeq 0.32$ of being more than one standard deviation away from the mean of the estimate for the concentration assumed to be unbiased in a Gaussian model (see the subsection 'The decision process of the anterior vs posterior *hunchback* expression'). The statistics of the fluctuations in likelihood space are controlled by the values of the Bcd concentrations: when Bcd concentration is low, small numbers of Bicoid ligands bind to the promoter (*Figure 2e*) and the *hb* gene spends little time in the active expression state (*Figure 2f*), which leads to a negative drift in the process and favors the lower one of the two possible concentrations (*Figure 2g*).

## Mean decision time: connecting drift-diffusion and Wald's approaches

In this section, we develop new methods to determine the statistics of gene switches between the OFF and ON expression states. Namely, by relating Wald's approach (*Wald, 1945a*) with drift-diffusion, we establish the equality between the drift and diffusion coefficients in decision making space for difficult decision problems, that is, when the discrimination is hard, we elucidate the reason underlying the equality. That allows us to effectively determine long-term properties of the likelihood log-ratio and compute mean decision times for complex architectures.

A gene architecture consists of $N$ binding sites and is represented by $N + 1$ Markov states corresponding to the number of bound TF, and the rates at which they bind or unbind (*Figure 3a*). For a given architecture, the dynamics of binding/unbinding events and the rules for activation define the two probability distributions $P_{\mathrm{OFF}}(t, L)$ and $P_{\mathrm{ON}}(s, L)$ for the duration of the OFF and ON times, respectively (*Figure 3b*). The two series are denoted $\{t_i\}_{1 \le i \le J^+}$ and $\{s_j\}_{1 \le j \le J^-}$, where $J^+$ and $J^-$ are the number of switching events in time $t$ from OFF to ON and vice versa. For those cases where the two concentrations $L_1$ and $L_2$ are close and the discrimination problem is difficult (which is the case of the *Drosophila* embryo), an accurate decision requires sampling over many activation and deactivation events to achieve discrimination. The logarithm of the ratio $R(t)$ can then be approximated by a drift–diffusion equation: $d \log R(t)/dt = V + \sqrt{2D}\eta$, where $V$ is the constant drift, that is, the bias in favor of one of the two hypotheses, $D$ is the diffusion constant and $\eta$ a standard Gaussian white noise with zero mean and delta-correlated in time (*Wald, 1945a*; *Siggia and Vergassola, 2013*). The decision time for the case of symmetric boundaries $K = -K_- = K_+$, is given by the mean first-

passage time for this biased random walk in the log-likelihood space (*Redner, 2001*; *Siggia and Vergassola, 2013*):

$$\langle T \rangle = \frac{K \tanh(VK/2D)}{V} . \tag{1}$$

Note that in this approximation all the details of the promoter architecture are subsumed into the specific forms of the drift $V$ and the diffusion $D$.

**Drift**. We assume for simplicity that the time series of OFF and ON times are independent variables (when this assumption is relaxed, see Appendix 8). This assumption is in particular always true when gene activation only depends on the number of bound Bicoid molecules. Under these assumptions, we can apply Wald's equality (*Wald, 1945b*; *Durrett, 2010*) to the log-likelihood ratio, $\log R(t)$. Wald considered the sum of a random number $M$ of independent and identically distributed (i.i.d.) variables. The equality that he derived states that if $M$ is independent of the outcome of variables with higher indices $(X_i)_{i>M}$ (i.e. $M$ is a stopping time), then the average of the sum is the product $\langle M \rangle \langle X_i \rangle$.

Wald's equality applies to our likelihood sum ($\sum_i^M \log R_i$ of the likelihoods, where $M$ is the number of ON and OFF times before a given (large) time $t$). We conclude the drift of the log-likelihood ratio, $\log R(t)$, is inversely proportional to $(\tau_{\mathrm{ON}} + \tau_{\mathrm{OFF}})$, where $\tau_{\mathrm{ON}}$ is the mean of the distribution of ON times $P_{\mathrm{ON}}(t, L)$ and $\tau_{\mathrm{OFF}}$ is the mean of the distribution of OFF times $P_{\mathrm{OFF}}(s, L)$. The term $(\tau_{\mathrm{ON}} + \tau_{\mathrm{OFF}})$ determines the average speed at which the system completes an activation/deactivation cycle, while the average $\langle \log R_i \rangle$ describes how much deterministic bias the system acquires on average per activation/deactivation cycle. The latter can be re-expressed in terms of the Kullback-Leibler divergence $D_{KL}(f||g) = \int_0^\infty dt' f(t') \log(f(t')/g(t'))$ between the distributions of the OFF and ON times calculated for the actual concentration $L$ and each one of the two hypotheses, $L_1$ and $L_2$ :

$$\begin{aligned} V \ &= \tfrac{1}{(\tau_{\mathrm{ON}} + \tau_{\mathrm{OFF}})} \big[ D_{KL}(P_{\mathrm{OFF}}(.,L)||P_{\mathrm{OFF}}(.,L_2)) - D_{KL}(P_{\mathrm{OFF}}(.,L)||P_{\mathrm{OFF}}(.,L_1)) \\ &+ D_{KL}(P_{\mathrm{ON}}(.,L)||P_{\mathrm{ON}}(.,L_2)) - D_{KL}(P_{\mathrm{ON}}(.,L)||P_{\mathrm{ON}}(.,L_1)) \big] . \end{aligned} \tag{2}$$

*Equation 2* quantifies the intuition that the drift favors the hypothetical concentration with the time distribution which is the closest to that of the real concentration $L$ (*Figure 3b*).

**Diffusivity : Why it is more involved to calculate and how we circumvent it**. While the drift has the closed simple form in *Equation 2*, the diffusion term is not immediately expressed as an integral. The qualitative reason is as follows. Computing the likelihood of the two hypotheses requires computing a sum where the addends are stochastic (ratios of likelihoods) and the number of terms is also stochastic (the number of switching events). These two random variables are correlated: if the number of switching events is large, then the times are short and the likelihood is probably higher for large concentrations. While the drift is linear in the above sum (so that the average of the sum can be treated as shown above), the diffusivity depends on the square of the sum. The diffusivity involves then the correlation of times and ratios (*Carballo-Pacheco et al., 2019*), which is harder to obtain as it depends a priori on the details of the binding site model (see the subsection 'Equality between drift and diffusivity' and the subsection 'When are correlations between the times of events leading to decision important?' of Appendix 3 for details).

We circumvent the calculation of the diffusivity by noting that the same methods used to derive *Equation (1)* also yield the probability of first absorption at one of the two boundaries, say $+K$ (see the subsection 'Equality between drift and diffusivity' of Appendix 3):

$$\Pi_K = \frac{e^{VK/D}}{1 + e^{\frac{KV}{D}}} . \tag{3}$$

By imposing $\Pi_K = 1 - e$, we obtain $VK/D = \log((1-e)/e)$ and the comparison with the expression of $K = \log((1-e)/e)$ leads to the equality $D = V$.

The above equality is expected to hold for difficult decisions only. Indeed, drift-diffusion is based on the continuity of the log-likelihood process and Wald's arguments assume the absence of substantial jumps in the log-likelihood over a cycle. In other words, the two approaches overlap if the hypotheses to be discriminated are close. For very distinct hypotheses (easy discrimination problems), the two approaches may differ from the actual discrete process of decision and among

themselves. We expect then that $V = D$ holds only for hypotheses that are close enough, which is verified by explicit examples (see the subsection 'Equality between drift and diffusivity' of Appendix 3). The Appendix subsection also verifies $V = D$ by expanding the general expression of $V$ and $D$ for close hypotheses. The origin of the equality is discussed below.

Using $V = D$, we can reduce the general formula *Equation (1)* to

$$\langle T \rangle = \frac{K \tanh(K/2)}{V} = \frac{K}{V}(1 - 2e),$$ (4)

which is formula 4.8 in *Wald, 1945a* and it is the expression that we shall be using (unless stated otherwise) in the remainder of the paper.

The additional consequence of the equality $V = D$ is that the argument $VK/2D$ of the hyperbolic tangent in *Equation (1)* is $\simeq K/2 = \ln\left(\sqrt{(1-e)/e}\right)$. It follows that for any problem where the error $e \ll 1$, the argument of the hyperbolic tangent is large and the decision time is weakly dependent on deviations to $V = D$ that occur when the two hypotheses differ substantially. A concrete illustration is provided in the subsection 'The first passage time to decision' of Appendix 3.

**Single binding site example**. As an example of the above equations, we consider the simplest possible architecture with a single binding site ($N = 1$), where the gene activation and de-activation processes are Markovian. In this case, the de-activation rate $\nu$ is independent of TF concentration and the activation rate is exponentially distributed $P_{\text{OFF}}(L, t) = kLe^{kLt}$. We can explicitly calculate the drift $V = \nu kL/(\nu + kL)(\log(L_1/L_2) + (L_2 - L_1)/L)$ (*Equation 2*) and expand it for $L_2 = L$ and $L_1 = L + \delta L$, at leading order in $\delta L$. Inserting the resulting expression into *Equation 4*, we conclude that

$$\langle T \rangle = \frac{\nu + kL}{\nu kL} \frac{2KL^2}{\delta L^2} \tanh\left(\frac{K}{2}\right),$$ (5)

decreases with increasing relative TF concentration difference $\delta L/L$ and gives a very good approximation of the complete formula (see *Appendix 3—figure 2a–c* with different values of $\delta L/L$).

*Equations 2 and 4* greatly reduce the complexity of evaluating the performance of architectures, especially when the number of binding sites is large. Alternatively, computing the correlation of times and log-likelihoods would be increasingly demanding as the size of the gene architecture transfer matrices increase. As an illustration, *Figure 3* compares the performance of different activation strategies : the 2-or-more rule ($k = 2$), which requires at least two Bcd-binding sites to be occupied for *hb* promoter activation (*Figure 3a–d* in blue), and the 4-or-more rule ($k = 4$) (*Figure 3a–d* in red) for fixed binding and unbinding parameters. *Figure 3c* shows that stronger drifts lead to faster decisions. The full decision time probability distribution is computed from the explicit formula for its Laplace transform (*Siggia and Vergassola, 2013*, *Figure 3d*). With the rates chosen for *Figure 3*, the $k = 4$ rule leads to an ON time distribution that varies strongly with the concentration, making it easier to discriminate between similar concentrations: it results in a stronger average drift that leads to a faster decision than $k = 2$ (*Figure 3d*).

**What is the origin of the $V = D$ equality?** The special feature of the SPRT random process is that it pertains to a log-likelihood. This is at the core of the $V = D$ equality that we found above. First, note that the equality is dimensionally correct because log-likelihoods have no physical dimensions so that both $V$ and $D$ have units of $time^{-1}$. Second, and more important, log-likelihoods are built by Bayesian updating, which constrains their possible variations. In particular, given the current likelihoods $P_1(t) = \Pi_j P_{\text{ON}}(t_j, L_1) P_{\text{OFF}}(s_j, L_1)$ and $P_2(t) = \Pi_j P_{\text{ON}}(t_j, L_2) P_{\text{OFF}}(s_j, L_2)$ at time $t$ for the two concentrations $L_1$ and $L_2$ and the respective probabilities $Q_1(t) = P_1/(P_1 + P_2)$ and $Q_2(t) = 1 - Q_1$ of the two hypotheses, it must be true that the expected values after a certain time $\Delta t$ remain the same if the expectation is taken with respect to the current $P_i(t)$ (see, e.g. *Reddy et al., 2016*). In formulae, this implies that the average variation of the probability $Q_2$ over a given time $\Delta t$ that is

$$\langle \Delta Q_2 \rangle = Q_1 \langle \Delta Q_2 \rangle_1 + Q_2 \langle \Delta Q_2 \rangle_2,$$ (6)

should vanish (see the subsection 'Equality between drift and diffusivity' of Appendix 3 for a derivation). Here, $\langle \Delta Q_2 \rangle_1$ is the expected variation of $Q_2$ under the assumption that hypothesis 1 is true and $\langle \Delta Q_2 \rangle_2$ is the same quantity but under the assumption that hypothesis 2 is true. We notice now

that $Q_2(t) = \frac{1}{1+e^{\mathcal{L}(t)}}$, where $\mathcal{L} = \log R$ is the log-likelihood, and that the drift-diffusion of the log-likelihood implies that $\langle \Delta \mathcal{L} \rangle_1 = V\Delta t$, $\langle \Delta \mathcal{L} \rangle_2 = -V\Delta t$ and $\langle (\Delta \mathcal{L} - \langle \Delta \mathcal{L} \rangle_1)^2 \rangle_1 = \langle (\Delta \mathcal{L} - \langle \Delta \mathcal{L} \rangle_2)^2 \rangle_2 = 2D\Delta t$. By using that $dQ_2/d\mathcal{L} = -Q_1 Q_2$ and $d^2 Q_2/d\mathcal{L}^2 = -Q_1 Q_2 (Q_2 - Q_1)$, we finally obtain that

$$\langle \Delta Q_2 \rangle = Q_1 Q_2 (Q_2 - Q_1) \Delta t [V - D], \tag{7}$$

and imposing $\langle \Delta Q_2 \rangle = 0$ yields the equality $V = D$. Note that the above derivation holds only for close hypotheses, otherwise the velocity and the diffusivity under the two hypotheses do not coincide.

## Additional embryological constraints on promoter architectures

In addition to the requirements imposed by their performance in the decision process (green dashed line in *Figure 4a*), promoter architectures are constrained by experimental observations and properties that limit the space of viable promoter candidates for the fly embryo. A discussion about their possible function and their relation to downstream decoding processes is deferred to the final section.

First, we require that the average probability for a nucleus to be active in the boundary region is equal to 0.5, as it is experimentally observed (*Lucas et al., 2013*; *Figure 1a*). This requirement mainly impacts and constrains the ratio between binding rates $\mu_i$ and unbinding rates $\nu_i$.

Second, there is no experimental evidence for active search mechanisms of Bicoid molecules for its targets. It follows that, even in the best case scenario of a Bcd ligand in the vicinity of the promoter always binding to the target, the binding rate is equal to the diffusion limited arrival rate $\mu_{\max}L \simeq 0.124 s^{-1}$ (Appendix 1). As a result, the binding rates $\mu_i$ are limited by diffusion arrivals and the number of available binding sites: $\mu_i \leq \mu_{\max}(7 - i)$ (black dashed line in *Figure 4b*), where $L$ is the concentration of Bicoid. This sets the timescale for binding events. In Appendix 1, we explore the different measured values and estimates of parameters defining the diffusion limit $\mu_{\max}L$ and their influence on the decision time (see *Appendix 1—table 1* for all the predictions).

Third, as shown in *Figure 1*, the *hunchback* response is sharp, as quantified by fitting a Hill function to the expression level *vs* position along the egg length. Specifically, the *hunchback* expression (in arbitrary units) $f_{\text{hb}}$ is well approximated as a function of the Bicoid concentration $L(x)$ by the Hill function $f_{\text{hb}}(x) \simeq L(x)^H / (L(x)^H + L_0^H)$, where $L_0$ is the Bcd concentration at the half-maximum *hb* expression point and $H$ is the Hill coefficient (*Figure 1a*). Experimentally, the measured Hill coefficient for mRNA expression from the WT *hb* promoter is $H \sim 7 - 8$ (*Lucas et al., 2018*; *Tran et al., 2018*). Recent work (*Tran et al., 2018*) suggests that these high values might not be achieved by Bicoid-binding sites only. Given current parameter estimates and an equilibrium binding model, (*Tran et al., 2018*) shows that a Hill coefficient of 7 is not achievable within the duration of an early nuclear cycle ($\simeq 5$ min). That points at the contribution of other mechanisms to pattern steepness. Given these reasons (and the fact that we limit ourselves only to a model with six equilibrium Bcd-binding sites only), we shall explore the space of possible equilibrium promoter architectures limiting the steepness of our profiles to Hill coefficients $H \sim 4 - 5$.

## Numerical procedure for identifying fast decision-making architectures

Using *Equations 2 and 4*, we explore possible *hb* promoter architectures and activation rules to find the ones that minimize the time required for an accurate decision, given the constraints listed in the paragraph 'Additional embryological constraints on promoter architectures'. We optimize over all possible binding rates $(\mu_i)_{1 \leq i \leq 6}$ ($\mu_1$ is the binding rate of the first Bcd ligand and $\mu_6$ the binding rate of the last Bcd ligand when 5 Bcd ligands are already bound to the promoter), and the unbinding rates $(\nu_i)_{1 \leq i \leq 6}$ ($\nu_1$ is the unbinding rate of a single Bcd ligand bound to the promoter and $\nu_6$ is the unbinding rate of all Bcd ligands when all six Bcd-binding sites are occupied). We also explore different activation rules by varying the minimal number of Bcd ligands $k$ required for activation in the *k*-or-more activation rule (*Estrada et al., 2016*; *Tran et al., 2018*). We use the most recent estimates of biological constants for the *hb* promoter and Bcd diffusion (see Appendix 1) and set the error rate at the border to 32% (*Gregor et al., 2007b*; *Petkova et al., 2019*). Reasons for this choice were given in the subsection 'The decision process of the anterior *vs* posterior *hunchback* expression' and will be revisited in the subsection 'How many nuclei make a mistake?', where we shall introduce some embryological considerations on the number of nuclei involved in the decision process and determine the error probability accordingly. The optimization procedure that minimized the

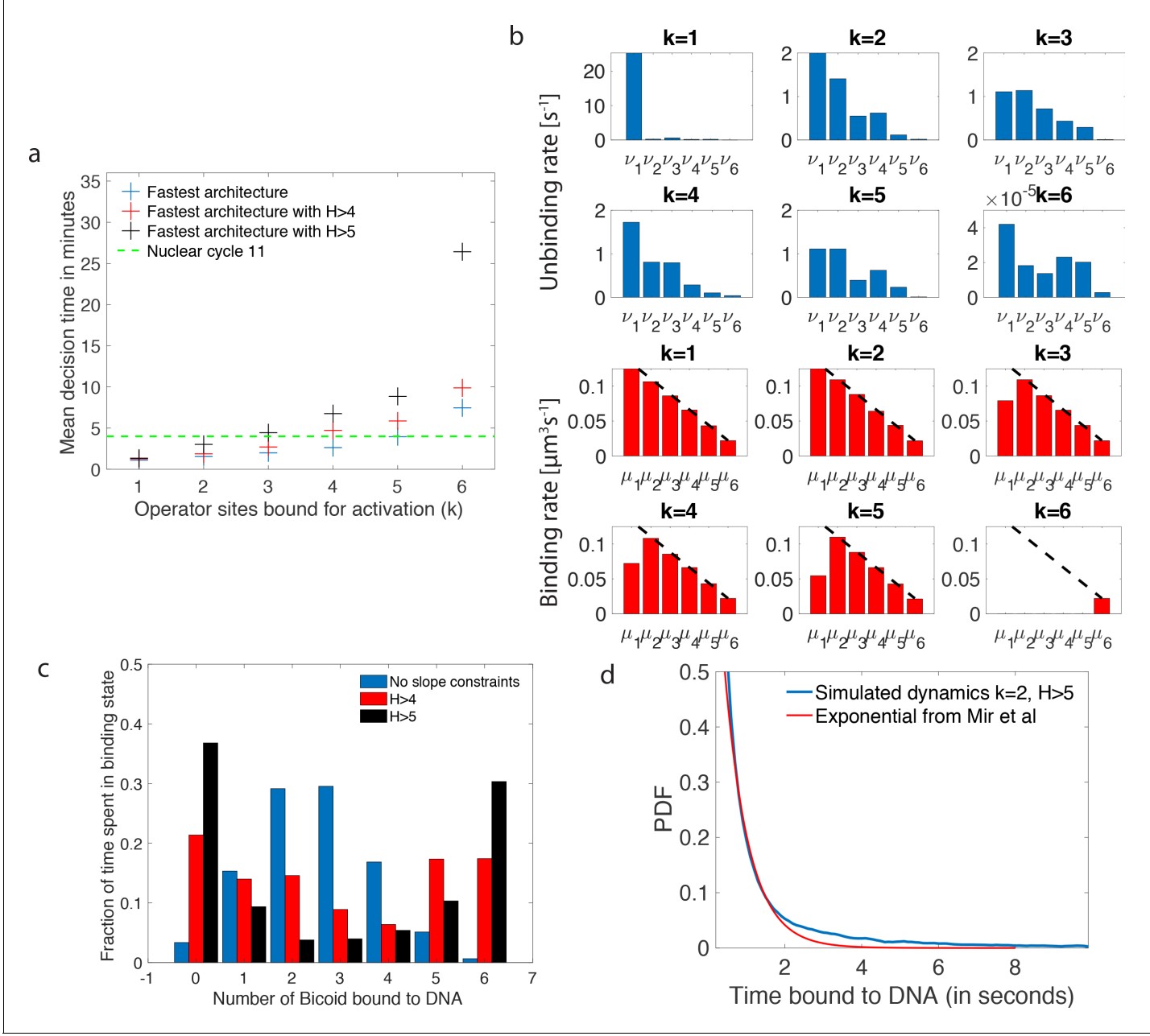

**Figure 4.** Performance, constraints and statistics of fastest decision-making architectures. (a) Mean decision time for discriminating between two concentrations with $|L_2 - L_1| = 0.1L$ and $e = 0.32$. Results shown for the fastest decision-making architectures for different activation rules and steepness constraints. For a given activation rule (k), we optimize over all values of ON rates $\mu_i$ and OFF rates $\nu_i$ (see **Figure 3a**) within the diffusion limit (0.124 $s^{-1}$ per site), constraining the steepness $H$ and probability of nuclei to be active at the boundary (see the paragraph 'Additional embryological constraints on promoter architectures'). The green lines denotes the interphase duration of nuclear cycle 11 and even for the strongest constraints ($H>5$) we identify architectures that make an accurate decision within this time limit. (b) The unbinding rates (blue) and binding rates (red) of the fastest decision-making architectures with $H>5$ – all these regulatory systems require cooperativity in TF binding to the promoter-binding sites. The dashed line on the ON rates plots shows the upper bound set by the diffusion limit. (c) Histogram of the probability distribution of the time spent in different Bcd-binding site occupancy states for the fastest decision-making architecture for $k = 3$ and no constraints on the slope (blue), $H>4$ (red) and $H>5$ (black). (d) Probability distribution of the time spent bound to th DNA by Bicoid molecules for the fastest decision-making architecture with $H>5$ and $k = 2$. Our prediction is compared to the exponential distribution with parameters fit by **Mir et al., 2017**, for the specific binding at the boundary. While the distributions are close, our simulated distribution is not exponential, as expected for the 6-binding site architecture. The non-exponential behavior in the experimental curve is likely masked by the convolution with non-specific binding. We use the boundary region concentration $L = 5.6$ $\mu m^{-3}$ (see panel **b**, $k = 2$ for rates).

average decision time for different values of $k$ and $H$ is implemented using a mixed strategy of multiple random starting points and steepest gradient descent (*Figure 4a*).

## Logic and properties of the identified fast decision architectures

The main conclusion we reach using the methodology presented in the 'Methodological setup' section is that there exist promoter architectures that reach the required precision within a few minutes and satisfy all the additional embryological constraints that were discussed previously (*Figure 4a*). The fastest promoters (blue crosses in *Figure 4a*) reach a decision within the time of nuclear cycle 11 (green line in *Figure 4a*) for a wide range of activation rules. Even imposing steep readouts ($H>4$) allows us to identify relatively fast promoters, although imposing the nuclear cycle time limit, pushes the activation rule to smaller $k$. Interestingly, we find that the fastest architectures identified perform well over a range of high enough concentrations (*Appendix 4—figure 1*). The optimal architectures differ mainly by the distribution of their unbinding rates (*Figure 4b*). We shall now discuss their properties, namely the binding times of Bicoid molecules to the DNA binding sites, and the dependence of the promoter activity on various features, such as activation rules and the number of binding sites in detail. Together, these results elucidate the logic underlying the process of fast decision-making.

### How many nuclei make a mistake?

The precision of a stochastic readout process is defined by two parameters: the resolution of the readout $\delta x$, and the probability of error, which sets the reproducibility of the readout. In *Figure 4*, we have used the statistical Gaussian error level (32%) to obtain our results. However, the error level sets a crucial quantity for a developing organism and it is important to connect it with the embryological process, namely how many nuclei across the embryo will fail to properly decide (whether they are positioned in the anterior or in the posterior part of the embryo). To make this connection, we compute this number for a given average decision time $t$ and we integrate the error probability along the AP axis to obtain the error per nucleus $e_s$. The expected number of nuclei that fail to correctly identify their position is given by $\langle n_{error} \rangle = e_s 2^{c-1}$, where $c$ is the nuclear cycle and we have neglected the loss due to yolk nuclei remaining in the bulk and arresting their divisions after cycle 10 (*Foe and Alberts, 1983*). Assuming a 270 s readout time – the total interphase time of nuclear cycle 11 (*Tran et al., 2018*) – for the fastest architecture identified above and an error rate of 32%, we find that $\langle n_{error} \rangle \simeq 0.3$, that is an essentially fail-proof mechanism. This number can be compared with >30 nuclei in the embryo that make an error in a $\langle T \rangle = 270s$ read-out in a Berg-Purcell-like fixed-time scheme (integrated blue area in *Figure 1c*).

Conversely, for a given architecture, reducing the error level increases drastically the mean first-passage time to decision: the mean time for decision as a function of the error rate for the fastest architecture identified with $H>5$ and $k=1$ is shown in *Appendix 2—figure 1*. The decision can be made in about a minute for $e=32\%$ but requires on average 10 min for $e=10\%$ (*Appendix 2—figure 1*). Note that, because the mean first-passage depends simply on the inverse of the drift per cycle (*Equation 4*), the relative performance of two architectures is the same for any error rate so that the fastest architectures identified in *Figure 4a* are valid for all error levels.

Just like for the fixed-time strategy (*Figure 1c and d*), nuclei located in the mid-embryo region are more likely to make mistakes and take longer on average to trigger a decision (*Appendix 4—figure 2*).

### Residence times among the various states

As shown in *Tran et al., 2018*, high Hill coefficients in the *hunchback* response are associated with frequent visits of the extreme expression states where available binding sites are either all empty (state 0), or all occupied (state 6). *Figure 4c* provides a concrete illustration by showing the distribution of residence times for the promoter architectures that yield the fastest decision times for $k=3$ and no constraints (blue bars), $H>4$ (red bars) and $H>5$ (black bars). When there are no constraints on the slope of the *hunchback* response, the most frequently occupied states are close to the ON-OFF transition (2 and 3 occupied binding sites in *Figure 4c*) to allow for fast back and forth between the active and inactive states of the gene and thereby gather information more rapidly by reducing $\tau_{ON} + \tau_{OFF}$ (see formulae 2 and 4).

We notice that for higher Hill coefficients, the system transits quickly through the central states (in particular states with 3 and 4 occupied Bcd sites, *Figure 4* red and black bars). As expected for high Hill coefficients, such dynamics requires high cooperativity. Cooperativity helps the recruitment of extra transcription factors once one or two of them are already bound and thus speeds up the transitioning through the states with 2, 3 and 4 occupied binding sites. An even higher level of cooperativity is required to make TF DNA binding more stable when 5 or 6 of them are bound, reducing the OFF rates $\nu_5$ or $\nu_6$ (*Figure 4b*).

## The (short) binding times of Bicoid on DNA

The distribution of times spent bound to DNA of individual Bicoid molecules is shown in *Figure 4d* obtained from Monte Carlo simulations using rates from the fastest architecture with $H>5$ and $k = 2$. We find an exponential decay, an average bound time of about 7.1 s and a median around 0.5 s. Our median-time-bound prediction is of the same order of magnitude as the observed bound times seen in recent experiments by *Mir et al., 2017*; *Mir et al., 2018*, who found short (mean ~0.629 s and median ~0.436 s based on exponential fits), yet quite distinguishable from background, bound times to DNA. These results were considered surprising because it seemed unclear how such short binding events could be consistent with the processing of ON and OFF gene switching events. Our results show that such short binding times may actually be instrumental in achieving the tradeoff between accuracy and speed, and rationalize how longer activation events are still achieved despite the fast binding and unbinding. High cooperativity architectures lead to non-exponential bound times to DNA (*Figure 4d*) for which the typical bound time (median) is short but the tail of the distribution includes slower dynamics that can explain longer activation events (the mean is much larger than the median). This result suggests that cells can use the bursty nature of promoter architectures to better discriminate between TF concentrations.

In *Mir et al., 2017*, the raw distribution comprises both non-specific and specific binding and cannot be directly compared to simulation results. Instead, we use the largest of the two exponents fit for the boundary region (*Mir et al., 2017*), which should correspond to specific binding. The agreement between the distributions in *Figure 4d* is overall good, and we ascribe discrepancies to the fact that (*Mir et al., 2017*) fit two exponential distributions assuming the observed times were the convolution of exponential specific and non-specific binding times. Yet the true specific binding time distribution is likely not exponential, e.g. due to the effect of binding sites having different binding affinities. We show in *Appendix 5—figure 1* that the two distributions are very similar and hard to distinguish once they are mixed with the non-specific part of the distribution.

## Activation rules

In the parameter range of the early fly embryo, the fastest decision-making architectures share the one-or-more ($k = 1$) activation rule : the promoter switches rapidly between the ON and OFF expression states and the extra binding sites are used for increasing the size of the target rather than building a more complex signal. Architectures with $k = 2$ and $k = 3$ activation rules can make decisions in less than 270 s and satisfy all the required biological constraints. Generally speaking, our analysis predicts that fast decisions require a small number of Bicoid-binding sites (less than three) to be occupied for the gene to be active. The advantage of the $k = 2$ or $k = 3$ activation rules is that the ON and OFF times are on average longer than for $k = 1$, which makes the downstream processing of the readout easier. We do not find any architecture satisfying all the conditions for the $k = 4, 5, 6$ activation rules, although we cannot exclude there could be some architectures outside of the subset that we managed to sample, especially for the $k = 4$ activation rules where we did identify some promoter structures that are close to the time constraint.

Activation rules with higher $k$ can give higher information per cycle for the ON rate, yet they do not seem to lead to faster decisions because of the much longer duration of the cycles. To gain insight on how the tradeoff between fast cycles and information affects the efficiency of activation rules, we consider architectures with only two binding sites, which lend to analytical understanding (*Figure 5a and b*). Both of these considered architectures are out of equilibrium and require energy consumption (as opposed to the two equilibrium architectures of *Figure 5c and d*).

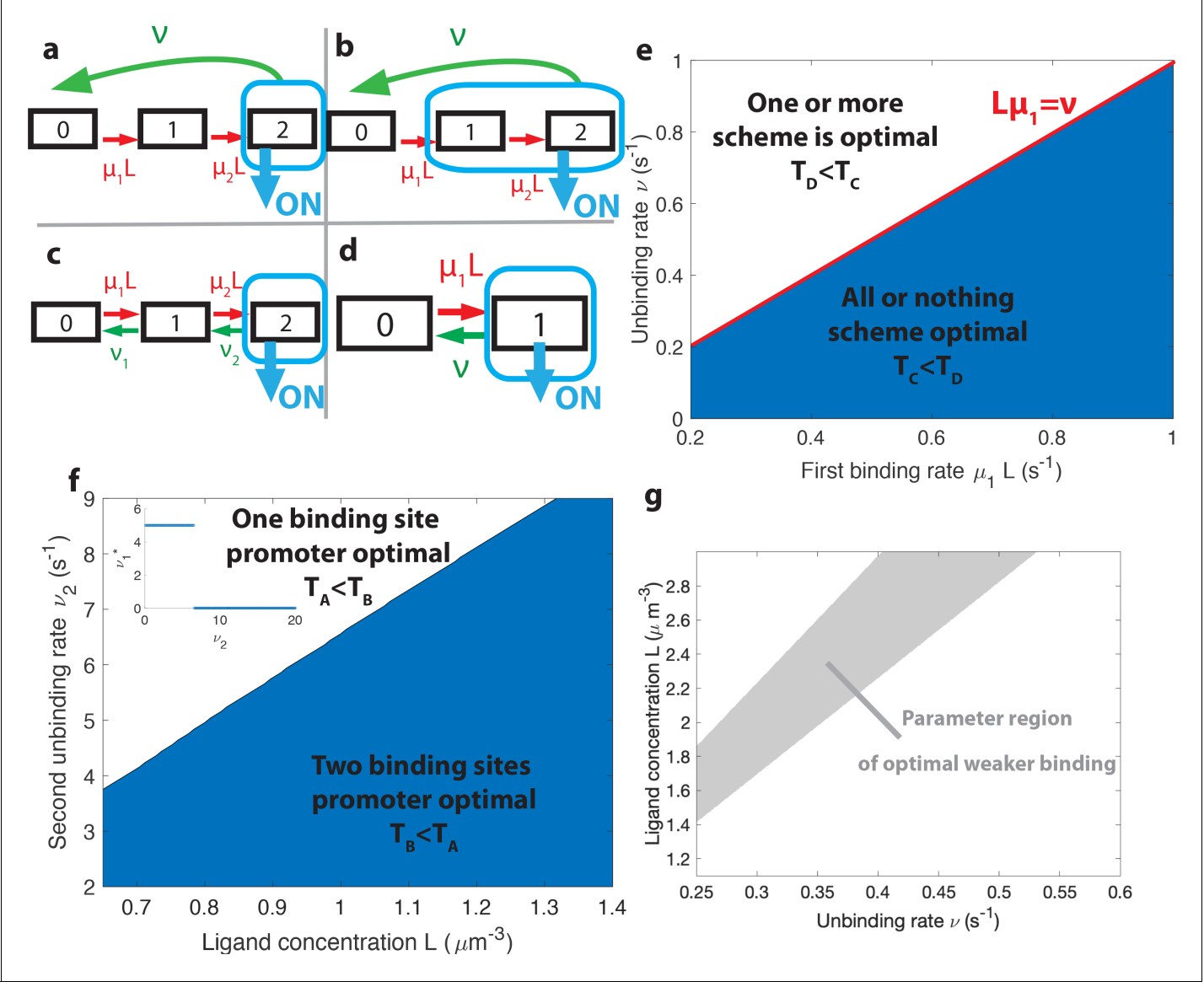

**Figure 5.** The effects of different promoter architectures on the mean decision time. We compare promoters of different complexity: the all-or-nothing $k = 2$ out-of-equilibrium model (a), the 1-or-more $k = 1$ out-of-equilibrium model (b), the two binding site all-or-nothing $k = 2$ equilibrium model (c) and the one binding site equilibrium model (d). (e) Comparison of the mean decision time between $k = 2$ (a) and $k = 1$ (b) activation schemes for the two binding site out-of-equilibrium models as a function of the unbinding rate $\nu$ and binding rate $\mu_1$. The binding rate $\mu_2$ is fixed to $\mu_2 = \mu_1/2$. The fastest decision-making solution associates the second binding with slower variables to maximize $V$. Along the line $L\mu_1 = \nu$ the activation rules $k = 1$ and $k = 2$ perform at the same speed. $e = 10\%$. (f) Comparison of the mean decision time for equilibrium architectures with one (d) and two (c) equilibrium TF-binding sites. $\mu_1, \mu_2, \nu_1$ are optimized at fixed $\nu_2$, $e = 5\%$. We set a maximum value of $5\ \mu m^3 s^{-1}$ for $\mu_1$ and $\mu_2$, corresponding to the diffusion limited arrival at the binding site. For $\nu_1$, the maximum value of $5\ s^{-1}$ corresponds to the inverse minimum time required to read the presence of a ligand, or to differentiate it from unspecific binding of other proteins. Additional binding sites are beneficial at high ligand concentrations and for small unbinding rates. In the blue region, the fastest mean decision time for a fixed accuracy assuming equilibrium binding, comes from a two binding site architecture with a non-zero first unbinding rate. In the white region, one of the binding rates $\rightarrow 0$ (see inset), which reduces to a one binding site model. $e = 5\%$. (g) Weaker binding sites can lead to faster decision times within a range of parameters (gray stripe). We consider the $k = 1$ activation scheme with two binding sites (b). For fixed L ($x$ axis), $\nu$ ($y$ axis) and $\mu_1 = 0.2\ \mu m^3 s^{-1}$, we optimize over $\mu_2$ while setting $\mu_2 < \mu_1$ (context of diffusion limited first and second bindings). The gray regions corresponds to parameters for which the optimal second unbinding rate $\mu_2^* < \mu_1$ and the second binding is weak. In the white region $\mu_2^* = \mu_1$. For all panels $L_2 = L$, $L_1 = 0.95L$, $e = 0.05$.

## When is 1-or-more faster than all-or-nothing activation?

A first model has the promoter consisting of two binding sites with the all-or-nothing rule $k = 2$ (*Figure 5a*). We consider the mathematically simpler, although biologically more demanding, situation where TFs cannot unbind independently from the intermediate states – once one TF binds, all the binding sites need to be occupied before the promoter is freed by the unbinding with rate ν of the entire complex of TFs. This situation can be formulated in terms of a non-equilibrium cycle model, depicted for two binding sites in *Figure 5a*. The activation time is a convolution of the exponential distributions $P_{\mathrm{OFF}}(t, L) = \frac{\mu_1 \mu_2 L}{\mu_2 - \mu_1} (e^{-\mu_1 Lt} - e^{-\mu_2 Lt})$. In the simple case, when the two binding rates are similar ($\mu_1 \simeq \mu_2$), the OFF times follow a Gamma distribution and the drift and diffusion can be computed analytically (see Appendix 4). When the two binding rates are not similar the drift and diffusion must be obtained by numerical integration (see Appendix 4).

In the first model described above (*Figure 5a*), deactivation times are independent of the concentration and do not contribute to the information gained per cycle and, as a result, to $V/(\tau_{\mathrm{ON}} + \tau_{\mathrm{OFF}})$. To explore the effect of deactivation time statistics on decision times, we consider a cycle model where the gene is activated by the binding of the first TF (the 1-or-more $k = 1$ rule) and deactivation occurs by complete unbinding of the TFs complex (*Figure 5b*). The resulting activation times are exponentially distributed and contribute to drift and diffusion as in the simple two state promoter model (*Figure 5d*). The deactivation times are a convolution of the concentration-dependent second binding and the concentration-independent unbinding of the complex and their probability distribution is $P_{\mathrm{ON}}(t, L) = \frac{\nu \mu_2 L}{\nu - \mu_2 L} (e^{-\mu_2 Lt} - e^{-\nu t})$. Drift and diffusion can be obtained analytically (Appendix 4). The concentration-dependent deactivation times prove informative for reducing the mean decision time at low TF concentrations but increase the decision time at high TF concentrations compared to the simplest irreversible binding model. In the limit of unbinding times of the complex ($1/\nu$) much larger than the second binding time ($1/\mu_2 L$), no information is gained from deactivation times. In the limit of $\mu_1 L/\nu \to \infty$, the $k = 1$ model reduces to a one binding site exponential model and the two architectures (*Figure 5b and d*) have the same asymptotic performance.

Within the irreversible schemes of *Figure 5a* and *Figure 5b*, the average time of one activation/deactivation cycle is the same for the all-or-nothing $k = 2$ and 1-or-more $k = 1$ activation schemes. The difference in the schemes comes from the information gained in the drift term $V/(\tau_{\mathrm{ON}} + \tau_{\mathrm{OFF}})$, which begs the question : is it more efficient to deconvolve the second binding event from the first one within the all-or-nothing $k = 2$ activation scheme, or from the deactivation event in the $k = 1$ activation scheme?

In general, the convolution of two concentration-dependent events is less informative than two equivalent independent events, and more informative than a single binding event. For small concentrations $L$, activation events are much longer than deactivation events. In the $k = 1$ scheme, OFF times are dominated by the concentration-dependent step $\mu_2 L$ and the two activation events can be read independently. This regime of parameters favors the $k = 1$ rule (*Figure 5e*). However, when the concentration $L$ is very large the two binding events happen very fast and for $\mu_2 L \gg \nu$, in the $k = 1$ scheme, it is hard to disentangle the binding and the unbinding events. The information gained in the second binding event goes to 0 as $L \to \infty$ and the one-or-more $k = 1$ activation scheme (*Figure 5b*) effectively becomes equivalent to a single binding site promoter (*Figure 5d*), making the all-or-nothing $k = 2$ activation (*Figure 5a*) scheme more informative (*Figure 5e*). The fastest decision time architecture systematically convolves the second binding event with the slowest of the other reactions (*Figure 5e*), with the transition between the two activation schemes when the other reactions have exactly equal rates ($\mu_1 L = \nu$ line in *Figure 5e*) (see Appendix 6 for a derivation).

## How the number of binding sites affects decisions

The above results have been obtained with six binding sites. Motivated by the possibility of building synthetic promoters (*Park et al., 2019*) or the existence of yet undiscovered binding sites, we investigate here the role of the number of binding sites. Our results suggest that the main effect of additional binding sites in the fly embryo is to increase the size of the target (and possibly to allow for higher cooperativity and Hill coefficients). To better understand the influence of the number of binding sites on performance at the diffusion limit, we compare a model with one binding site (*Figure 5d*) to a reversible model with two binding sites where the gene is activated only when both binding sites are bound (all-or-nothing $k = 2$, *Figure 5c*). Just like for the six binding site

architectures, we describe this two binding site reversible model by using the transition matrix of the $N + 1$ Markov chain and calculate the total activation time $P_{\text{OFF}}(t, L)$.

For fixed values of the real concentration $L$, the two hypothetical concentrations $L_1$ and $L_2$, the error $e$ and the second off-rate $\nu_2$, we optimize the remaining parameters $\mu_1$, $\mu_2$ and $\nu_1$ for the shortest average decision time.

For high gene deactivation rates $\nu_2$, the fastest decision time is achieved by a promoter with one binding site (*Figure 5f*): once one ligand has bound, the promoter never goes back to being completely unbound ($\nu_1^* = 0$ in *Figure 5c*) but toggles between one and two bound TF (*Figure 5d* with $\nu = \nu_2$ and $\mu = \mu_2$). For lower values of gene deactivation rates $\nu_2$, there is a sharp transition to a minimal $\langle T \rangle$ solution using both binding sites. In the all-or-nothing activation scheme that is used here, the distribution of deactivation times is ligand independent and the concentration is measured only through the distribution of activation times, which is the convolution of the distributions of times spent in the 0 and 1 states before activation in the two state. For very small deactivation $\nu_2$ rates, it is more informative to 'measure' the ligand concentration by accumulating two binding events every time the gene has to go through the slow step of deactivating (*Figure 5c*). However, for large deactivation rates little time is 'lost' in the uninformative expressing state and there is no need to try and deconvolve the binding events but rather use direct independent activation/deactivation statistics from a single binding site promoter (*Figure 5d*, see Appendix 7 for a more detailed calculation).

## The role of weak binding sites

An important observation about the strength of the binding sites that emerge from our search is that the binding rates are often below the diffusion limit $\mu_{\max} L_0 \simeq 0.124 s^{-1}$ (see black dashed line in *Figure 4b*) : some of the ligands reach the receptor, they could potentially bind but the decision time decreases if they do not. In other words, binding sites are 'weak' and, since this is also a feature of many experimental promoters (*Gertz et al., 2009*), the purpose of this section is to investigate the rationale for this observation by using the models described in *Figure 5*.

Naively, it would seem that increasing the binding rate can only increase the quality of the readout. This statement is only true in certain parameter regimes, and weaker binding sites can be advantageous for a fast and precise readout. To provide concrete examples, we fix the deactivation rate $\nu$ and the first binding rate $\mu_1$ in the 1-or-more irreversible binding model of *Figure 5b* and we look for the unbinding rate $\mu_2^*$ that leads to the fastest decision. We consider a situation where the two binding sites are not interchangeable and binding must happen in a specific order. In this case, the diffusion limit states that $\mu_2 \leq \mu_1$ if the first binding is strong and happens at the diffusion limit. We optimize the mean decision time for $0 \leq \mu_2 \leq \mu_1$ (see *Appendix 9—figure 1* for an example) and find a range of parameters where the fastest-decision value $\mu_2^* < \mu_1$ is not as fast as parameter range allows (*Figure 5g*). We note that this weaker binding site that results in fast decision times can only exist within a promoter structure that features cooperativity. In this specific case, the first binding site needs to be occupied for the second one to be available. If the two binding sites are independent, then the diffusion limit is $\mu_2 \leq \mu_1$ and the fastest $\langle T \rangle$ solution always has the fastest possible binding rates.

## Predictions for Bicoid-binding sites mutants

In addition to results for wild type embryos, our approach also yields predictions that could be tested experimentally by using synthetic *hb* promoters with a variable numbers of Bicoid-binding sites (*Figure 6a*). For any of the fast decision-making architectures identified and activation rules chosen, we can compute the effects of reducing the number of binding sites. Specifically, our predictions for the $k = 3$ activation rule and $H > 4$ in *Figure 6b* can be compared to FISH or fluorescent live imaging measurements of the fraction of active loci at a given position along the anterior-posterior axis. Bcd-binding site mutants of the WT promoter have been measured by immunostaining in cycle 14 (*Park et al., 2019*), although mRNA experiments in earlier cell cycles of well characterized mutants are needed to provide for a more quantitative comparison.

An important consideration for the comparison to experimental data is that there is a priori no reason for the *hb* promoter to have an *optimal* architecture. We do find indeed many architectures that satisfy all the experimental constraints and are not the fastest decision-making but 'good

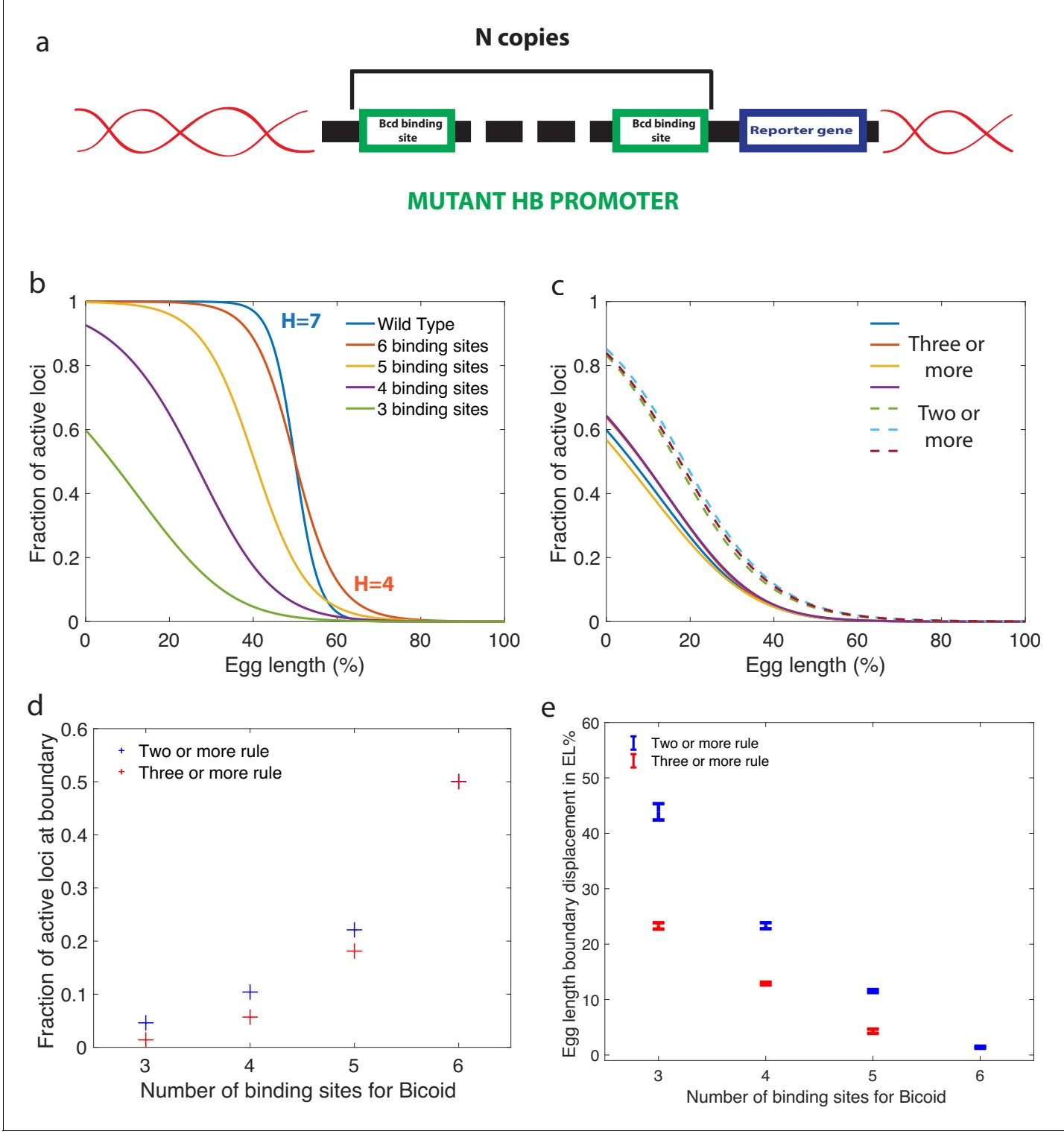

**Figure 6.** Predictions for experiments with synthetic *hb* promoters. (**a**) We consider experiments involving mutant *Drosophilae* where a copy of a subgroup of the Bicoid-binding sites of the *hunchback* promoter is inserted into the genome along with a reporter gene to measure its activity. (**b**) The prediction for the activation profile across the embryo for wild type and mutants for the fastest decision time architecture for $H>4$ and $k=3$. (**c**) The gene activation profile for several architectures for $H>4$ and $k=3$ (full lines) and $k=2$ (dashed lines) that results in mean decision times < 3 minutes. Groups of profiles gather in two distinct clusters. (**d**) Fraction of genes that are active on average at the *hb* expression boundary using the minimal $\langle T \rangle$ architecture identified for $H>4$ and $k=3$ as a function of the number of binding sites in the *hb* promoter. Predictions for the six-binding site cases coincide because having half the nuclei active at the boundary is a requirement in the search for valid architectures. (**e**) Predicted displacement of the

*Figure 6 continued on next page*

*Figure 6 continued*

boundary region defined as the site of half *hunchback* expression in terms of egg length as a function of the number of binding sites. The architectures shown result in the fastest decisions for $H>4$ and $k=2$ and $k=3$. Error bar width is the standard deviation of the various architectures that are close to minimal $\langle T \rangle$. For all panels, $L(x)$ has an exponentially decreasing profile with decay length one fifth of total egg length with $L_0 = 5.6 \ \mu m^{-3}$ at the boundary. Parameters are given in Appendix 10.

enough' *hb* promoters. A relevant question then is whether or not similarity in performance is associated with similarity in the microscopic architecture. This point is addressed in *Figure 6c*, where we compare the fraction of active loci along the AP axis using several constraint-conforming architectures for $H>4$ and the $k=2$ and $k=3$ activation rules. The plot shows that solutions corresponding to the same activation rule are clustered together and quite distinguishable from the rest. This result suggests that the precise values of the binding and unbinding constants are not important for satisfying the constraints, that many solutions are possible, and that FISH or MS2 imaging experiments can be used to distinguish between different activation rules. The fraction of active loci in the boundary region is an even simpler variable that can differentiate between different activation rules (*Figure 6d*). Lastly, we make a prediction for the displacement of the anterior-posterior boundary in mutants, showing that a reduced numbers of Bcd sites results in a strong anterior displacement of the *hb* expression border compared to six binding sites, regardless of the activation rule (*Figure 6e*). Error bars in *Figure 6e*, that correspond to different close-to-fastest architectures, confirm that these share similar properties and different activation rules are distinguishable.

## Joint dynamics of Bicoid enhancer and promoter

The Bicoid transcription factor has been shown to target more than a thousand enhancer loci in the *Drosophila* embryo with a wide concentration range of sensitivities (*Driever and Nüsslein-Volhard, 1988*; *Struhl et al., 1989*; *Hannon et al., 2017*). Enhancers are of special interest because they can be located far away from the promoter (*Ribeiro et al., 2010*; *Krivega and Dean, 2012*) and perform a statistically independent sample of the concentration that is later combined with that of the promoter. Evidence suggests that promoter-based conformational changes can be stable over long times (*Fukaya et al., 2016*), which mimics information storage during a process of signal integration. To explore these effects, we consider a simple model of enhancer dynamics where a Bicoid-specific enhancer switches between two states ON and OFF independently of the promoter dynamics. We assume a simple rule for the gene activity: the gene is transcribed when both the promoter and the enhancer are ON. As an example, we consider an enhancer made of one binding site so that the ON rate $e_{ON}$ of the enhancer is limited by the diffusion rate $\mu_{max}L$. As an example, we perform a parameter search for the promoter activation rule $k=2$ (see Appendix 12), while still assuming that about half the nuclei are active at the boundary and a required Hill coefficient greater than 4, looking for architectures yielding the shortest decision time for an error rate of 32% for the 10% relative concentration difference discrimination problem. We find that the enhancer improves the performance of the readout, reducing the time to decision by about $\simeq 6\%$. We find that adding extra binding sites to the promoter increases the computing power of the enhancer-promoter system and can reduce the time to decision to about 60% of the performance of the best architectures without enhancers.

## Estimating the log-likelihood function with RNA concentrations

To illustrate how a biochemical network can approximate the calculation of the log-likelihood, we consider the case of the fastest architecture identified in the paragraph 'Numerical procedure for identifying fast decision-making architectures' with $k=1$ and $H>4$. In *Figure 7a*, we show the contributions of the OFF times (blue) and the ON times (red) to the log-likelihood for this architecture. We notice that the behavior of the log-likelihood contributions at long times is simply linear in time with a positive rate for ON times and a negative rate for OFF times. Conversely, short ON times contribute negatively to the log-likelihood while short OFF times contribute positively to the log-likelihood (*Figure 7a*). This observation suggests a simple model of RNA production with delay to approximate the computation of the log-likelihood. We consider a model of RNA production with five parameters (*Figure 7b*, details of the model are given in Appendix 11). We assume that when the promoter is in the ON state, polymerase is loaded and RNA is transcribed at a constant rate $r_{ON}$ while when the promoter is in the OFF state, RNA is produced at a lower basal rate $r_b$. In order to approximate the

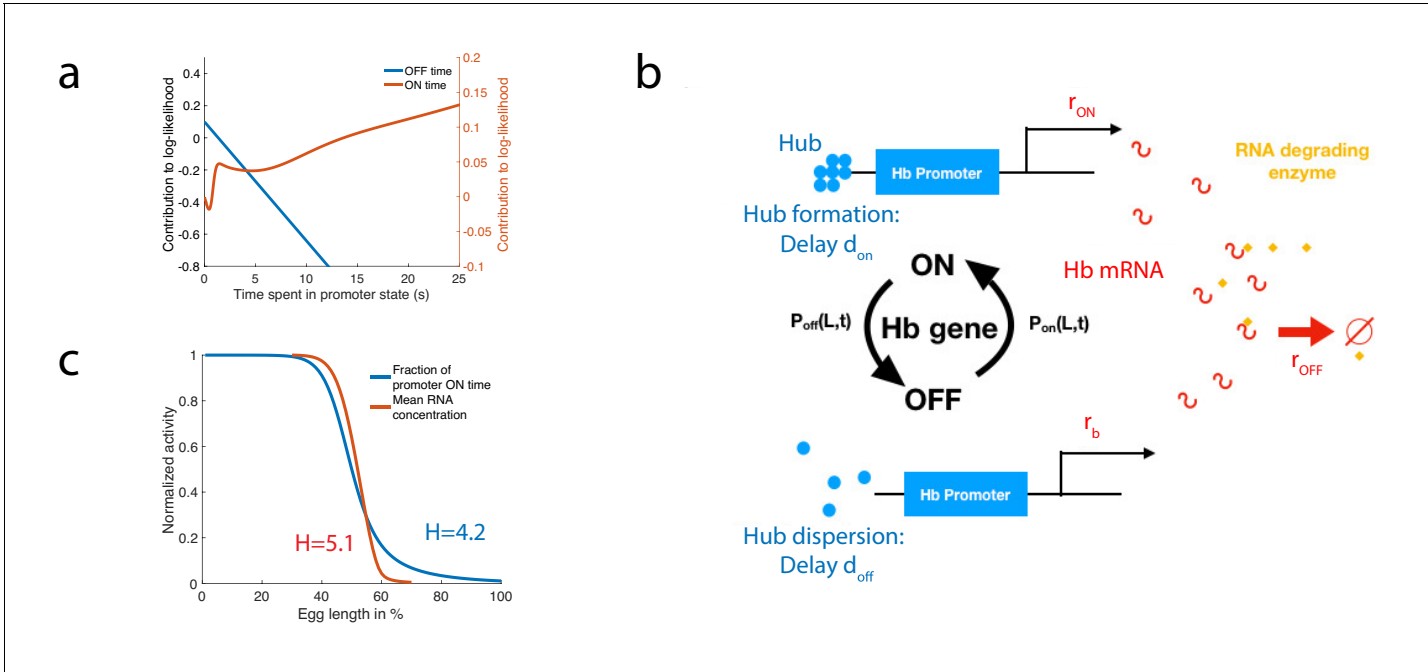

**Figure 7.** A model of RNA production and degradation approximates the contributions to log-likelihood. (a) The log-likelihood of different times spent ON (red) and OFF (blue) for the fastest architecture identified with $k = 1$, $H>4$ assuming $L = L_1$. The log-likelihood varies linearly with time for long times. (b) The hunchback promoter switches from ON to OFF and from OFF to ON according to the time distributions determined by its gene architecture and activation rule. When ON, after a delay $d_{ON}$ associated with the formation of a cluster or hub (*Cho et al., 2016b*; *Cho et al., 2016a*; *Mir et al., 2018*), RNA is being produced at rate $r_{ON}$. When OFF, after a delay $d_{OFF}$, the gene switches to basal rate $r_b$. Hunchback RNA is being degraded actively by an enzyme at rate $r_{OFF}$. The RNA is in excess for this reaction. (c) The model of RNA production with delay that yields an error of less than 32% in less than 3 min produces a profile of RNA production with high Hill coefficient $\simeq 5.2$ (red lines, renormalized RNA profile) that is higher than the Hill coefficient of renormalized gene activity $H \simeq 4$ (blue line). Parameters for promoter activity are those of the fastest architecture identified with $k = 1$ and $H>4$, parameters for RNA production are $d_{ON} = 0.047s$, $d_{OFF} = 1.6s$, $r_b = 0.2\text{s}^{-1}$, $r_{OFF} = 0.5\text{s}^{-1}$, $r_{ON} = 0.805\text{s}^{-1}$.

linear behavior of the log-likelihood function at long times we suggest the existence of an enzyme actively degrading hunchback RNA (*Chanfreau, 2017*) and include in our model of RNA production the delay $d_{ON}$ and inertia $d_{OFF}$ associated with promoter dynamics and polymerase loading. RNA levels fluctuate and trigger decisions when the concentration $[\text{RNA}]$ is greater than a threshold $c_1(t)$ (anterior decision) or lower than a threshold $c_2(t)$ (posterior decision). Since many forms of switch have already been presented in the literature (*Goldbeter and Koshland, 1981*; *Tyson et al., 2003*; *Ozbudak et al., 2004*; *Siggia and Vergassola, 2013*; *Sandefur et al., 2013*), we shall concentrate on the log-likelihood calculation and refer to previous references for the implementation of a decision when reaching a threshold.

We look for parameters that satisfy both a high speed and high accuracy requirement for a decision between two points located 2% egg lengths apart across the mid-embryo boundary. For the fastest architecture identified with $k = 1$ and $H>4$, we identify parameters that satisfy $e<32\%$ and a mean decision time $T<3$ min (see *Appendix 11—figure 1*). We check that this model produces a profile of RNA that is consistent with the observed high Hill coefficient (*Figure 7c*). Interestingly, we find that for this particular set of parameters the RNA profile Hill coefficient is increased by the delayed transcription dynamics and the active degradation from $\simeq 4$ (blue line in *Figure 7c*) up to $\simeq 5.2$ (red line in *Figure 7c*, details of the calculation are given in Appendix 11). This result could shed new light on the fundamental limits to Hill coefficients in the context of cooperative TF binding (*Estrada et al., 2016*; *Tran et al., 2018*) and provide a possible mechanism to explain how mRNA profiles can reach higher steepness than the corresponding TF activities do. We also looked for parameter sets that approximate the log-likelihood for the optimal architecture identified for $k = 2$ and $H>4$ and find several candidates that fall close to the requirement of speed and accuracy (*Appendix 11—figure 1*). Together these results show that implementing the log-likelihood using a

molecular circuit with a hb promoter is possible. They do not show this is what is happening in the embryo itself.

## Discussion

The issue of precision in the Bicoid readout by the *hunchback* promoter has a long history (*Nüsslein-Volhard and Wieschaus, 1980*; *Tautz, 1988*). Recent interest was sparked by the argument that the amount of information at the *hunchback* locus available during one nuclear cycle is too small for the observed 2% EL distance between neighboring nuclei that are able to make reproducible distinct decisions (*Gregor et al., 2007b*). By using updated estimates of the biophysical parameters (*Porcher et al., 2010*; *Tran et al., 2018*), and the Berg-Purcell error estimation, we confirm that establishing a boundary with 2% variability between neighbouring nuclei would take at least about 13.4 min – roughly the non-transient expression time in nuclear cycle 14 (*Lucas et al., 2018*; *Tran et al., 2018*) (Appendix 1). This holds for a single Bicoid-binding site. An intuitive way to achieve a speed up is to increase the number of binding sites: multiple occupancy time traces are thereby made available, which provides a priori more information on the Bicoid concentration.

Possible advantages of multiple sites are not so easy to exploit, though. First, the various sites are close and their respective bindings are correlated (*Kaizu et al., 2014*), so that their respective occupancy time traces are not independent. That reduces the gain in the amount of information. Second, if the activation of gene expression requires the joint binding of multiple sites, the transition to the active configuration takes time. The overall process may therefore be slowed down with respect to a single binding site model, in spite of the additional information. Third, and most importantly, information is conveyed downstream via the expression level of the gene, which is again a single time trace. This channeling of the multiple sites' occupancy traces into the single time trace of gene expression makes gene activation a real information bottleneck for concentration readout. All these factors can combine and even lead to an increase in the decision time. To wit, an all-or-nothing equilibrium activation model with six identical binding sites functioning at the diffusion limit and no cooperativity takes about 38 min to achieve the same above accuracy. In sum, the binding site kinetics and the gene activation rules are essential to harness the potential advantage of multiple binding sites.

Our work addresses the question of which multisite promoters architecture minimize the effects of the activation bottleneck. Specifically, we have shown that decision schemes based on continuous updating and variable decision times significantly improve speed while maintaining the desired high readout accuracy. This should be contrasted to previously considered fixed-time integration strategies. In the case of the *hunchback* promoter in the fly embryo, the continuous update schemes achieve the 2% EL positional resolution in less than 1 min, always outperforming fixed-time integration strategies for the same promoter architecture (see *Appendix 1—table 1*). While 1 min is even beyond what is required for the fly embryo, this margin in speed allows to accommodate additional constraints, viz. steep spatial boundary and biophysical constraints on kinetic parameters. Our approach ultimately yields many promoter architectures that are consistent with experimental observables in fly embryos, and results in decision times that are compatible with a precise readout even for the fast nuclear cycle 11 (*Lucas et al., 2018*; *Tran et al., 2018*).

Several arguments have been brought forward to suggest that the duration of a nuclear cycle is the limiting time period for the readout of Bicoid concentration gradient. The first one concerns the reset of gene activation and transcription factor binding during mitosis. In that sense, any information that was stored in the form of Bicoid already bound to the gene is lost. The second argument is that the hunchback response integrated over a single nuclear cycle is already extremely precise. However, none of these imply that the *hunchback* decision is made at a fixed-time (corresponding to mitosis) so that strategies involving variable decision times are quite legitimate and consistent with all the known phenomenology.

We have performed our calculations in a worst-case scenario. First, we did not consider averaging of the readout between neighbouring nuclei. While both protein (*Gregor et al., 2007a*) and mRNA concentrations (*Little et al., 2013*) are definitely averaged, and it has been shown theoretically that averaging can both increase and decrease (*Erdmann et al., 2009*) readout variability between nuclei, we do not take advantage of this option. The fact that we achieve less than 3 min in nuclear cycle 11, demonstrates that averaging is a priori dispensable. Second, we demand that the

*hunchback* promoter results in a readout that gives the positional resolution observed in nuclear cycle 14, in the time that the *hunchback* expression profile is established in nuclear cycle 11. The reason for this choice is twofold. On the one hand, we meant to show that such a task is possible, making feasible also less constrained set-ups. On the other hand, the *hunchback* expression border established in nuclear cycle 11 does not move significantly in later nuclear cycles in the WT embryo, suggesting that the positional resolution in nuclear cycle 11 is already sufficient to reach the precision of later nuclear cycles. The positional resolution that can be observed in nuclear cycle 11 at the gene expression level is $\sim 10\%$ EL (*Tran et al., 2018*), but this is also due to smaller nuclear density.

Two main factors generally affect the efficiency of decisions: how information is transmitted and how available information is decoded and exploited. Decoding depends on the representation of available information. Our calculations have considered the issue of how to convey information across the bottleneck of gene activation, under the constraint of a given Hill coefficient. The latter is our empirical way of taking into account the constraints imposed by the decoding process. High Hill coefficients are a very convenient way to package and represent positional information: decoding reduces to the detection of a sharp transition, an edge in the limit of very high coefficients. The interpretation of the Hill coefficient as a decoding constraint is consistent with our results that an increase in the coefficient slows down the decision time. The resulting picture is that promoter architecture results from a balance between the constraints imposed by a quick and accurate readout and those stemming from the ease of its decoding. The very possibility of a balance is allowed by the main conclusion demonstrated here that promoter structures can go significantly below the time limit imposed by the duration of the early nuclear cycles. That leaves room for accommodating other features without jeopardising the readout timescale. While the constraint of a fixed Hill coefficient is an effective way to take into account constraints on decoding, it will be of interest to explore in future work if and how one can go beyond this empirical approach. That will require developing a joint description for transmission and decoding via an explicit modeling of the mechanisms downstream of the activation bottleneck.

Recent work has shed light on the role of out of equilibrium architectures on steepness of response (*Estrada et al., 2016*) and gradient establishment (*Tran et al., 2018*; *Park et al., 2019*). Here, we showed that equilibrium architectures perform very well and achieve short decision times, and that out of equilibrium architectures do not seem to significantly improve the performance of promoters, except for making some switches from gene states a bit faster. Non-equilibrium effect can, however, increase the Hill coefficient of the response without adding extra binding sites, which is useful for the downstream readout of positional information that we formulated above as decoding.

We also showed how short bound times of Bicoid molecules to the DNA (*Mir et al., 2017*; *Mir et al., 2018*) are translated into accurate and fast decisions. Our fast decision-making architectures also display short DNA-bound times. However, the constraint of high cooperativity means that the distribution of bound times to the DNA is non-exponential and the rare long binding times that occur during the bursty binding process are exploited during the read-out. The combination of high cooperativity and high temporal variance due to bursty dynamics is a possible recipe for an accurate readout.

At the technical level, we developed new methods for the mean decision time of complex gene architectures within the framework of variable time decision-making (SPRT). This allowed us to establish the equality $V = D$ between drift and diffusion of the log-likelihood between two close hypotheses. Its underlying reason is the martingale property that the conditional expectation of probabilities for two hypotheses, given all prior history, is equal to their present value. The methodology developed here will be useful for the broad range of decision processes where SPRT is relevant, including neuroscience (*Gold and Shadlen, 2007*; *Bitzer et al., 2014*) and synthetic biology (*O'Brien and Murugan, 2019*; *Pittayakanchit et al., 2018*).

We made predictions about how promoter architectures with different activation schemes can be compared in synthetic embryos with different numbers of Bcd binding sites. Furthermore, experiments that change the composition of the syncytial medium would influence the diffusion constant and assay the assumption of diffusion-limited activation. Our model predicts that these changes would result in modifications of *hunchback* activation profiles: higher or lower diffusion rates slide the *hunchback* profile towards the anterior or the posterior end of the embryo, respectively, similarly to an increase or a decrease of the number of Bicoid binding sites. Any of the above experiments

would greatly advance our understanding of the molecular control of spatial patterning in *Drosophila* embryo and, more generally, of regulatory processes.

Finally, we showed how a simple model of RNA production with delay and active degradation could easily approximate the seemingly complex log-likelihood calculation. The specific implementation that we described is tentative and aimed at simplicity, yet it illustrates several points, namely that the log-likelihood can be approximated by preserving a high Hill coefficients, observed characteristics of transcription dynamics and ensuring speed-accuracy limits. Future experiments could identify candidates for the enzyme responsible for the active degradation of RNA, or image the formation and dissolution of clusters using super-resolution imaging methods. This mechanism is also very close to the multifactor clusters (*Mir et al., 2018*) observed recently in early *Drosophila* in conjunction with active transcription sites. We suggest this mechanism as an example of implementation of the log-likelihood calculation, but note that the added complexity of the computation could happen at different levels of the expression machinery, including upstream of promoter activation through enhancer dynamics and chromatin arrangement.

## Acknowledgements

We are grateful to Gautam Reddy and Huy Tran for multiple relevant discussions. This work was supported by the National Science Foundation, Program PoLS, Grant 1411313. JD was partially supported by the NSF-Simons Center for Quantitative Biology (Simons Foundation SFARI/597491-RWC and the National Science Foundation 17764421). The funders had no role in study design, data collection and analysis, decision to publish, or preparation of the manuscript.

## Additional information

### Competing interests
Aleksandra M Walczak: Reviewing editor, *eLife*. The other authors declare that no competing interests exist.

### Funding

| Funder | Grant reference number | Author |
| --- | --- | --- |
| National Science Foundation | PoLS Grant 1411313 | Massimo Vergassola |

The funders had no role in study design, data collection and interpretation, or the decision to submit the work for publication.

### Author contributions
Jonathan Desponds, Conceptualization, Software, Validation, Investigation, Methodology; Massimo Vergassola, Aleksandra M Walczak, Conceptualization, Supervision, Validation, Investigation, Methodology

### Author ORCIDs
Jonathan Desponds https://orcid.org/0000-0001-7112-3217
Massimo Vergassola https://orcid.org/0000-0002-7212-8244
Aleksandra M Walczak http://orcid.org/0000-0002-2686-5702

### Decision letter and Author response
Decision letter https://doi.org/10.7554/eLife.49758.sa1
Author response https://doi.org/10.7554/eLife.49758.sa2

## Additional files
### Supplementary files
- Source code 1. Search for optimal architectures.

• Transparent reporting form

### Data availability

All data analyzed during this study were previously published in the literature and references are included in the paper.

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

## Appendix 1

### Biological parameters in the embryo

To build a model of the promoter, we combine parameters from recent experimental work.

The embryo at nuclear cycle $n$ is modelled as having $2^{n-1}$ nuclei/cells. For simplicity, we assume they are equidistributed on the periphery of the embryo and across the embryo's length and do not take into account the effect of the geometry of the embryo because the curvature of the embryo is small (the embryo is about 500 μm-long along the anterior-posterior axis and only about 100 μm-long along the dorso-ventral axis). We also neglect the few nuclei forming pole cells and remaining in the bulk (**Foe and Alberts, 1983**).

The Bicoid concentration in the embryo is given by $L(x) = L_0 e^{-(x-x_0)/\lambda}$, where $x$ is the position along the anterior-posterior (AP) embryo axis measured in % of the egg length, $\lambda$ is the decay length also measured in % of the egg length ($\lambda \simeq 100 \mu m$ which is roughly 20% of the egg length (**Gregor et al., 2007b**) and $x_0$ is the position of half-maximum *hunchback* expression ($x_0$ is of the order of 250 μm and varies slightly depending on cell cycle, usually close to 45% egg length for the WT *hunchback* promoter (**Porcher et al., 2010**). $L_0 \simeq 5.6 \mu m^{-3}$ is the concentration of free Bicoid molecules at the AP boundary (**Abu-Arish et al., 2010**) (that also corresponds to the point of largest *hunchback* expression slope (**Tran et al., 2018**).

To compute the diffusion limited arrival rate at the locus, we use the following parameters: diffusivity $D \simeq 7.4 \mu m^2 s^{-1}$ (**Porcher et al., 2010**; **Abu-Arish et al., 2010**), concentration of free Bicoid molecules at the AP boundary $L_0$, size of the binding target $a \simeq 3nm$ (**Gregor et al., 2007a**), which leads to an effective $\mu_{max} L_0 = DaL_0 \simeq 0.124 s^{-1}$ at the boundary. This value is an upper bound, assuming that every encounter between a transcription factor and a binding site results in successful binding. We note in Main Text **Figure 4b** that most of the ON rates are close to the diffusion limit. We conclude that in this parameter regime, the most efficient strategy is to have ON events that are as fast as possible. The only reason to reduce them is to achieve the required Hill coefficient. That can be done by either adjusting the ON or the OFF rates.

The above estimate $\mu_{max} L_0 = 0.124 s^{-1}$ may be inaccurate for various reasons and we ought to explore the sensitivity of results to those uncertainties. **Appendix 1—table 1** recapitulates the time-performance of different strategies for different choices of the parameters. A first source of uncertainty is the value of the diffusivity, which is estimated to vary between 4 and $7 \mu m^2 s^{-1}$ (**Porcher et al., 2010**; **Abu-Arish et al., 2010**). We consider then two possible values for the diffusivity from **Porcher et al., 2010**: $D_1 = 4.5 \mu m^2 s^{-1}$ and the aforementioned value $D_2 = 7.4 \mu m^2 s^{-1}$. A second source of uncertainty is that the actual Bicoid concentration at the boundary could vary by up to a factor two depending on estimates of the concentration and its decay length (**Gregor et al., 2007b**; **Tran et al., 2018**; **Abu-Arish et al., 2010**). We consider then two possible value for the concentration: the aforementioned $L_0^{(1)} = 5.6$ molecules per $\mu m^3$, and $L_0^{(2)} = 11.2$ molecules per $\mu m^3$. Finally, we assumed above that the size of the target is the full Bicoid operator site, which we took about ten base pairs following **Gregor et al., 2007a**. However, assuming that the TF must reach a specific position on the promoter-binding site could reduce the size of the target to a single base pair, that is $a$ by a factor 10. In terms of parameters, we consider then two possible values for $a$ : either $a_1 \simeq 3.10^{-4} \mu m$, or $a_2 = 3.10^{-3} \mu m$. All in all, taking into account the various sources of uncertainty $\mu_{max} L_0$ can range in the interval $[0.0076, 0.25] s^{-1}$.

We consider four possible decision strategies. The first one is a single binding site making a fixed-time decision. This computation is made using the original Berg-Purcell formula (**Berg and Purcell, 1977**). In the Berg-Purcell strategy, the concentration of the ligand is inferred based on the total time that the receptor or the binding site has spent occupied by ligands. Due to averaging, the relative error of the concentration readout is inversely proportional to the number of independent measurements of the concentration that can be made within the total fixed time $T$, that is, to the arrival rate of new Bicoid molecules at the binding site, multiplied by the probability to find the binding site empty (in our case, at the boundary, the probability is roughly one half). Since the rate of arrival of new Bicoid molecules to the binding site is $\mu_{max} L_0 = DaL_0$, the relative error of the concentration readout is given by

$$\frac{\delta L_0}{L_0} = \sqrt{\frac{1}{2(1-\bar{n})DaL_0T}},\tag{8}$$

where $L_0$ is the estimate of the concentration, $T$ is the integration time and $\bar{n}$ is the probability that the binding site is full.

In the second strategy, there are two sets of six binding sites being read independently. In that case, the information from each binding site can be accessed individually and their contributions averaged to give a more precise estimate. This calculation again can be made using the original Berg-Purcell formula (*Berg and Purcell, 1977*) for several receptors, dividing the relative error by the square root of the total number of binding sites in *Equation 8*:

$$\frac{\delta L_0}{L_0} = \sqrt{\frac{1}{2N(1-\bar{n})DaL_0T}},\tag{9}$$

where $N$ is the total number of binding sites (in our case, $N = 12$).

For the third possibility, we consider a decision made at a fixed time using the fastest architecture identified (Main Text *Figure 4*) without constraint on the slope (activation rule $k = 1$). We compute the decision time using the drift-diffusion approximation with fixed time (see Appendix 2).

Finally, we consider the fastest architecture identified with a random decision time and the Sequential Probability Ratio Test (SPRT) strategy.

The result of the above calculations is that for a single receptor estimating Bcd concentration with 10% precision within a fixed-time Berg-Purcell type calculation (see Appendix 2), decisions take between 6 min for the fastest binding rates and ~4 hr for the slowest estimates. Conversely, by using the on-the-fly SPRT decision-making process and the one-or-more $k = 1$ scheme at equilibrium, the time needed to make decisions with 10% precision and an error rate of 32% at the boundary is $\simeq$ 30s for the fastest rates and $\simeq$ 17 min for the slowest rates. For all sets of parameters, the on-the-fly SPRT decision-making process gives a ~3.5-fold faster decision time than the $N = 12$ Berg-Purcell estimate and a >10-fold faster decision making time than the one-binding-site Berg-Purcell estimate. For the fastest rates, a decision with an error rate of less than 5% can be achieved in about 5 min within the SPRT scheme.

**Appendix 1—table 1.** Mean decision times for various choices of parameters and four different decision processes (see sections 'Biological parameters in the embryo' and 'Error rate and decision time for the fixed-time decision strategy').

For the optimal architectures identified (third and fourth lines of the table) we take the fastest architectures without any constraints on the slope. These architectures systematically use the activation rule $k = 1$. Highlighted in red are the results for the range of parameters presented in the text. All calculations are made with $e = 32\%$, $L = L_1 = 1.05 \cdot L_0$, $L_2 = 0.95 \cdot L_0$. The diffusion limited ON rate is $\mu_{\max}L_0 = DaL_0$. For the two Berg-Purcell architectures, both the ON rate and OFF rate per site are equal to $\mu_{\max}$. For the optimal architectures, we have $\mu_iL_0 = \mu_{\max}L_0 \cdot (7-i)$ and $\nu_i = i \cdot \mu_{\max}L_0 \cdot 0.5^{1/6}/(1-0.5^{1/6})$ to keep half the genes active at the boundary.

| | $a_1$ | | | | $a_2$ | | | |
| | $D_1$ | | $D_2$ | | $D_1$ | | $D_2$ | |
| | $L_0^{(1)}$ | $L_0^{(2)}$ | $L_0^{(1)}$ | $L_0^{(2)}$ | $L_0^{(1)}$ | $L_0^{(2)}$ | $L_0^{(1)}$ | $L_0^{(2)}$ |
|---|---|---|---|---|---|---|---|---|
| Berg-Purcell one operator site | 4.0 hr | 100 min | 117 min | 59 min | 20 min | 10 min | 12 min | 5.9 min |
| Berg-Purcell twelve operator sites independently read | 58 min | 29 min | 34 min | 17 min | 5.8 min | 2.9 min | 3.4 min | 1.7 min |
| Optimal equilibrium architecture fixed-time decision ($e = 0.32$) | 24 min | 13 min | 15 min | 7.7 min | 2.8 min | 1.6 min | 1.8 min | 1.1 min |
| Optimal equilibrium architecture SPRT decision ($e = 0.32$) | 17 min | 8.5 min | 9.9 min | 5.0 min | 1.7 min | 51 s | 1.0 min | 30 s |

## Appendix 2

### Error rate and decision time for the fixed-time decision strategy

In this section, we describe how we compute the decision time for a fixed-time strategy (or 'Berg-Purcell type decision') and a complex promoter architecture.

The classic Berg-Purcell calculation is based on the idea of averaging the time spent by the ligand bound to a receptor (or, in our case, a binding site). The original calculation assumed that the waiting times between binding and unbinding are exponential and that the bound times are not informative about the concentration. Neither of these assumptions hold in the case of the *hunchback* promoter. Indeed, in the context of a complex promoter architecture, the waiting times that are available to the nucleus or cell downstream are the gene's ON and OFF switching times. They are not exponentially distributed, and, depending on the activation rule, the OFF times can be just as informative about the concentration as the ON times. For these two reasons, we ought to readapt the Berg-Purcell idea to compute the mean decision time.

To that purpose, we consider a decision with a given rate of error $e$, fix a time of decision $T$ and choose the concentration that has the highest likelihood between the two options $L_1$ and $L_2$. In other words, if $\log R(T) = \log P(L_1)/P(L_2) > 0$ then the nucleus chooses $L = L_1$, while if $\log R(T) < 0$ then it chooses $L = L_2$. For instance, if the actual concentration $L = L_1$, then the probability of error at time $T$ is given by $P(\log R(t) < 0)$.

To calculate the above error, we use the drift-diffusion approximation for $\log R(t)$, compute the drift $V$ and diffusivity $D$ from Main Text *Equations 1-4* and approximate the distribution of $\log R(T)$ by a normal distribution with mean $VT$ and standard deviation $\sqrt{DT}$. We compute the error rate $e$ for the fixed-time decision process based on the Gaussian approximation. Finally, to find the mean decision time for a given error rate, we perform a quick one-dimensional search over $T$, which yields the value of the fixed decision time appropriate for the prescribed error $e$.

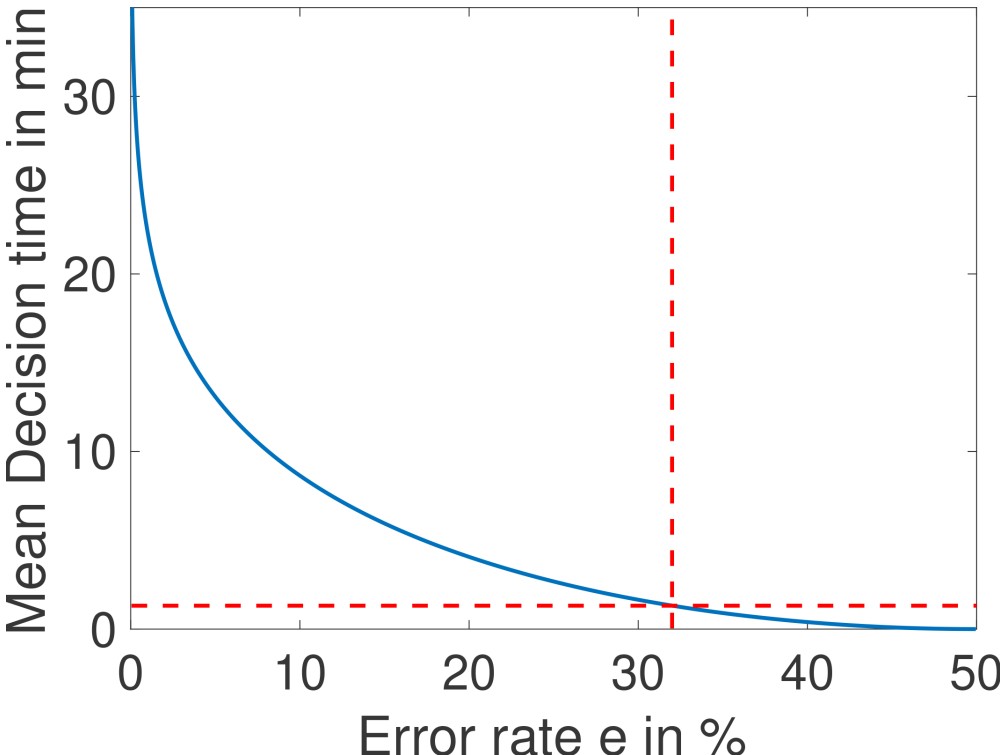

**Appendix 2—figure 1.** The mean decision time as a function of the error rate $e$ for the fastest architecture identified with $H>5$ and $k=1$ (see Main Text *Figure 4a*). When the error rate is 32%, decisions are made in less than a minute (red dashed lined). Parameters are $L = L_1 = 1.05 \cdot 5.6 \mu m^{-3}$, $L_2 = 0.95 \cdot 5.6 \mu m^{-3}$, $\mu_1 = 0.13 \mu m^3 s^{-1}$, $\mu_2 = 0.11 \mu m^3 s^{-1}$, $\mu_3 = 0.086 \mu m^3 s^{-1}$, $\mu_4 = 0.066 \mu m^3 s^{-1}$,

$\mu_5 = 0.043 \mu m^3 s^{-1}$, $\mu_6 = 0.022 \mu m^3 s^{-1}$, $\nu_1 = 4.5 s^{-1}$, $\nu_2 = 0.042 s^{-1}$, $\nu_3 = 0.11 s^{-1}$, $\nu_4 = 0.033 s^{-1}$, $\nu_1 = 0.037 s^{-1}$, $\nu_1 = 0.053 s^{-1}$.

## Appendix 3

### The mean first-passage time for the decision-making process

To investigate the role of promoter architectures in decision-making, we apply the SPRT approach (Sequential Probability Ratio Test) (*Wald, 1945a*; *Siggia and Vergassola, 2013*). In the simplest formulation of this approach, a decision is made between two hypotheses about the concentration of a surrounding TF: either the TF is at concentration $L_1$ or it is at concentration $L_2$. The decision is based on the binding and unbinding events of the TF to a promoter. At each point in time, the error of committing to a given concentration (scenario) is estimated by computing the ratio $R(t)$ of the probability of one hypothesis, $P(L_1)$ (the surrounding concentration is $L_1$) over the other (the concentration is $L_2$), $P(L_2)$:

$$R(t) = \frac{P(L_1)}{P(L_2)}.$$ 
(10)

Technically, the time dependent likelihood ratio, $R(t)$ undergoes a random walk as TFs bind and unbind to the promoter, activating and deactivating the gene. The process is terminated and a decision is made when the likelihood ratio falls below the desired error level, which is expressed in terms of absorbing boundaries at $K_+$ (the concentration is $L_1$) and $K_-$ (the concentration is $L_2$): $\log R(t) \leq K_-$ or $\log R(t) \geq K_+$ (see Main Text *Figure 2*). The boundaries $K_\pm$ can be expressed in terms of the error $e$ and for symmetric errors, $K_+ = -K_- = (1-e)/e$ (*Siggia and Vergassola, 2013*). In our case $e = 32\%$. The mean time for decision can be computed for each discrimination task as the mean first-passage time of a random walk (in the limit of long decision times). In *Appendix 2—figure 1*, we show the mean decision-time for different values of $e$.

Assuming the gene has two levels of activation ON and OFF, the information available to downstream mechanisms is a series of ON times $s_i$ and OFF times $t_j$ of gene activity. The probability of a concentration hypothesis is then $P(L_m) = P(\{t_i\}, \{s_j\}|L_m)$. If successive ON and OFF times are independent then the probabilities factorize. The log ratio is then written as a function of the ON (respectively OFF) time probability distribution $P_{\mathrm{ON}}(t, L)$ (respectively $P_{\mathrm{OFF}}(t, L)$)

$$\log R(t) = \sum_{i=1}^{J_-} (\log P_{\mathrm{ON}}(s_i, L_1) - \log P_{\mathrm{ON}}(s_i, L_2)) + \sum_{i=1}^{J_+} (\log P_{\mathrm{OFF}}(t_i, L_1) - \log P_{\mathrm{OFF}}(t_i, L_2)),$$ 
(11)

where $J_-$ is the number of ON to OFF switching events and $J_+$ the number of OFF to ON switching events (*Siggia and Vergassola, 2013*). To understand the dynamics of the log ratio, it is necessary to compute the distributions $P_{\mathrm{ON}}(t, L)$ and $P_{\mathrm{OFF}}(t, L)$ based on the promoter dynamics, which is the focus of the following subsection.

### From binding to gene activation

In this subsection, we describe how the high-dimensional complete state of the promoter is mapped onto the low-dimensional activity of the gene. We use a formalism similar to that of *Desponds et al., 2016*; *Tran et al., 2018*.

The promoter features $N$ binding sites : its complete state at time $T$ is described by an $N$ dimensional vector $\vec{B}(t)$, where $B_i = 0/1$ if the binding site $i$ is empty/full. We assume the gene has two levels of activity: either it is OFF and no mRNA is produced, or it is ON and mRNA is produced at a fixed rate. Activity is then described by a simple Boolean variable $a(t)$ equal to 0/1, corresponding to the gene being OFF/ON.

We also assume that the activity of the gene depends only on the total number of transcription factors bound to the promoter so that there is an integer $1 \leq k \leq N$ such that $a(t) = \mathbb{1}_{\sum_i B_i \geq k}$ (where $\mathbb{1}$ is the indicator function). From the point of view of gene activity, we are only interested in the dynamics of $\mathbf{B}(t) = \sum_i B_i(t)$. We make another simplifying assumption: the probabilities of $\mathbf{B}(t)$ increasing or decreasing only depend on the value of $\mathbf{B}(t)$ and not on which binding sites specifically are bound or unbound, that is $\mathbf{B}(t)$ itself has a Markovian dynamics in $\{0, 1, ...N\}$.

At a given time $t$, we describe the stochastic state of the promoter as a vector of probability $\vec{X}(t)$ where $X_i(t)$ is the probability of having $\mathbf{B} = i - 1$ and $\sum_i X_i(t) = 1$. The Markovian dynamics of B

translate into an $(N+1) \times (N+1)$ transition matrix $M$ for $\vec{X} : \vec{X}(t+s) = e^{Ms}\vec{X}(t)$. $M$ describes the dynamics of the promoter from the point of view of gene activity. If transcription factors do not dimerize or form complexes, then $M_{i,j} \neq 0$ only if $|i-j| \leq 1$ since the probability of two of them binding or unbinding decreases as the square of time for short times.

Let us now relate the statistics of the switching times for the gene activity $a$ to the transition matrix $M$. Starting from a distribution of OFF states $\vec{X_0}$ we compute the cumulative distribution function of the waiting time $\tau$ before switching to the $ON$ state

$$P(\tau \leq t) = \text{Trans}(\vec{X}_{\text{ON}}) e^{M^*_{\text{OFF}}t} \vec{X_0}, \tag{12}$$

where $Trans$ is the transpose, $\vec{X}_{\text{ON}}$ is a vector of 1 for states corresponding to active genes and 0 for states corresponding to inactive genes, and $M^*_{\text{OFF}}$ is a modified version of $M$ where all ON states act as 'sinks' (i.e. all the transition rates from ON states have been set to 0). The expression in *Equation 12* computes the cumulative probability of transitions to an ON state before time $t$, that is $P_{\text{OFF}}(t) = dP(\tau \leq t)/dt$.

For simplicity, we restrict ourselves to promoter structures where there is only one point of entry into the ON or the OFF states (i.e. any switching event from ON to OFF or *vice versa* will end with the promoter having a specific number of binding sites). Under this hypothesis, the probability vector $\vec{X_0}$ in *Equation 12* is uniquely determined and independent of the specific dynamics of the previous ON or OFF times. We relax these constraints on the promoter structure in Appendix 8. We denote by $\tau_{\text{OFF}}$ (respectively $\tau_{\text{ON}}$) the average of the OFF (respectively ON) times for the real concentration $L$.

## Random walk of the log ratio

When the two hypothesized concentrations are very similar to each other as compared to the concentration scale $L$ set by the actual concentration value, $|L_1 - L_2|<<L$, the discrimination is hard and requires a large number of binding and unbinding events. In this limit, the random walk $\log R(t)$ can be approximated by a drift-diffusion process with drift $V$ and diffusion $D$:

$$\partial_t \log R(t) = V + \sqrt{2D}\eta, \tag{13}$$

where $\eta$ is a Gaussian white noise. The decision time for the case of symmetric boundaries $K_+ = -K_-$ is a random variable $T$ with mean given by *Equation 1* in the main text (*Siggia and Vergassola, 2013*). $\langle T \rangle$ depends on the details of the biochemical sensing of the TF concentration and promoter architecture via the drift and diffusivity. Computing $V$ and $D$ in *Equation 13* is enough to derive the mean first-passage time to the decision and even its full distribution by computing its Laplace transform as a solution of the drift-diffusion equation using standard techniques of first-passage for Gaussian processes (*Siggia and Vergassola, 2013*).

## Drift

For many switching events $J^+>>1$ and $J^->>1$, the sums in *Equation 11* can be replaced by continuous averages over binding time distributions. Since $J_-$ and $J_+$ differ at most by one, we can also replace the two values $J_- \simeq J_+$ by a unique value $J$, which is the number of ON-OFF cycles. The times are distributed according to the ON and OFF probability distributions for the real concentration $L$. At a given large time $t$, the number $J$ of terms in the sum and the log-ratios appearing in *Equation 11* are a priori correlated but Wald's equality (*Wald, 1945a*) ensures that the average of the sum is the product of the two averages. Since the expected number of cycles grows linearly in time as $\langle J \rangle \propto t/(\tau_{\text{ON}} + \tau_{\text{OFF}})$, we conclude that the drift term in *Equation 13* is given by:

$$
\begin{aligned}
V &= \frac{\langle J \rangle}{t} \times \langle \log P_{\mathrm{ON}}(t,L_1) - \log P_{\mathrm{ON}}(t,L_2) + \log P_{\mathrm{OFF}}(t,L_1) - \log P_{\mathrm{OFF}}(t,L_2) \rangle_L \\
&= \frac{1}{\tau_{\mathrm{ON}} + \tau_{\mathrm{OFF}}} \left[ \int_0^{+\infty} dt P_{\mathrm{ON}}(t,L) \left( \log \frac{P_{\mathrm{ON}}(t,L_1)}{P_{\mathrm{ON}}(t,L)} - \log \frac{P_{\mathrm{ON}}(t,L_2)}{P_{\mathrm{ON}}(t,L)} \right) \right. \\
&\quad + \left. \int_0^{+\infty} dt P_{\mathrm{OFF}}(t,L) \left( \log \frac{P_{\mathrm{OFF}}(t,L_1)}{P_{\mathrm{OFF}}(t,L)} - \log \frac{P_{\mathrm{OFF}}(t,L_2)}{P_{\mathrm{OFF}}(t,L)} \right) \right] \\
&= \frac{1}{\tau_{\mathrm{OFF}} + \tau_{\mathrm{ON}}} [D_{KL}(P_{\mathrm{OFF}}(.,L)||P_{\mathrm{OFF}}(.,L_2)) - D_{KL}(P_{\mathrm{OFF}}(.,L)||P_{\mathrm{OFF}}(.,L_1)) \\
&\quad + D_{KL}(P_{\mathrm{ON}}(.,L)||P_{\mathrm{ON}}(.,L_2)) - D_{KL}(P_{\mathrm{ON}}(.,L)||P_{\mathrm{ON}}(.,L_1))],
\end{aligned}
\tag{14}
$$

where $D_{KL}(P||Q) = \int_0^\infty ds P(s) \log(P(s)/Q(s))$ is the Kullback-Leibler divergence between the two distributions $P$ and $Q$.

In sum, the drift is determined by how informative the distribution of waiting times in the ON and OFF states are about the concentration differences, with an average rate equal to the inverse of the cycle time $\tau_{\mathrm{OFF}} + \tau_{\mathrm{ON}}$. The Kullback-Leibler divergence appearing in *Equation 14* is intuitive : it represents the distance between the real concentration and the hypothetical concentration in the space of probabilistic models of switching times. The larger that distance, the easier it becomes to tell the difference between the two distributions through random sampling.

## Expansion of the drift for small concentration differences

When the two candidate concentrations are similar, $L \simeq L_1 \simeq L_2$, the quantities computed in the previous subsection can be expanded at first order in the differences in concentrations $\delta L_1 = L_1 - L$ and $\delta L_2 = L_2 - L$. Starting from *Equation 14*, we expand the drift $V$ at first and second orders:

$$
\begin{aligned}
V(\tau_{\mathrm{ON}} + \tau_{\mathrm{OFF}}) &= \int_0^{+\infty} dt P_{\mathrm{OFF}}(t,L) \left[ \log \frac{P_{\mathrm{OFF}}(t,L_1)}{P_{\mathrm{OFF}}(t,L)} - \log \frac{P_{\mathrm{OFF}}(t,L_2)}{P_{\mathrm{OFF}}(t,L)} \right] + \hookrightarrow_{P_{\mathrm{ON}}} \\
&= \int_0^{+\infty} dt P_{\mathrm{OFF}}(t,L) \left[ \log \frac{P_{\mathrm{OFF}}(t,L) + \delta L_1 \partial_L P_{\mathrm{OFF}}(t,L) + \delta L_1^2 \partial_{L,L}^2 P_{\mathrm{OFF}}(t,L)/2}{P_{\mathrm{OFF}}(t,L)} \right] \\
&\quad - \int_0^{+\infty} dt P_{\mathrm{OFF}}(t,L) \left[ \log \frac{P_{\mathrm{OFF}}(t,L) + \delta L_2 \partial_L P_{\mathrm{OFF}}(t,L) + \delta L_2^2 \partial_{L,L}^2 P_{\mathrm{OFF}}(t,L)/2}{P_{\mathrm{OFF}}(t,L)} \right] + \hookrightarrow_{P_{\mathrm{ON}}} \\
&= \delta L_1 \int_0^{+\infty} dt \partial_L P_{\mathrm{OFF}}(t,L) + \frac{\delta L_1^2}{2} \int_0^{+\infty} dt \partial_{L,L}^2 P_{\mathrm{OFF}}(t,L) - \int_0^{+\infty} dt \frac{\delta L_1^2}{2} \frac{(\partial_L P_{\mathrm{OFF}}(t,L))^2}{P_{\mathrm{OFF}}(t,L)} \\
&\quad - \delta L_2 \int_0^{+\infty} dt \partial_L P_{\mathrm{OFF}}(t,L) - \frac{\delta L_2^2}{2} \int_0^{+\infty} dt \partial_{L,L}^2 P_{\mathrm{OFF}}(t,L) + \int_0^{+\infty} dt \frac{\delta L_2^2}{2} \frac{(\partial_L P_{\mathrm{OFF}}(t,L))^2}{P_{\mathrm{OFF}}(t,L)} \hookrightarrow_{P_{\mathrm{ON}}},
\end{aligned}
\tag{15}
$$

where $\hookrightarrow_{P_{\mathrm{ON}}}$ means that the same operations and integrations are performed for $P_{\mathrm{ON}}$.

Due to conservation of probability, the integral of the first and second $L$-derivatives of $P_{\mathrm{OFF}}(t,L)$ vanish. The first-order expansion of $V$ vanishes and we have at first non-vanishing order in $|L_i - L|$

$$
V = \frac{1}{2(\tau_{\mathrm{ON}} + \tau_{\mathrm{OFF}})} \left[ \int_0^{+\infty} dt \frac{(\partial_L P_{\mathrm{OFF}}(t,L))^2}{P_{\mathrm{OFF}}(t,L)} \right] [\delta L_2^2 - \delta L_1^2] + \hookrightarrow_{P_{\mathrm{ON}}}.
\tag{16}
$$

If for instance $|L_2 - L| > |L_1 - L|$ then the drift is positive, favouring the concentration $L_1$, closer to the real concentration.

## Equality between drift and diffusivity

In this subsection, we present different ways of proving the equality between $V$ and $D$ in SPRT when the two hypotheses are close by connecting different approaches. We show that this equality is a general property of random walks in Bayesian belief space. We check that these results are true in the controlled case of one binding site in *Appendix 3—figure 1*.

### First approach: the exit points of the decision process

As proved by Wald in the original paper where he introduced sequential probability ratio tests (*Wald, 1944*), the nature of the test (the ratio between the likelihood of two hypotheses) requires a specific relationship between the error and the boundaries that define the regions of decision. In our case, we assume the same probability of calling $L_1$ when $L_2$ is true and calling $L_2$ when $L_1$ is true,

leading to the definition of only one error level $e$ and two symmetric boundaries $K$ and $-K$. Specifically, Wald shows that $K = \log\frac{1-e}{e}$ (see also *Siggia and Vergassola, 2013*).

When the two hypotheses are close enough that the variations of the log-likelihood can be approximated by a Gaussian process, one can compute the exit probabilities in terms of the drift $V$ and diffusivity $D$. The equation for the probability of absorption $\Pi(X)$ at the upper boundary $K$ is

$$\left[V\frac{d}{dx} + D\frac{d^2}{dx^2}\right]\Pi(x) = 0, \tag{17}$$

with the boundary condition $\Pi(K) = 1$ and $\Pi(-K) = 0$. The corresponding solution is

$$\Pi(x) = \frac{e^{\frac{VK}{D}} - e^{\frac{-Vx}{D}}}{e^{\frac{VK}{D}} - e^{\frac{-VK}{D}}}, \tag{18}$$

as shown in the supplementary material of *Siggia and Vergassola, 2013*. Setting $x = 0$ in *Equation 18*, we find that

$$\Pi(0) = \frac{e^{VK/D} - 1}{e^{VK/D} - e^{-VK/D}} = \frac{e^{VK/D}}{1 + e^{VK/D}}. \tag{19}$$

We note that this probability of absorption is also $1 - e$ by definition of the error (assuming for instance that $L = L_1$) leading to

$$e = \frac{1}{1 + e^{VK/D}}, \tag{20}$$

which in turn gives

$$e^{VK/D} = \frac{1-e}{e} = e^K. \tag{21}$$

And so we find that in the limit of close hypotheses, $V = D$.

## Second approach: expansion for small concentration differences

Let us consider the SPRT process between two hypotheses $L_1$ and $L_2$ and assume for simplicity that $L = L_2$. In this version of the proof, we consider that the difference $\delta L = L_1 - L$ is small compared to $L$ (i.e $\delta L/L << 1$) and expand the expressions for drift and diffusion in increasing orders of $\delta L/L$ to find that they match. We have shown in the subsection 'Expansion of the drift for small concentration differences' of Appendix 3 that the first non-vanishing term in the expansion of the drift is of order 2 in $\delta L/L$ and is given by *Equation 16*. An integral formula for the second moment of the *log*-likelihood is given in Equation A15 of *Carballo-Pacheco et al., 2019* from which we get that the diffusivity of the *log* ratio is the sum of four terms. The first term is given by

$$D_1 = \frac{V^2}{(\tau_{\text{ON}} + \tau_{\text{OFF}})^3}\int_0^{+\infty} dt P_{\text{ON}}(t,L)\int_0^{\infty} ds P_{\text{OFF}}(s,L)(t+s)^2. \tag{22}$$

$D_1$ is proportional to $V^2$ and so is of order $(\delta L/L)^4$. The second term is given by

$$\begin{aligned}D_2 &= \frac{-2V}{(\tau_{\text{ON}} + \tau_{\text{OFF}})^2}\left(\tau_{\text{ON}}\int_0^{\infty} ds P_{\text{OFF}}(s,L)\log\frac{P_{\text{OFF}}(s,L_1)}{P_{\text{OFF}}(s,L_2)} + \tau_{\text{OFF}}\int_0^{\infty} dt P_{\text{ON}}(t,L)\log\frac{P_{\text{ON}}(t,L_1)}{P_{\text{ON}}(t,L_2)}\right. \\ &\left. + \int_0^{\infty} ds P_{\text{OFF}}(s,L)s\log\frac{P_{\text{OFF}}(s,L_1)}{P_{\text{OFF}}(s,L_2)} + \int_0^{\infty} dt P_{\text{ON}}(t,L)t\log\frac{P_{\text{ON}}(t,L_1)}{P_{\text{ON}}(t,L_2)}\right).\end{aligned} \tag{23}$$

The prefactor $V$ is of order $(\delta L/L)^2$. Expanding the first two terms in the brackets gives the same type of terms as in *Equation 15*. Because of probability conservation we find that these terms are of the same order as $V$ (i.e $(\delta L/L)^2$). The last two terms have prefactors $s$ and $t$ respectively which break the argument for vanishing first order terms in *Equation 15* and these terms are of order $\delta L/L$. Putting the pieces together, we find that $D_2$ is of order $(\delta L/L)^3$. The third term is given by

$$D_3 = (\tau_{\text{ON}} + \tau_{\text{OFF}})^{-1} \int_0^{+\infty} dt P_{\text{ON}}(t,L) \int_0^{+\infty} ds P_{\text{OFF}}(s,L) \left( \log \frac{P_{\text{ON}}(t,L_1)}{P_{\text{ON}}(t,L_2)} + \log \frac{P_{\text{OFF}}(s,L_1)}{P_{\text{OFF}}(s,L_2)} \right)^2. \tag{24}$$

Because ON and OFF times are independent, cross terms in $D_3$ are products of two $\langle \log P(.,L_1)/P(.,L_2) \rangle$ and are of subleading order $(\delta L/L)^4$. Using the symmetry between $P_{\text{ON}}$ and $P_{\text{OFF}}$, we expand one of the square terms in $D_3$

$$\begin{aligned}
\int_0^{+\infty} dt P_{\text{ON}}(t,L) \left( \log \frac{P_{\text{ON}}(t,L+\delta L)}{P_{\text{ON}}(t,L)} \right)^2 &= \int_0^{+\infty} dt P_{\text{ON}}(t,L) \left( \delta L \frac{\partial_L P_{\text{ON}}(t,L)}{P_{\text{ON}}(t,L)} + O\left(\frac{\delta L^2}{L^2}\right) \right)^2 \\
&= \int_0^{+\infty} dt \frac{(\partial_L P_{\text{ON}}(t,L))^2}{P_{\text{ON}}(t,L)} + O(\frac{\delta L^3}{L^3}).
\end{aligned} \tag{25}$$

The fourth term is $-V^2$ which is of order $(\delta L/L)^4$. So we find that, at leading order, $D \simeq D_3$ which has the same expansion as $V$ (see *Equation 16*). And so we recover that for small concentration differences, $V = D$.

In the case of one binding site with ON rate $\mu L$ and OFF rate $\nu$ we check that the formula from *Carballo-Pacheco et al., 2019* gives the exact expression computed in *Siggia and Vergassola, 2013*

$$D = \frac{\nu \mu L}{(\nu + \mu L)^3} \left( \mu^2 (L_2 - L_1)^2 + \frac{1}{2} \left( \log \frac{L_1}{L_2} \right)^2 (\nu^2 + \mu^2 L^2) + \mu (L_2 - L_1)(\nu - \mu L) \log \frac{L_1}{L_2} \right). \tag{26}$$

Replacing $L_2$ with $L$ in *Equation 26* and expanding in $\delta L/L$ we have

$$\begin{aligned}
D &= \frac{\nu \mu L}{(\nu + \mu L)^3} \left( \mu^2 \delta L^2 + \frac{1}{2} \left( \frac{\delta L}{L} \right)^2 (\nu^2 + \mu^2 L^2) + \mu \delta L (\nu - \mu L) \frac{\delta L}{L} + o\left(\frac{\delta L^2}{L^2}\right) \right) \\
&= \frac{\nu \mu L}{2(\nu + \mu L)^3} \frac{\delta L^2}{L^2} \left( (\nu^2 + \mu^2 L^2) + \mu L \nu + o\left(\frac{\delta L^2}{L^2}\right) \right) = \frac{1}{\nu^{-1} + (\mu L)^{-1}} \frac{\delta L^2}{2L^2} + o\left(\frac{\delta L^2}{L^2}\right).
\end{aligned} \tag{27}$$

We identify the drift computed for the one binding site example in the paragraph 'Mean decision time : connecting drift-diffusion and Wald's approaches' of the main text and find that at first order drift and diffusivity are equal.

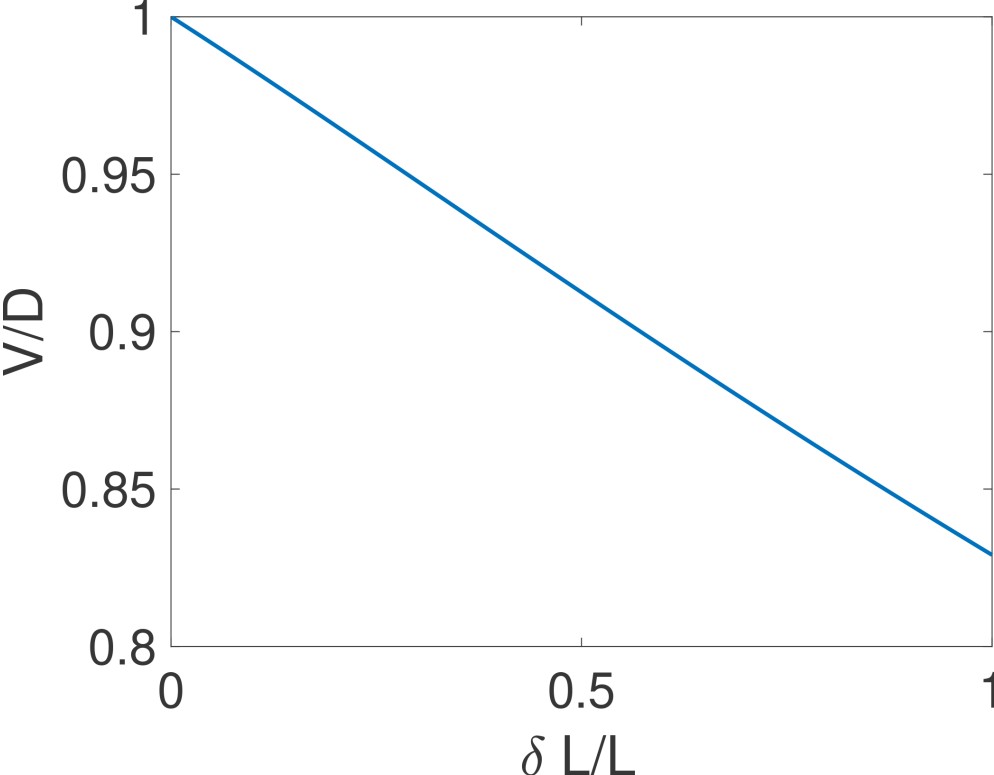

**Appendix 3—figure 1.** The equality $V = D$ holds for close hypotheses. We compare the exact drift and diffusion computed from the formulae derived in *Siggia and Vergassola, 2013* for the case of one binding site. We find that drift and diffusion are approximately equal for close concentrations (i. e small $\delta L/L$). Parameters are $\nu = 1\ s^{-1}$, $\mu = 1\ \mu m^3 s^{-1}$, $L_2 = L = 1\mu m^{-3}$, $L_1 = L + \delta L$.

### Third approach: general properties of Bayesian random walks

To understand the origin of the equality between drift and diffusion, we go back to the definition of SPRT as the update of the Bayesian beliefs in two competing hypotheses. We no longer condition the process on one hypothesis being true but rather on the probability that each one is true given the value of the *log* ratio a certain time point. A very general property of posterior distributions updated with evidence is that their average does not change. This martingale property of belief update can be shown in the following way: assume that the nucleus receives extra evidence $\theta$ with probability $p(\theta)$. Before that evidence comes, the probability that hypothesis 1 is true $p_1$ has a certain distribution $P(p_1)$. Then we have that the probability that hypothesis 1 is true varies as :

$$\langle p_1|\theta\rangle = \int d\theta \int dp_1 P(p_1|\theta)p_1 p(\theta) = \int d\theta \int dp_1 p_1 P(p_1)p(\theta|p_1), \tag{28}$$

where we used Bayes rule. The integral of $p(\theta|p_1)$ over $\theta$ sums up to one and we are left with

$$\langle p_1|\theta\rangle = \int dp_1 p_1 P(p_1) = \langle p_1\rangle. \tag{29}$$

In the context of two hypothetical concentrations, we consider the normalized likelihood of the two hypotheses at time $t$: $Q_2(t) = P_2(t)/(P_1(t) + P_2(t))$ and $Q_1(t) = P_1(t)/(P_1(t) + P_2(t))$, where $P_i(t)$ is the likelihood that hypothesis $i$ is true based on the evidence accumulated up to time $t$. The martingale property translates as

$$0 = \langle \Delta Q_2\rangle = Q_1\langle \Delta Q_2\rangle_1 + Q_2\langle \Delta Q_2\rangle_2, \tag{30}$$

where $\langle\rangle_\alpha$ are averages taken assuming hypothesis $\alpha$ is true and $\Delta Q$ is the variation of $Q$ through time (or accumulated evidence). We note that $Q_1 = e^{\mathcal{L}(t)}/(1 + e^{\mathcal{L}(t)})$ and $Q_2 = 1/(1 + e^{\mathcal{L}(t)})$, where

$\mathcal{L}(t) = \log P_1(t)/P_2(t)$ is the *log*-likelihood ratio at a given time. We express the variations of the probabilities in terms of the *log*-likelihood so that we can connect it to the drift-diffusion parameters. The drift and diffusivity that depend a priori on the real concentration are the average and variance of the variation of $\mathcal{L}$, respectively. We have $\langle \Delta \mathcal{L}(t) \rangle_1 = V_{(1)} \Delta t$, $\langle \Delta \mathcal{L}(t) \rangle_2 = V_{(2)} \Delta t$, $\langle (\Delta \mathcal{L}(t) - \langle \Delta \mathcal{L} \rangle_1)^2 \rangle_1 = 2D_{(1)} \Delta t$ and $\langle (\Delta \mathcal{L}(t) - \langle \Delta \mathcal{L} \rangle_2)^2 \rangle_2 = 2D_{(2)} \Delta t$ (where $\Delta$ is the total change over time $\Delta t$). When the two concentrations are close, the statistics of the acquisition of evidence are symmetric for $L_1$ and $L_2$ and we get that $D_{(1)} = D_{(2)} = D$ and $V_{(1)} = -V_{(2)} = V$. As explained in the paragraph 'Mean decision time : connecting drift-diffusion and Wald's approaches' of the main text, computing the derivatives of $Q_i$ with respect to $\mathcal{L}$ is straightforward. Plugging these relationships into *Equation 30* and gathering terms in $V$ and $D$ we find that

$$
\begin{aligned}
0 &= \partial_{\mathcal{L}} \langle \Delta Q_2 \rangle [Q_1 \langle \Delta \mathcal{L}(t) \rangle_1 + Q_2 \langle \Delta \mathcal{L}(t) \rangle_2] + \frac{1}{2} \partial_{\mathcal{L}}^2 \langle \Delta Q_2 \rangle [Q_1 \langle \Delta^2 \mathcal{L}(t) \rangle_1 + Q_2 \langle \Delta^2 \mathcal{L}(t) \rangle_2] \\
&= -Q_1 Q_2 [Q_1 V \Delta t - Q_2 V \Delta t] + \frac{1}{2} Q_1 Q_2 (Q_1 - Q_2)[2 Q_1 D \Delta t + 2 Q_2 D \Delta t] \\
&= \Delta t V Q_1 Q_2 (Q_2 - Q_1) + \Delta t D Q_1 Q_2 (Q_1 - Q_2)(Q_1 + Q_2) \\
&= \Delta t Q_1 Q_2 (Q_1 - Q_2)(V - D),
\end{aligned}
\tag{31}
$$

from which we derive that $V = D$.

## The first passage time to decision

The first exit time of a drift-diffusion process starting from 0 with two symmetric boundaries at $+K$ and $-K$ is given by $\langle T \rangle = K \tanh(VK/2D)/V$ (see for instance [*Redner, 2001*]). Plugging in $V = D$, we recover *Equation 4* of the main text:

$$
\langle T \rangle = \frac{K \tanh(K/2)}{V}.
\tag{32}
$$

We check that this approximation is correct for small concentration differences in *Appendix 3— figure 2a,b and c*. As mentioned in the main text, because the hyperbolic tangent is very flat for high values of its argument, we find that the mean decision time depends weakly on corrections to $V = D$ when the error rate $e$ is small (which is equivalent to $K$ large). We check that this result in true in *Appendix 3—figure 2d*. We note that from $K = \log(1-e)/e$ we have $\tanh(K/2) = 1 - 2e$. And so we recover the second part of *Equation 4* from the main text

$$
\langle T \rangle = \frac{K}{V}(1 - 2e).
\tag{33}
$$

In one of the original papers about SPRT (*Wald, 1945a*), Wald derives a similar formula for the total number of events before the decision:

$$
\langle J_{\mathrm{exit}} \rangle = \frac{K}{V(\tau_{\mathrm{ON}} + \tau_{\mathrm{OFF}})}(1 - 2e),
\tag{34}
$$

where $J_{exit}$ is the number of ON-OFF events when decision happens. Combining *Equation 34* with Wald's equality (*Wald, 1945b*; *Durrett, 2010*) that states that $\langle T \rangle = \langle J_{\mathrm{exit}} \rangle (\tau_{\mathrm{ON}} + \tau_{\mathrm{OFF}})$, we recover *Equation 33*.

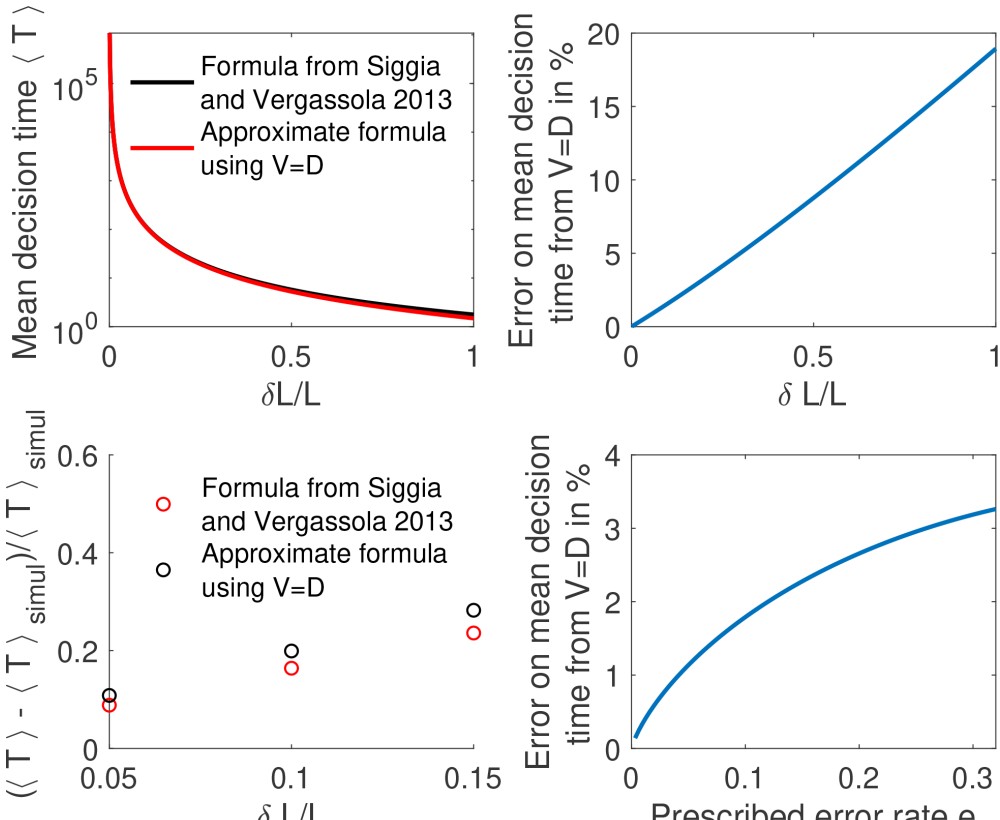

**Appendix 3—figure 2.** Comparing methods to compute the mean decision time for the one binding site case. (a) We compute the mean decision time for one binding site using the method from *Siggia and Vergassola, 2013* $\langle T \rangle_{\text{SV}}$ (i.e $\langle T \rangle_{\text{SV}} = K \tanh(VK/2D)/V$, *Equation 2* from main text and *Equation 26*) in black and mean decision time $\langle T \rangle$ from the approximate method using $V = D$, *Equation 2* and *Equation 4* from main text in red. In panel (b) we show for the same results the error made from using $V = D$ : $100 * |\langle T \rangle_{\text{SV}} - \langle T \rangle| / \langle T \rangle_{\text{SV}}$. We find that the difference is small for small concentration differences $\delta L/L$. (c) We show the relative error for the methods to compute the mean decision time compared to the mean time to decision $\langle T \rangle_{\text{simul}}$ in a Monte-Carlo simulation. We find again that the error is small for small concentration differences. Parameters are $e = 0.32$, $L = L_2 = 1\ \mu m^{-3}$, $L_1 = L + \delta L$, $\mu = 1\ \mu m^3 s^{-1}$, $\nu = 1\ s^{-1}$. (d) We show the relative error from $V = D$ (i.e $100 * |\langle T \rangle_{\text{SV}} - \langle T \rangle| / \langle T \rangle_{\text{SV}}$) for fixed $\delta L/L = 0.2$ as a function of the error rate. We find that the correction to $V = D$ is weak for small errors. Other parameters are the same as **a,b,c**.

## When are correlations between the times of events leading to decision important?

To compute the drift or diffusivity, one must compute the variation of the log-likelihood up to a given time $T$. The fact that the total time $T$ is fixed introduces correlation between the duration of the different ON-OFF events. Can these correlations be ignored, leading to an effective drift or diffusion per cycle? They can only when the two concentrations are close, as we check in *Appendix 3—figure 3*.

Following the arguments in *Carballo-Pacheco et al., 2019*, let us consider the following generating function for the cumulants of the log-likelihood difference $e^{\lambda(\mathcal{L}_2 - \mathcal{L}_1)}$ at a given time $T$ assuming hypothesis $i$ is true ($i = 1$ or $2$) :

$$M(\lambda) = \sum_{n=1}^{\infty} \int \left[ e^{\lambda(\mathcal{L}_2 - \mathcal{L}_1)} \left( \prod_{j=1}^{n} P_{\text{ON}}(s_j, L_i) P_{\text{OFF}}(t_j, L_i)\, dt_j\, ds_j \right) \Theta\left( T - \sum_{j=1}^{n} (t_j + s_j) \right) \right], \tag{35}$$

where $\mathcal{L}_\alpha$ is the *log*-likelihood of hypothesis $\alpha$, $n$ is the number of ON-OFF cycles and $s_j$ and $t_j$ are respectively the ON and OFF waiting times. This is not exactly what appears in *Carballo-*

*Pacheco et al., 2019*, but it is sufficient to grasp the consequences of the correlations introduced by a global constraint on the duration of the process, that is, the $\Theta$ function. Note that such constraint breaks the statistical independence of the various cycles, and introduces dependencies even though the $P_{\text{ON}}$ and $P_{\text{OFF}}$ factorize. We rewrite the Heaviside function using the Laplace transform form $\Theta(x) = \int_{\gamma-i\infty}^{\gamma+i\infty} \frac{d\sigma}{2i\pi\sigma} e^{\sigma x}$ and the product of the probabilities as $e^{\sum p_{\text{ON}}(s_j, L_\alpha) + p_{\text{OFF}}(t_j, L_\alpha)}$, where $p_{\text{ON}}(., L_\alpha)$ and $p_{\text{OFF}}(., L_\alpha)$ are the log-likelihoods for hypothesis $\alpha$. Putting the pieces together we obtain

$$M(\lambda) = \sum_{n=1}^{\infty} \int_{\gamma-i\infty}^{\gamma+i\infty} \frac{d\sigma}{2i\pi\sigma} e^{\sigma T} f^n(\sigma, \lambda), \tag{36}$$

where

$$f(\sigma, \lambda) \equiv \int \int e^{\lambda[(p_{\text{ON}}(s,L_2) - p_{\text{ON}}(s,L_1)) + (p_{\text{OFF}}(t,L_2) - q_{\text{OFF}}(t,L_2))]} \times e^{p_{\text{ON}}(s,L_i) + q_{\text{OFF}}(t,L_i) - \sigma(t+s)} \, dt \, ds. \tag{37}$$

One can remark that the expression is factorized, that is, $f$ is raised to the power $n$, and even $f$ itself can be interpreted as 'expectation value' of $e^{\lambda[p_{\text{ON}}(s,L_2) - p_{\text{ON}}(s,L_1)]}$ (and same with $p_{\text{OFF}}(t, L_\alpha)$) over the 'distribution' $e^{p_{\text{ON}}(s,L_i) + q_{\text{OFF}}(t,L_i) - \sigma(t+s)}$, that is, the constraint introduces an exponential factor that distorts the weight of the various durations. These elements are consistent with the idea of a random variable per cycle that can be averaged to compute the drift and diffusivity. However, the idea of ignoring correlations between duration of events is not valid because $\sigma$ is a function of $\lambda$. Indeed, summing the series over $n$ gives $f(\sigma, \lambda)/(1 - f(\sigma, \lambda))$ and the asymptotic behavior is then determined by the first singularity encountered as $\gamma$ is moved to the left for the inverse Laplace transform. For each value of $\lambda$, this determines a value of $\sigma$, that is $\sigma_s(\lambda)$ (see *Carballo-Pacheco et al., 2019* for the complete calculation).

What the above says more qualitatively is that the variables per cycle are correlated because of the global constraint and there is not a single-cycle effective probability distribution that can account for that because of the dependence of $\sigma$ on $\lambda$. In other words, different moments involve different configurations and distortions of the weights for the durations of the cycles. The only exception is when $d\sigma_s(\lambda)/d\lambda$ vanishes, which is the case for two close hypotheses. Indeed, from $f(\sigma_s(\lambda), \lambda) = 1$, one obtains $d\sigma_s/d\lambda \propto \partial f/\partial\lambda$ and it also holds $\partial f/\partial\lambda(\lambda = 0) = \langle (p_{\text{ON}}(s, L_2) - p_{\text{ON}}(s, L_1)) + (p_{\text{OFF}}(t, L_2) - p_{\text{ON}}(t, L_1)) \rangle$. We checked explicitly in the subsection 'Equality between drift and diffusivity' of Appendix 3 that in that limit in the formula from *Carballo-Pacheco et al., 2019* the first two terms are negligible and the diffusivity reduces to the variance of the log-likelihoods only.

Indeed, when the two hypothetical concentrations are close and the correlations can be ignored, a diffusivity per cycle can be defined naturally as

$$D_{\text{pc}} = D_3 - V^2, \tag{38}$$

where $D_3$ is the average of the squared *log*-likelihood variation (see *Equation 24*). We check in *Appendix 3—figure 3* that this formula is a good approximation for small concentration differences in the context of one binding site. In that context we have

$$D_{\text{pc}} = \frac{1}{2} \frac{1}{\nu^{-1} + (\mu L)^{-1}} \left( \frac{L_1 - L_2}{L} \right)^2, \tag{39}$$

where $\nu$ is the OFF rate and $\mu L$ is the ON rate. We find that this approximation is better than the $V = D$ approximation only when the cycle times depend weakly on the concentration (i.e, for one binding site, when $\nu$ is small, see *Appendix 3—figure 3*).

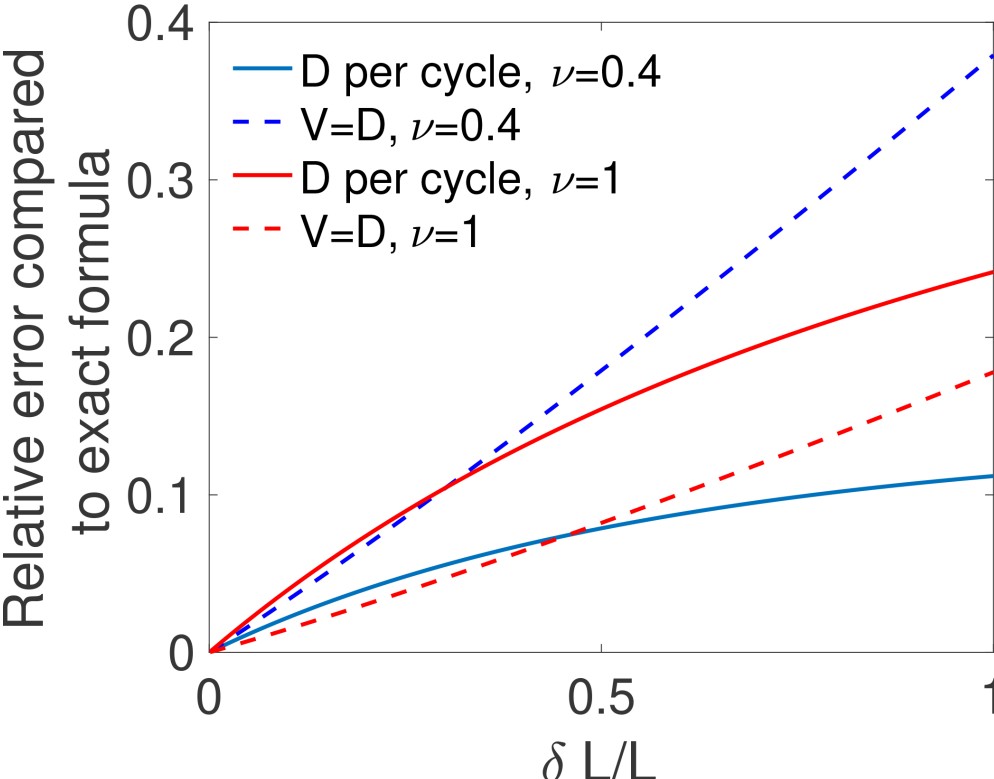

**Appendix 3—figure 3.** Comparing two approximations for the diffusivity. For different values of the relative difference in concentrations $\delta L/L$ and for one binding site with ON rate $\mu L$ and OFF rate $\nu$, we compute the mean time to decision using the exact formula from *Siggia and Vergassola, 2013* $\langle T \rangle_{\text{SV}}$ (computing $D$ according to *Equation 26*), the mean time to decision computing $D$ as a variable per cycle as in *Equation 39* $\langle T \rangle_{\text{pc}}$ and the mean time per cycle using $V = D$. We show for both the per cycle (full lines) and the $V = D$ (dotted lines) method, the relative error made in computing the mean decision time by comparing it to the exact time $\langle T \rangle_{\text{SV}}$. We find that all methods agree for small $\delta L/L$, but that the per cycle method is a better approximation only for small values of the OFF rate $\nu$ (blue curves), i.e when the times do not depend strongly on the concentration. Parameters are $e = 0.32$, $L = L_2 = 1 \; \mu m^{-3}$, $L_1 = L + \delta L$, $\mu = 1 \; \mu m^3 s^{-1}$.

# Appendix 4

## The mean first-passage time for different architectures

### Activation by multiple TF - irreversible binding all-or-nothing models

We first consider the all-or-nothing $k = 2$ two binding site cycle model depicted in Main Text *Figure 5a*.

In this model, ON events end when the complex formed by two copies of the TF unbinds. It follows that ON times are exponentially distributed and independent of the concentration $P_{\mathrm{ON}}(t) = \nu e^{-\nu t}$. Because they are independent of the concentration, they will not contribute to the log ratios. The mean ON time is given by $\tau_{\mathrm{ON}} = 1/\nu$.

The activation time (OFF time) requires two copies of the TF to bind and is the sum of the times it takes for each of them to bind. From a probabilistic point of view, this sum becomes a convolution and the OFF times are given by

$$
\begin{aligned}
P_{\mathrm{OFF}}(t,L) &= \int_0^t ds \mu_1 L e^{-\mu_1 L s} \mu_2 L e^{-\mu_2 L(t-s)} = \mu_1 \mu_2 L^2 e^{-\mu_2 L t} \int_0^t ds e^{-sL(\mu_1 - \mu_2)} \\
&= \mu_1 \mu_2 L^2 e^{-\mu_2 L t} \frac{1}{L(\mu_2 - \mu_1)} \left( e^{Lt(\mu_2 - \mu_1)} - 1 \right) = \frac{\mu_1 \mu_2 L}{\mu_2 - \mu_1} \left( e^{-\mu_1 L t} - e^{-\mu_2 L t} \right).
\end{aligned}
\tag{40}
$$

The average OFF time is $\tau_{\mathrm{OFF}} = 1/\mu_1 L + 1/\mu_2 L$.
We can now compute the drift (the ON times do not contribute to information)

$$
\begin{aligned}
V &= \left( \frac{1}{\mu_1 L} + \frac{1}{\mu_2 L} + \frac{1}{\nu} \right)^{-1} \int dt \frac{\mu_1 \mu_2 L}{\mu_2 - \mu_1} \left( e^{-\mu_1 L t} - e^{-\mu_2 L t} \right) \\
&\quad \log \left[ \frac{\mu_1 \mu_2 L_1}{\mu_2 - \mu_1} \left( e^{-\mu_1 L_1 t} - e^{-\mu_2 L_1 t} \right) \frac{\mu_2 - \mu_1}{\mu_1 \mu_2 L_2} \left( e^{-\mu_1 L_2 t} - e^{-\mu_2 L_2 t} \right)^{-1} \right] \\
&= \left( \frac{1}{\mu_1 L} + \frac{1}{\mu_2 L} + \frac{1}{\nu} \right)^{-1} \left[ \log \frac{L_1}{L_2} + \int dt \frac{\mu_1 \mu_2 L}{\mu_2 - \mu_1} \left( e^{-\mu_1 L t} - e^{-\mu_2 L t} \right) \log \frac{e^{-\mu_1 L_1 t} - e^{-\mu_2 L_1 t}}{e^{-\mu_1 L_2 t} - e^{-\mu_2 L_2 t}} \right].
\end{aligned}
\tag{41}
$$

The calculations above are only valid when the two binding rates are different $\mu_1 \neq \mu_2$. Let us now assume that $\mu_1 = \mu_2$. The ON time distribution is unchanged but the OFF time distribution is now

$$
P_{\mathrm{OFF}}(t,L) = \int_0^t ds \mu_1 L e^{-\mu_1 L s} \mu_2 L e^{-\mu_1 L(t-s)} = \mu_1^2 L^2 t e^{-\mu_1 L t} = \gamma(2, 1/\mu_1 L),
\tag{42}
$$

where we have identified $\gamma(2, 1/\mu_1 L)$, the standard Gamma distribution with exponent 2 and parameter $1/\mu_1 L$. The expression of the drift simplifies as:

$$
\begin{aligned}
V &= \left( \frac{1}{\mu_1 L} + \frac{1}{\mu_2 L} + \frac{1}{\nu} \right)^{-1} \int_0^{+\infty} dt \mu_1^2 L^2 t e^{-\mu_1 L t} \log \frac{\mu_1^2 L_1^2 t e^{-\mu_1 L_1 t}}{\mu_1^2 L_2^2 t e^{-\mu_1 L_2 t}} \\
&= \left( \frac{1}{\mu_1 L} + \frac{1}{\mu_2 L} + \frac{1}{\nu} \right)^{-1} \left[ 2 \log \frac{L_1}{L_2} + 2 \frac{L_2 - L_1}{L} \right].
\end{aligned}
\tag{43}
$$

When the two binding rates $\mu_1$ and $\mu_2$ are similar but not exactly equal the activation time distribution can be approximated by the Gamma distribution $P_{\mathrm{OFF}}(t,L) \approx \gamma \left( 2, \frac{\mu_1 \mu_2 L}{\mu_2 + \mu_1} \right)$. It is then convenient to use *Equation 43* with $\frac{\mu_1 \mu_2 L}{\mu_2 + \mu_1}$ as a parameter.

### Activation by multiple TF – irreversible binding with 1-or-more $k = 1$ activation

We now consider the non-equilibrium two binding site 1-or-more $k = 1$ activation model depicted in Main Text *Figure 5b*. The OFF times are exponentially distributed $P_{\mathrm{OFF}}(t,L) = \mu_1 L e^{-\mu_1 L t}$. They will contribute as in the one binding-site exponential model.

The ON times are now a convolution of a concentration-dependent exponential step with rate $\mu_2 L$ and a concentration-independent unbinding with rate $\nu$. Their distribution is

$$P_{\mathrm{ON}}(t,L) = \int_0^t ds \mu_2 L e^{-\mu_2 L s} \nu e^{-\nu(t-s)} = \frac{\mu_2 L \nu}{\nu - \mu_2 L}\left(e^{-\mu_2 L t} - e^{-\nu t}\right). \tag{44}$$

The expression is valid for $\mu_2 L \neq \nu$ and, as previously, reduces to a $\gamma$ distribution with exponent two in case of equality.

The corresponding drift is

$$
\begin{aligned}
V &= \left(\frac{1}{\mu_1 L} + \frac{1}{\mu_2 L} + \frac{1}{\nu}\right)^{-1}\left[\int_0^\infty dt \mu_1 L e^{-\mu_1 L t} \log \frac{\mu_1 L_1 e^{-\mu_1 L_1 t}}{\mu_1 L_2 e^{-\mu_1 L_2 t}}\right.\\
&\quad + \left.\int_0^\infty dt \frac{\mu_2 L \nu}{\nu - \mu_2 L}\left(e^{-\mu_2 L t} - e^{-\nu t}\right)\log\left[\frac{\mu_2 L_1 \nu}{\nu - \mu_2 L_1}\frac{\nu - \mu_2 L_2}{\mu_2 L_2 \nu}\frac{(e^{-\mu_2 L_1 t} - e^{-\nu t})}{(e^{-\mu_2 L_2 t} - e^{-\nu t})}\right]\right]\\
&= \left(\frac{1}{\mu_1 L} + \frac{1}{\mu_2 L} + \frac{1}{\nu}\right)^{-1}\left[2\log\frac{L_1}{L_2} + \frac{L_2 - L_1}{L} + \log\frac{\nu - \mu_2 L_2}{\nu - \mu_2 L_1}\right.\\
&\quad + \left.\int_0^\infty dt \frac{\mu_2 L \nu}{\nu - \mu_2 L}\left(e^{-\mu_2 L t} - e^{-\nu t}\right)\log\left(\frac{e^{-\mu_2 L_1 t} - e^{-\nu t}}{e^{-\mu_2 L_2 t} - e^{-\nu t}}\right)\right].
\end{aligned}
\tag{45}
$$

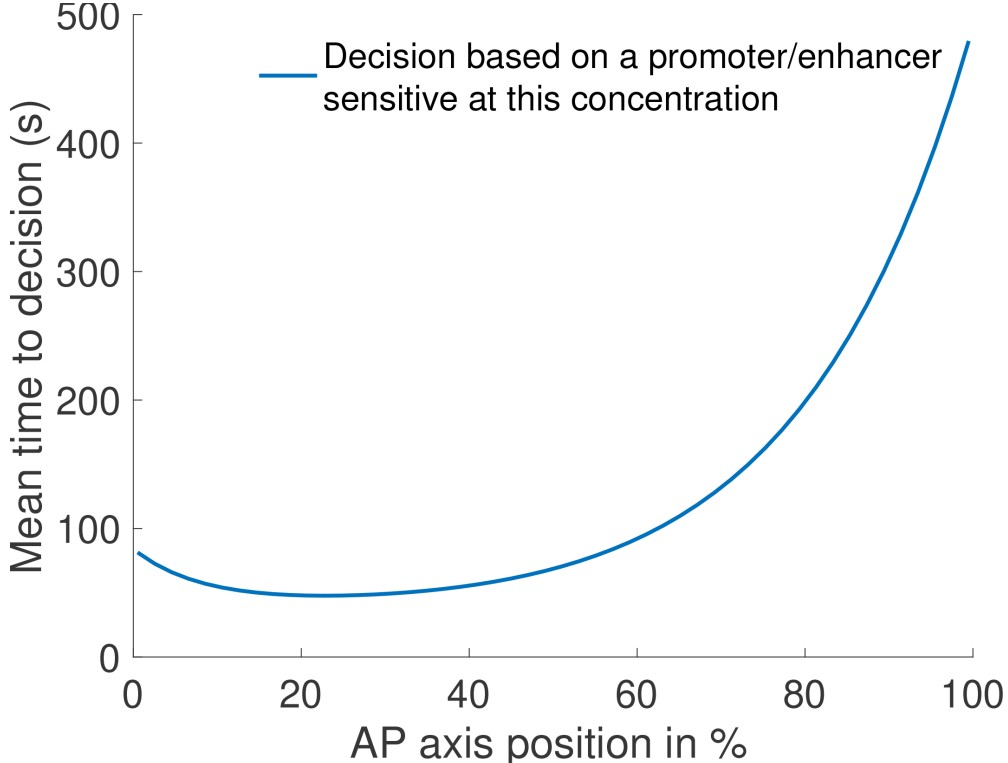

**Appendix 4—figure 1.** Mean time to decision across the embryo based on local Bicoid concentration difference. For each position across the embryo, we compute the performance of the fastest identified architecture ($k = 1$, no Hill coefficient constraint) if it was implemented to distinguish between two concentrations with 10% relative variation, and an average equal to the Bicoid concentration at the AP position denoted on the x-axes. We find that our architecture performs well in the anterior region but is slow in the posterior region, due to low concentrations.

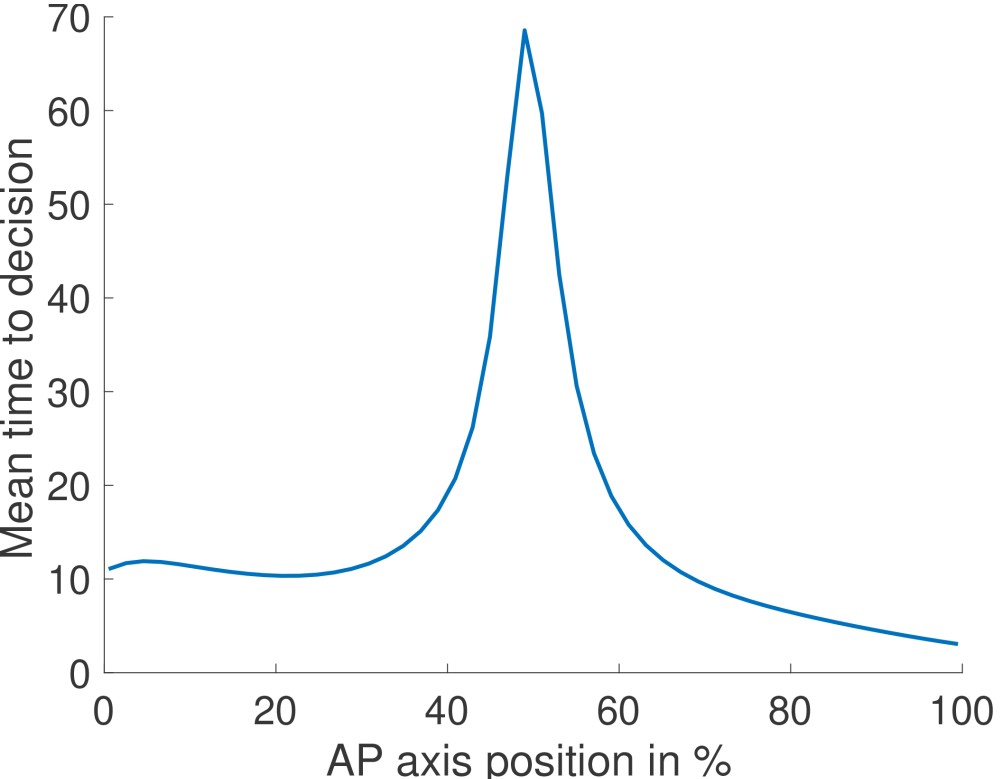

**Appendix 4—figure 2.** Mean time to decision across the embryo assuming fixed thresholds and log-likelihood functions. We compute for each position along the embryo the mean time to decision for the fastest architecture identified with $k = 1$ and no Hill coefficient constraints. We assume one biological mechanism for the readout meaning that, at every position, the log-likelihood of the waiting times are computed using the same function (one function across the embryo for the ON times and one for the OFF times). This function is determined by the log-likelihoods at the two edges of the mid-embryo region because it is the hardest discrimination task. The thresholds corresponding to deciding for anterior and posterior are also fixed throughout the embryo. We find that our model predicts that nuclei located close to the boundary take more time to trigger a decision.

## Appendix 5

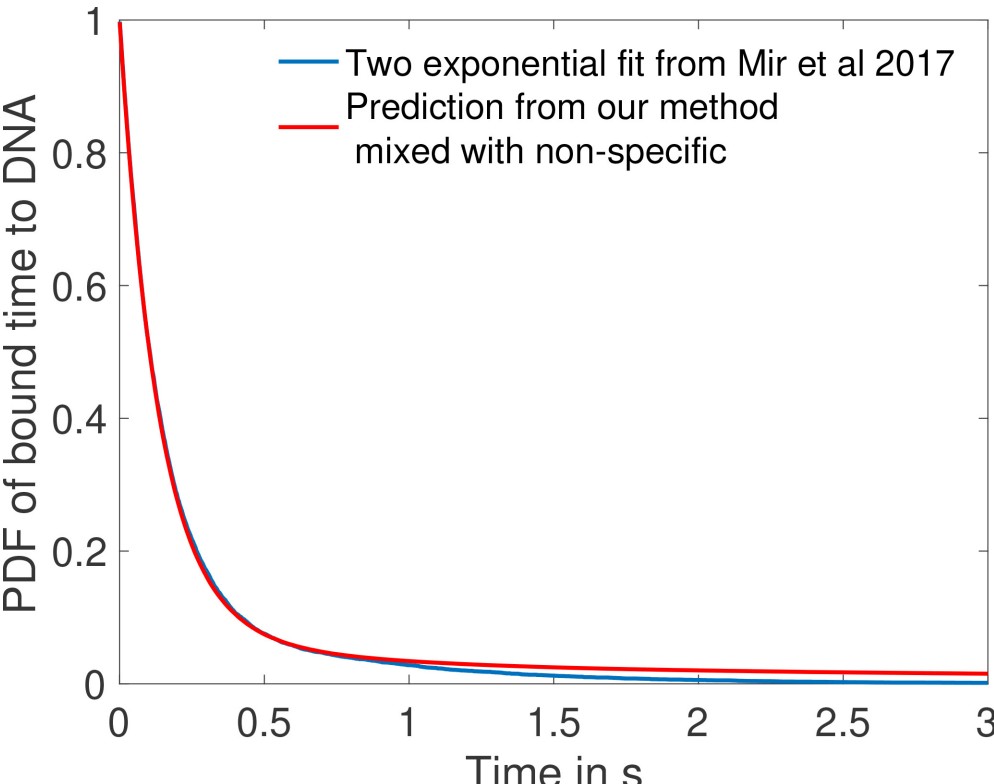

**Appendix 5—figure 1.** Comparison of fit to data from *Mir et al., 2017* and prediction from one of the fastest architectures for time spent bound to DNA by Bicoid molecules. In blue we show the two exponential fit for the pdf of the time spent bound to the DNA by Bicoid molecules at the boundary for the Zelda null conditions in *Mir et al., 2017*. They fit two exponentials to the data obtaining a coefficient of determination above 0.99 for a least square fit. Their fit finds 12.9% specific traces with mean 0.629 s and 87.1% non-specific traces with mean 0.131 s. In blue we show the PDF of time spent bound to DNA for a mix of 87.1% non-specific traces with the same mean 0.131 s and 12.9% specific traces drawn from the distribution of time spent to DNA based on the fastest promoter architecture identified for $k = 2$, $H>5$ (obtained from Monte Carlo simulation). Our prediction fits the blue curve from *Mir et al., 2017* with a coefficient of determination above 0.99 as well.

# Appendix 6

## All-or-nothing vs 1-or-more activation

In this section, we give a more detailed derivation of the result given in the subsection 'Activation rules' of the main text. Specifically, we compare two activation rules for the same promoter architecture: two binding sites with binding rates $\mu_1$ and $\mu_2$. We consider that the two binding sites bind TF independently so that $\mu_1 = 2\mu_2$ (there are two free targets for the first binding). The two copies of the TF unbind together with unbinding rate $\nu$. It is a cycle model. We consider the 1-or-more $k = 1$ activation rule (Main Text *Figure 5b*) and the all-or-nothing activation rule (Main Text *Figure 5a*). We will prove that the fastest activation scheme is the all-or-nothing scheme when $\mu_1 L > \nu$ and 1-or-more when $\mu_1 L < \nu$. This result corresponds to the following intuitive idea : given a choice to associate the second binding with either the first binding or the unbinding for the readout by the cell, it is more informative to convolve it with the fastest of the rates to maximize the information extracted from it. This is in general true if the time per cycle is the same for the different architectures considered.

The total time it takes to go through a cycle is the same for the two architectures: $\tau_{\text{ON}} + \tau_{\text{OFF}} = 1\mu_1 + 1/\mu_2 + 1/\nu$, so the difference in performance will come from the amount of information extracted per cycle. We are interested in the limit of similar concentrations, that is, $L = L_2$ and $L_1 = L + \delta L$, $\delta L \ll L$. Since $D = V$ and the decision time is inversely proportional to $V$, it is enough to compare $V^{(\text{all})}$ (the drift in the all-or-nothing scheme) with $V^{(\text{one})}$ (the drift in the 1-or-more scheme) to determine the fastest architecture.

To compute $V^{(\text{all})}$, we start from *Equation 41*, plugging in $L_2 = L$, $L_1 = L + \delta L$ and $\mu_1 = 2\mu_2$. We expand for $\delta L/L \to 0$:

$$
\begin{aligned}
V^{\text{all}}(\tau_{\text{ON}} + \tau_{\text{OFF}}) &= \log\frac{L+\delta L}{L} + \int dt\, 2\mu_2 L\big(e^{-\mu_2 Lt} - e^{-2\mu_2 Lt}\big)\log\frac{e^{-2\mu_2(L+\delta L)t} - e^{-\mu_2(L+\delta L)t}}{e^{-2\mu_2 Lt} - e^{-\mu_2 Lt}} \\
&= \frac{\delta L}{L} - \frac{\delta L^2}{2L^2} + \int dt\, 2\mu_2 L\big(e^{-\mu_2 Lt} - e^{-2\mu_2 Lt}\big)\log\frac{e^{-\mu_2\delta Lt} - e^{-\mu_2 t(L+2\delta L)}}{1 - e^{-\mu_2 Lt}} \\
&\simeq \frac{\delta L}{L} - \frac{\delta L^2}{2L^2} + \int dt\, 2\mu_2 L\big(e^{-\mu_2 Lt} - e^{-2\mu_2 Lt}\big)\log\Big[1 - \delta L\mu_2 t\frac{1-2e^{-\mu_2 Lt}}{1-e^{-\mu_2 Lt}} + \delta L^2\frac{\mu_2^2 t^2}{2}\frac{1-4e^{-\mu_2 Lt}}{1-e^{-\mu_2 Lt}}\Big] \\
&\simeq \frac{\delta L}{L} - \frac{\delta L^2}{2L^2} + \int dt\, \mu_2 L e^{-\mu_2 Lt}\Big[-2\delta L\mu_2 t\big(1-2e^{-\mu_2 Lt}\big) \\
&\quad + \delta L^2\mu_2^2 t^2\big(1-4e^{-\mu_2 Lt}\big) - \delta L^2\mu_2^2 t^2\frac{(1-2e^{-\mu_2 Lt})^2}{1-e^{-\mu_2 Lt}}\Big] \\
&= \frac{\delta L}{L} - \frac{\delta L^2}{2L^2} - \frac{2\delta L}{L} + \frac{\delta L}{L} - \int dt\, \mu_2^3 t^2 L e^{-\mu_2 Lt}\frac{1}{e^{L\mu_2 t}-1} = -\frac{\delta L^2}{2L^2} - \frac{2\delta L^2}{L^2} + \frac{2\zeta(3)\delta L^2}{L^2},
\end{aligned}
$$

(46)

where $\zeta$ is the Riemann zeta function. Eventually we have

$$
V^{\text{all}}(\tau_{\text{ON}} + \tau_{\text{OFF}}) = \frac{\delta L^2}{2L^2}[4\zeta(3) - 5].
$$

(47)

While the expansion of $V^{\text{one}}$ does not take a simple form in general, it can be computed for the specific value of $\mu_1 L = \nu$. When $\nu = L\mu_1 = 2L\mu_2$, $L = L_2$ and $L_1 = L + \delta L$, expanding to second order *Equation 45* becomes

$$
\begin{aligned}
V^{(\text{one})}(\tau_{\text{ON}} + \tau_{\text{OFF}}) &= 2\log\frac{L+\delta L}{L} - \frac{\delta L}{L} + \log\frac{L}{L-\delta L} \\
&\quad + \int_0^{+\infty} dt\, 2\mu_2 L\big(e^{-\mu_2 Lt} - e^{-2L\mu_2 t}\big)\log\left[\frac{\big(e^{-\mu_2(L+\delta L)t} - e^{-2L\mu_2 t}\big)}{\big(e^{-\mu_2 Lt} - e^{-2L\mu_2 t}\big)}\right] \\
&\simeq \frac{2\delta L}{L} - \frac{\delta L^2}{2L^2} + \int_0^{+\infty} dt\, 2\mu_2 L\left(-\delta L\mu_2 t e^{-L\mu_2 t} - \frac{\delta L^2\mu_2^2 t^2}{2}\frac{e^{-L\mu_2 t}}{1-e^{\mu_2 Lt}}\right) \\
&= \frac{2\delta L}{L} - \frac{\delta L^2}{2L^2} - \frac{2\delta L}{L} - 2\frac{\delta L^2}{L^2} + 2\zeta(3)\frac{\delta L^2}{L^2} = V^{(\text{all})}(\tau_{\text{ON}} + \tau_{\text{OFF}}).
\end{aligned}
$$

(48)

We can now use the above equality and the following arguments to reach the conclusion stated

at the beginning of the section. $V^{(\mathrm{all})}$ is independent of $\nu$ because no information is gained about the concentrations from the step involving unbinding. Conversely, $V^{(\mathrm{one})}$ is an increasing function of $\nu$: as $\nu$ decreases, the unbinding part takes over in the convolution for the ON times. Since the difference between the two convolutions can only decrease as $\nu$ decreases, it becomes harder and harder to differentiate the two ON time distributions, their Kullback-Leibler divergence becomes smaller, and the drift term $V^{(\mathrm{one})}$ is reduced.

In sum, for $\nu = \mu_1 L$, $V^{(\mathrm{all})} = V^{(\mathrm{one})}$; $V^{(\mathrm{all})}$ is independent of $\nu$ whilst $V^{(\mathrm{one})}$ increases with $\nu$. We conclude that for $\nu > \mu_1 L$, $V^{(\mathrm{one})} > V^{(\mathrm{all})}$ and the 1-or-more scheme is preferred. Conversely, for $\nu < \mu_1 L$, $V^{(\mathrm{one})} < V^{(\mathrm{all})}$ and the all-or-nothing scheme is preferred.

# Appendix 7

## Comparing one and two binding-site architectures

In this section we compare the performance of a one binding-site architecture (Main Text *Figure 5d*) to that of an equilibrium all-or-nothing $k = 2$ two binding-site architecture (Main Text *Figure 5c*). The motivation is to provide detailed explanations for the results given in subsection 'How the number of binding sites affects decisions' of the main text. To rank their performance, we compare, as in the previous appendix 6, the drift for the two architectures.

To do the comparison, we optimize the two binding site architecture rates $\mu_1$, $\mu_2$ and $\nu_1$ for a fixed value of the error rate $e$, the real concentration $L$, the hypothetical concentrations $L_1$ and $L_2$ and the second unbinding rate $\nu_2$. The fixed second unbinding rate sets a time scale for the process. For concreteness, we suppose that the real concentration $L = L_2$ and decision is hard $L_1 = L + \delta L$, $\delta L / L << 1$.

Decisions can be trivially sped up by increasing indefinitely all the rates to very high values. To avoid this, in the optimization process we set an upper bound to be $5s^{-1}$. We choose this upper bound to be smaller than the largest values of $\nu_2$ considered ($9s^{-1}$) and larger than the smaller values of $\nu_2$ considered ($1s^{-1}$). From a biological point of view, this upper bound for the ON rates can correspond to the diffusion limited arrival rate. For the OFF rates, it can correspond to a minimum bound time required to trigger a downstream mechanism (for instance, the activation of the gene). We check in *Appendix 7—figure 2* that curves and effects similar to those discussed below are obtained if the upper bounds are modified.

For all values of parameters, we find that the optimal ON rates are maximal ($\mu_1 = \mu_2 = 5s^{-1}$). We observed and discussed in subsection 'How the number of binding sites affects decision' of the main text that for certain values of the parameters ($\nu$ and $L$), the optimal first unbinding rate $\nu_1$ reaches the upper bound $5s^{-1}$ while for smaller values the optimal first unbinding rate remains at 0 (Main Text *Figure 5f* inset). The transition between the two regions is sharp. Setting $\nu_1$ to 0 is equivalent to having an effective one binding site model because the promoter never goes back to the 0 state. For that reason, we want to compare the performance of a two binding site model with parameters $\mu_1$, $\mu_2$, $\nu_1$ and $\nu_2$ to that of a one binding site model with parameters $\mu_2$ and $\nu_2$.

Since we are in the limit of small concentration differences, the speed of decision is set by $V$ and $\tau = \tau_{\mathrm{ON}} + \tau_{\mathrm{OFF}}$. We denote the mean drifts of the one and the two binding-site models as $V^{(1)}$ and $V^{(2)}$, respectively. Similarly, $\tau_1$ and $\tau_2$ are the mean times per cycle of the two models.

The one binding-site architecture activates with rate $\mu_2 L$ and deactivates with rate $\nu_2$. Both waiting times for ON and OFF expression states are exponential. Following results in the paragraph 'Mean decision time : connecting drift-diffusion and Wald's approaches' of the main text, we have $\tau_1 = 1/\nu_2 + 1/(\mu_2 L)$ and $V^{(1)} = (1/\nu_2 + 1/\mu_2 L)[\log(L_1/L_2) + (L_2 - L_1)/L]$. We conclude that :

$$V^{(1)} = \left[\frac{1}{\nu_2} + \frac{1}{\mu_2 L}\right]^{-1}\left[\log\frac{L+\delta L}{L} - \frac{\delta L}{L}\right]. \tag{49}$$

In the two binding site architecture, the ON times are exponentially distributed with rate $\nu_2$. The OFF times are more complex as they can result from several cycles from a promoter with a binding site occupied to an empty promoter before finally switching to two full binding sites and gene activation. We compute $P_{\mathrm{OFF}}(t,L)$ using the modified transition matrix of the Markov chain with the ON states acting as sinks as described in Appendix 3. The OFF time distribution is given by the derivative of *Equation 12*.

In the two binding site model, the transition matrix is a 3 by 3 matrix and can be diagonalized analytically to compute explicitly the exponential in *Equation 12* . We set $\mu_1 L_{\mathrm{eq}} = \mu_2 L_{\mathrm{eq}} = \nu_1$ since it is always the case in the identified optimal architectures for the two binding site model (where $L_{\mathrm{eq}}$ is the specific value of $L$ for which upper bounds on $\nu_1$ and $\mu L$ are equal, $L_{\mathrm{eq}} = 1 \ \mu m^3$ in Main Text *Figure 5f*). We compute the distribution of the OFF times explicitly and find

$$P_{\mathrm{OFF}}(L,t) = \frac{\nu_1 r}{2+8r}e^{-\frac{1}{2}\nu_1 t(1+2r+\sqrt{1+4r})}\left[1+4r+\sqrt{1+4r}+(1+4r-\sqrt{1+4r})e^{\sqrt{1+4r}\nu_1 t}\right], \tag{50}$$

where $r = L/L_{\mathrm{eq}}$.

Computing the mean of the distribution in *Equation 50* , we find that the average time for a cycle in the two binding site architecture is

$$\tau^{(2)} = \frac{1+r}{r}\frac{1}{\nu_1 r} + \frac{1}{\nu_2}.\tag{51}$$

We can now numerically integrate $V^{(2)} = \tau^{(2)} \int P_{\text{OFF}}(t,L)\log(P_{\text{OFF}}(t,L_1)/P_{\text{OFF}}(t,L))$. We use this simplified formula to compare the performance of the two architectures and recover the results of Main Text *Figure 5f* . We show an example for a specific value of $L$, varying $\nu_2$ in *Appendix 7—figure 1*.

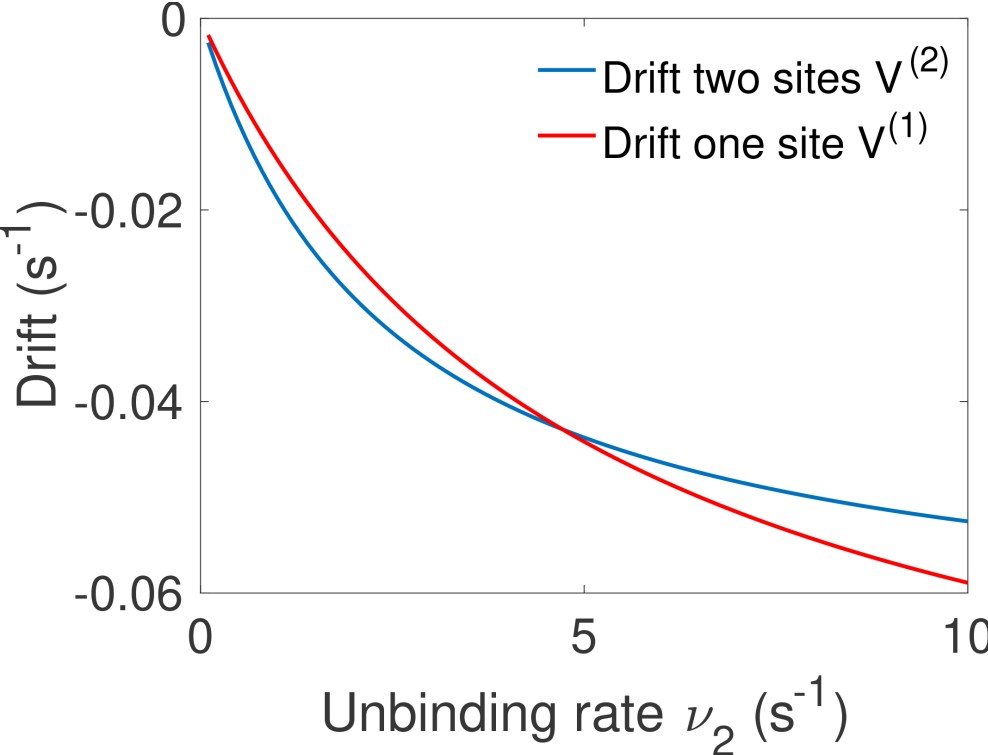

**Appendix 7—figure 1.** Comparison of the drift for the one and two binding site equilibrium architectures. We compare the drift $V^{(1)}$ of the one binding-site architecture of Main Text *Figure 5d* and the drift $V^{(2)}$ for the two binding site equilibrium architecture of Main Text *Figure 5c*. Both have the same parameters $L = L_2 = 1\mu m^{-3}$, $L_1 = 1.1\mu m^{-3}$, $\mu_1 = \mu_2 = 5\mu m^3 s^{-1}$, $\nu_1 = 5s^{-1}$ and we vary $\nu_2$. The fastest architecture is the one with the highest absolute value of the drift (lowest on the graph). We recover the transition observed in Main Text *Figure 5f* between the optimal architectures.

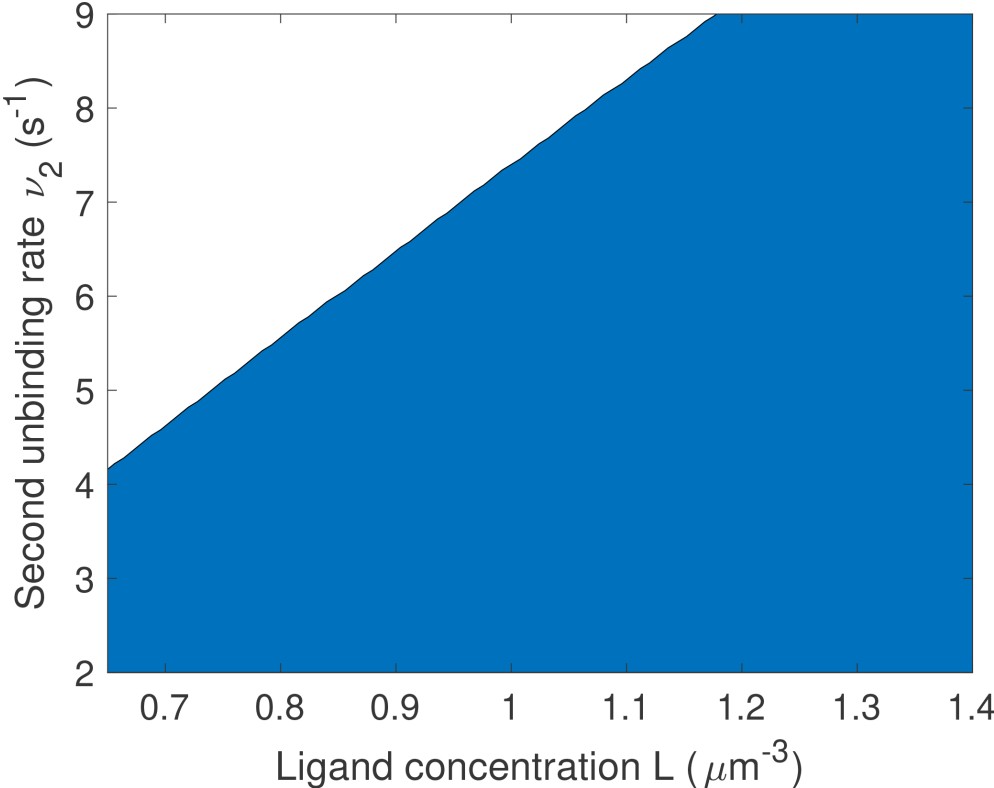

**Appendix 7—figure 2.** Changing the bounds of the rates does not change the qualitative results on the optimal number of binding sites. We proceed as in Main Text *Figure 5f*: for a given value of *L* and ν we compare one- and two-binding site architectures. The blue region corresponds to two binding site promoter being optimal and the white region to one binding site promoter being optimal. Parameters are the same as in Main Text *Figure 5f* except that the upper bounds for $\mu_1$ and $\mu_2$ are increased to 5 $\mu m^3 s^{-1}$ and that of $\nu_1$ is increased to $5.5 s^{-1}$. We find that the results are qualitatively similar to that of Main Text *Figure 5f*. although the transition to optimal one binding site region is shifted up.

## Appendix 8

## Out of equilibrium architectures and averaging over several steps

In some architectures where several copies of the TF can bind or unbind at once, there can be correlations between successive ON and OFF times. This depends on the structure of the promoter Markov chain. It is made of OFF and ON states. There can be correlations between the OFF and ON times if the chain can enter the ON state subgraph through different ON states, coming from different OFF states (or the same situation reverting ON and OFF states). In that case, the time that the chain will spend in the ON state will depend on the initial entry state, and so it will depend on the OFF state from which the chain entered the ON subgraph, giving a correlation with the previous OFF time to the ON time. If the structure is particularly complex, these correlations can span over several ON-OFF cycle. We do however assume that the chain is ergodic so that they vanish at long times.

To deal with this situation, a solution is to average the contribution to the information of ON/OFF events over several ON/OFF times, until the correlation with the initial times is lost. The event in the log ratio becomes a series of ON and OFF times and their joint probability (as they no longer factorize). The next series of events can be considered approximately (or exactly in certain cases) independent of the previous series and the rest of the theory can be applied to this sum of independent variables.

We did not explore these architectures in detail as they are extremely complex, do not seem to increase the performance of the promoter for the readout, and most likely the type of information about the concentration that is hidden in the correlation between ON and OFF would be very hard to decode for downstream mechanisms.

## Appendix 9

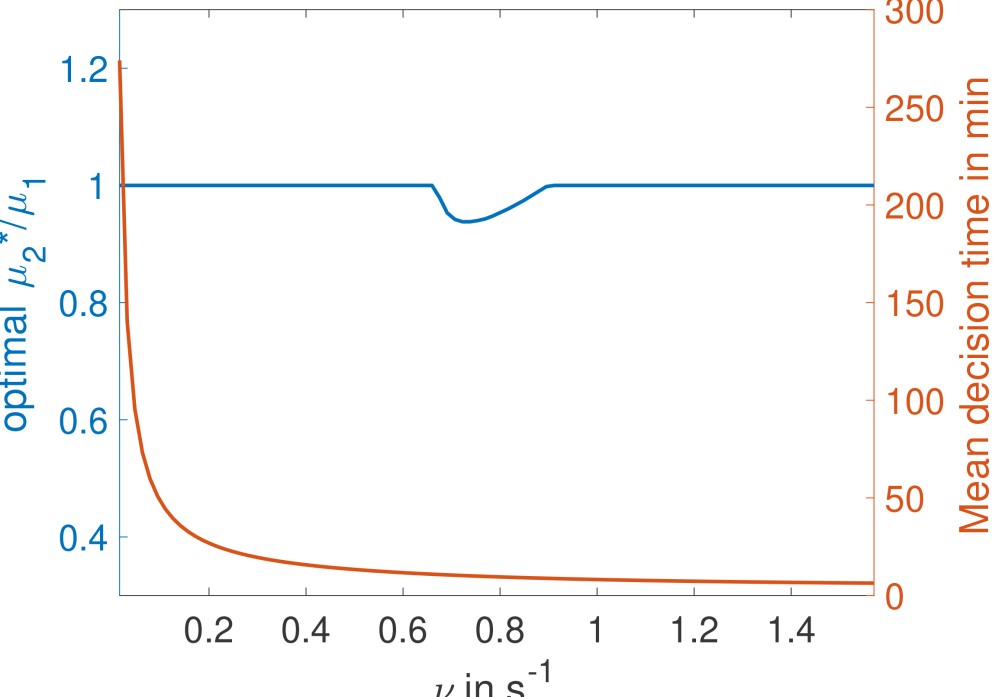

**Appendix 9—figure 1.** Weak binding sites are optimal for a range of parameters. In the $k = 1$, two binding site cycle architecture of Main Text *Figure 5c*, we fix $\nu$ and $L$ and optimize for second binding rate $\mu_2$ for a given value of $\mu_1$, imposing $\mu_2 \leq \mu_1$. We identify the value $\mu_2^*$ for which the architecture is fastest and plot the ratio $\mu_2^*/\mu_1$. We find that for certain values of $\nu$ the second binding is not as fast as it could be, corresponding to a weaker binding site (binding probability is below one and the ON rate is below the diffusion limit). Parameters are $L = 0.5~\mu m^{-3}$, $L_2 = L$, $L_1 = 0.9L$, $e = 5\%$, $\mu_1 = 1~\mu m^3 s^{-1}$.

## Appendix 10

### Parameters for Main Text *Figure 6*

Parameters for Main Text *Figure 5* panel a, panel b blue full line, panel c red and panel d red are those identified as optimal for $k = 3$, $H > 4$: $\mu_1 = 0.1262\ \mu m^3 s^{-1}$, $\mu_2 = 0.1104\ \mu m^3 s^{-1}$, $\mu_3 = 0.0868\ \mu m^3 s^{-1}$, $\mu_4 = 0.0662\ \mu m^3 s^{-1}$, $\mu_5 = 0.0441\ \mu m^3 s^{-1}$, $\mu_6 = 0.0220\ \mu m^3 s^{-1}$, $\nu_1 = 1.0793 s^{-1}$, $\nu_2 = 0.5933 s^{-1}$, $\nu_3 = 0.7961 s^{-1}$, $\nu_4 = 0.5169 s^{-1}$, $\nu_5 = 0.0908 s^{-1}$, $\nu_6 = 0.1225 s^{-1}$. Parameters for Main Text *Figure 5* panel b green dashed line, panel c blue and panel d blue are those identified as optimal for $k = 2$, $H > 4$: $\mu_1 = 0.1325\ \mu m^3 s^{-1}$, $\mu_2 = 0.1088\ \mu m^3 s^{-1}$, $\mu_3 = 0.0861\ \mu m^3 s^{-1}$, $\mu_4 = 0.0662\ \mu m^3 s^{-1}$, $\mu_5 = 0.0431\ \mu m^3 s^{-1}$, $\mu_6 = 0.0215\ \mu m^3 s^{-1}$, $\nu_1 = 1.0384 s^{-1}$, $\nu_2 = 1.5926 s^{-1}$, $\nu_3 = 0.6007 s^{-1}$, $\nu_4 = 0.2966 s^{-1}$, $\nu_5 = 0.1581 s^{-1}$, $\nu_6 = 0.0947 s^{-1}$. Other lines in panel b correspond to architectures that are close to optimality, with parameters in the same range as the ones given above.

## Appendix 11

### Approximating the log-likelihood function with RNA levels

In this section, we give details of the calculations linked to the model of RNA production presented in the section 'Estimating the log-likelihood function with RNA concentrations' of the main text. In this model, we assume that when the promoter enters the ON state, it takes a time $d_{\mathrm{ON}}$ before polymerase is loaded. This time can be associated with the formation of polymerase clusters at the transcription sites (*Cho et al., 2016a*; *Cho et al., 2016b* or of more complex multifactor transient hubs recently observed in *Drosophila* embryos (*Park et al., 2019*). Once the promoter returns to the OFF state, our model also includes a delay $d_{\mathrm{OFF}}$ during which the gene continues loading polymerase. This delay can be associated with the dissolution of the clusters and hubs mentioned above or simply with the inertia of polymerase loading. Under these assumptions, the contribution of an ON time $t$ to RNA level is

$$\delta_{\mathrm{RNA}}^{\mathrm{ON}}(t) = r_{\mathrm{OFF}}t + \min(t, d_{\mathrm{ON}})r_b + \max(t - d_{\mathrm{ON}}, 0)r_{\mathrm{ON}}. \tag{52}$$

The contribution of an OFF time $s$ to RNA level is given by

$$\delta_{\mathrm{RNA}}^{\mathrm{OFF}}(s) = r_{\mathrm{OFF}}s + \min(s, d_{\mathrm{OFF}})r_{\mathrm{ON}} + \max(s - d_{\mathrm{OFF}}, 0)r_b. \tag{53}$$

We have that the RNA level at time T is

$$\mathrm{RNA}(T) = \mathrm{RNA}(0) + \sum_i \delta_{\mathrm{RNA}}^{\mathrm{ON}}(t_i) + \sum_j \delta_{\mathrm{RNA}}^{\mathrm{OFF}}(s_j), \tag{54}$$

where the $\{t_i\}_i$ are the ON times up to time $T$ and $\{s_j\}_j$ the OFF times up to time T.

We look for sets of parameters that give a positive drift $V_1$ when $L>1.05L_0$ and negative drift $V_2$ when $L<0.95L_0$. For such architectures, the ON and OFF production levels will roughly balance each other at the boundary. To ensure a positive RNA level at the boundary, we redefine the ON rate: $r_{\mathrm{ON}}^* = r_{\mathrm{ON}} - r_b > 0$ so that the basal rate is now a factor proportional to time in both ON time and OFF time RNA contributions. In that sense, the basal rate is a systematic drift of the RNA level that is proportional to time and can be removed from the equations by defining moving boundaries $c_1(t) = c_1(0) + r_b t$ and $c_2(t) = c_2(0) + r_b t$. With these redefined rates we now look for architectures such that the drift at the anterior boundary is positive:

$$V_1 = \int_t P_{\mathrm{ON}}(t, 1.05L_0)\big[tr_{\mathrm{OFF}} + (t - d_{\mathrm{ON}})^+ r_{\mathrm{ON}}^*\big] + \int_s P_{\mathrm{OFF}}(t, 1.05L)\big[tr_{\mathrm{OFF}} + \min(t, d_{\mathrm{OFF}})r_{\mathrm{ON}}^*\big] > 0, \tag{55}$$

and the drift at the posterior boundary is negative:

$$V_2 = \int_t P_{\mathrm{ON}}(t, 0.95L_0)\big[tr_{\mathrm{OFF}} + (t - d_{\mathrm{ON}})^+ r_{\mathrm{ON}}^*\big] + \int_s P_{\mathrm{OFF}}(t, 0.95L)\big[tr_{\mathrm{OFF}} + \min(t, d_{\mathrm{OFF}})r_{\mathrm{ON}}^*\big] < 0. \tag{56}$$

Specifically, we look for architectures that satisfy these conditions and for which the ratio of the drift over the variance contributions to the log-likelihood is as large as possible to avoid dynamics that are dominated by noise. Finally we look for a set of threshold concentrations that give the most favorable speed-accuracy tradeoff. We show the results of our non-exhaustive search in **Appendix 11—figure 1**. For one of the architectures that fall below the error level of 32% and the decision time threshold of 3 min, we check the RNA expression profile (Main text *Figure 7*). For RNA degradation, we use the rate $r_{\mathrm{OFF}}[\mathrm{RNA}]/(1 + [\mathrm{RNA}])$ that reduces to $r_{\mathrm{OFF}}$ close to the boundary and yields the desired linear in time degradation level, but that is proportional to $[\mathrm{RNA}]$ for small values of the concentration. For this function, we compute the average number of RNA molecules in the cell after 3 min of dynamics (Main text *Figure 7*) and find that the RNA profile has a Hill coefficient of 5.2.

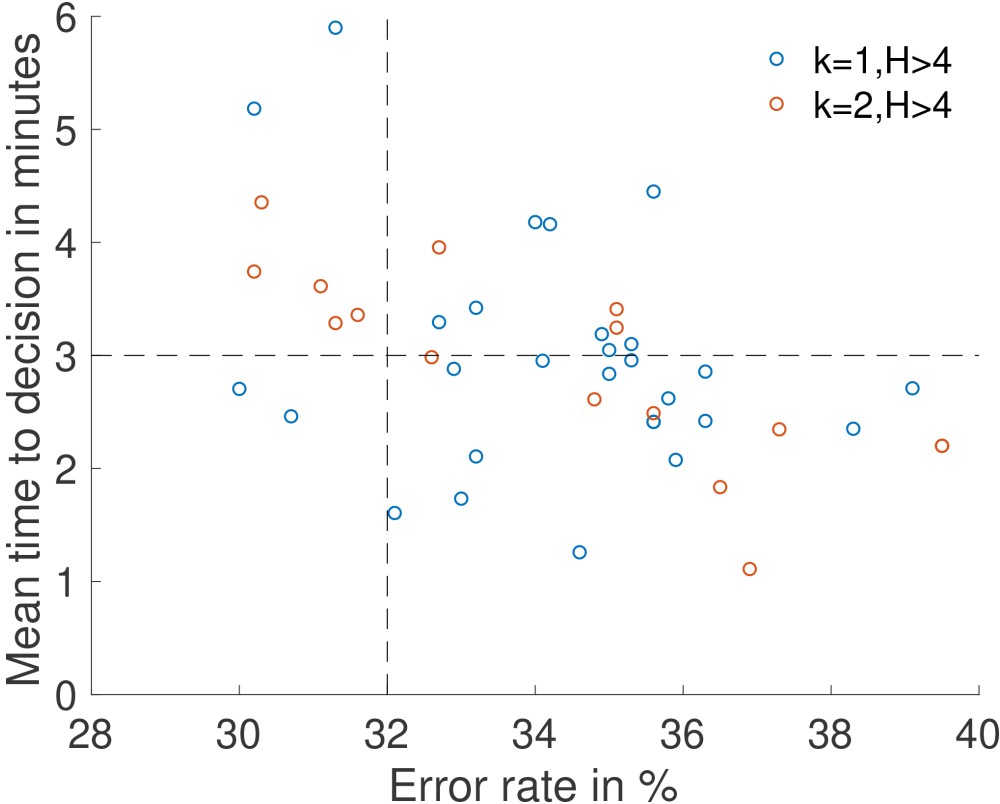

**Appendix 11—figure 1.** Results of a search for sets of parameters $r_{\mathrm{ON}}$, $r_{\mathrm{OFF}}$, $r_b$, $d_{\mathrm{ON}}$, $d_{\mathrm{OFF}}$, $c_1$ and $c_2$ that lead to an accurate decision ($e<32\%$) in a short time ($\langle T \rangle<180\mathrm{s}$). The ON and OFF times are drawn from the fastest architectures identified with rules $k=1$, $H>4$ (blue dots) and $k=2$, $H>4$ (red dots). Concentrations are $L_1 = 1.05 L_0$ and $L_2 = 0.95 L_0$.

## Appendix 12

### Exploring the effect of joint enhancer and promoter dynamics

In this section, we explore the effect of the computational power of an enhancer when added to that of the promoter. Bicoid is known to target many gene enhancers (*Driever and Nüsslein-Volhard, 1988*; *Struhl et al., 1989* following complex concentration dependent patterns (*Hannon et al., 2017*) . We choose one of the many possible models of enhancer dynamics as en example. For simplicity, we assume a two-state enhancer that is made of one Bicoid-binding site, with dynamics independent of the local promoter dynamics. We assume that the enhancer turns on with a rate $e_{ON}$ and turns off with a rate $e_{OFF}$ following Markovian dynamics (i.e the waiting times are exponentially distributed). We take a simple rule for gene activity: the gene is ON when both the promoter (which is still a six binding site promoter as in the previous models) and the enhancer are in the ON states.

From a mathematical point of view, the complete system can be viewed as a 14-state Markov chain with states $(i, j)$, $0 \leq i \leq 6$, $0 \leq j \leq 1$ corresponding to the promoter having $i$ Bicoid molecules bound and the enhancer having $j$ Bicoid molecules bound. We use this Markov chain to compute the waiting time statistics of the ON and OFF states of the hunchback gene activity. An added difficulty is that the OFF state can now be entered through two transitions: the promoter turning OFF, or the enhancer turning OFF. To account for this degeneracy, we compute the OFF time statistics for each transition. We then compute the complete probability distribution using a weighted average of the two distributions with specific transitions, where the weights are the probability of each transition happening first. We check that this method predicts the correct waiting times (see *Appendix 12— figure 1*). We note that this method overlooks the correlation between successive ON and OFF times. A perfect Bayesian machine could take advantage of these correlations to improve its estimate of the concentration. Here we assume that the cell cannot.

Once the waiting times for the ON and OFF states of the gene are computed, the drift and diffusion of the SPRT process is obtained by integration and the first passage time to decision between anterior and posterior given by *Equation 32*. Using that scheme, we optimize the parameters of the enhancers and promoter for the fastest decision with a given error between the anterior and posterior boundary regions. As an example, we picked the case $k = 2$ for the promoter activation rule and imposed $H > 4$. We still impose that half the nuclei at the boundary be active on average (i.e the product of the activities of the enhancer and the promoter at the boundary is equal to $1/2$). We also limit the binding rate to be diffusion limited. We find that enhancer dynamics improve the performance of the cell: the optimal scheme performs the decision in an average of $100s$ instead of $108s$ for the promoter alone. In these optimal schemes the enhancer is mostly ON due to slower binding dynamics than the promoter (only one binding site versus 6). Assuming the enhancer target is six times larger (or equivalently that the enhancer also has six binding sites), but that the enhancers dynamics can still be approximated by exponential waiting times, we find that in the optimal scheme, the enhancer is ON for a reasonable fraction of the time (about 75 %) and that the performance is improved down to $\simeq 70s$.

Enhancer presence seems to improve the flexibility of promoter states to get good performance while maintaining a high Hill coefficient and half the genes active at the boundary.

A complete study of enhancer dynamics is beyond the scope of this paper but would include other models such as genes that activate when either the enhancer or the promoter are ON, or generally dynamics where the enhancer state influences the transition rates of the promoter.

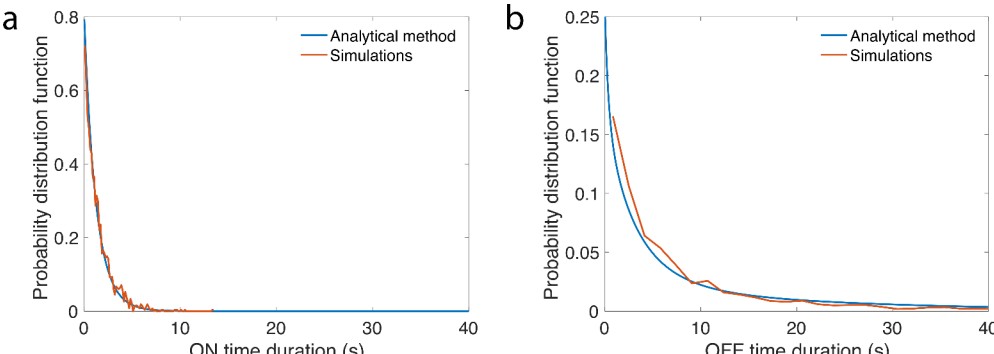

**Appendix 12—figure 1.** Computing gene ON and OFF times statistics for the enhancer dynamics. We check that our analytical method (see Appendix 12) properly predicts the ON (**a**) and OFF (**b**) waiting time PDF for the gene assuming a seven state promoter and a two state enhancer with independent dynamics and a gene that is ON when both the promoter and enhancer are ON. Parameters are $L = 1\mu m^{-3}$, $(\mu_i)_{1 \leq i \leq 6} = (0.2, 0.8, 0.15, 0.4, 1.2, 0.7)s^{-1}$, $(\nu_i)_{1 \leq i \leq 6} = (1.3, 0.2, 0.4, 0.6, 0.7, 0.8)s^{-1}$, $e_{ON} = 0.2s^{-1}$, $e_{OFF} = 0.6s^{-1}$.

