## [Decision Letter]

**Acceptance summary:**

This paper tackles an interesting and neglected question in developmental gene regulation – the speed at which regulatory decisions in development must be made – using mathematical models of promoters active in the early *Drosophila* embryo. The math in the paper will likely be a challenge for many readers to follow, but the questions, principles and inferences made should stimulate the many experimentalists working in this field to incorporate this thinking into their future genetic dissections and analyses of promoters and enhancers in developmental systems.

**Decision letter after peer review:**

Thank you for submitting your article "*hunchback* promoters can readout morphogenetic positional information in less than a minute" for consideration by *eLife*. Your article has been reviewed by Michael Eisen as the Senior Editor and Reviewing Editor, and two reviewers. The reviewers have opted to remain anonymous.

We are incredibly sorry for the delay in the reviews. It was unusually difficult to find reviewers and there were then additional delays. The consensus of the reviewers is that this is a thought-provoking manuscript; it uses theory to explore an important question – how developmental decisions can be made within the time-constraints imposed by development. However, the reviewers feel that the manuscript does not yet succeed in providing the kind of connection between theory and underlying biology required to reach the broad readership of *eLife*. They also found it difficult to evaluate the details of the manuscript without having this broader context. We hope that this major concern can be addressed in a revised manuscript. The comments of the reviewers are appended below.

Reviewer #1:

The manuscript "hunchback promoters can readout morphogenetic positional information in less than a minute" was interesting to read and definitely thought provoking. The authors develop a theoretical framework for the analysis of the time-dependent readout of the positional information contained in morphogen profiles, which is based on a statistical method for making optimal decisions while steadily accumulating information (the "Sequential Probability Ratio Test" developed by Wald).

The presentation is generally quite clear (assuming a good background in information theory and/or statistical physics). However, even after reading the entire manuscript, I was left with one crucial question/concern that I would need to be clarified before I can give a recommendation on this manuscript:

According to my current understanding, the presented statistical formalism is basically a theory for optimal decision making, but the notion of a "decision" in the context of interpreting a morphogen profile is left somewhat vague. The formalism takes as the information input for the decision the full statistics of the time series of transcription factor binding and unbinding events, and this information is processed by the transcription machinery into a time series of the binary variable gene ON/OFF. Up to that point, the logic seems pretty clear to me. However, then the theoretical framework seems to assume that the system can "compute" its output, i.e. whether hunchback settles into the high or the low state in a given nucleus, based on the time-dependent log-likelihood ratio, as presented in the manuscript. How should the system ("the promoter") do this? A priori, the output of the gene is just proportional to the fraction of time that the transcriptional machinery is in the ON state, since that determines the total mRNA and protein levels. I realize that in the real system there will be nonlinear effects, e.g. a direct or indirect feedback of hunchback on its own expression, which make the system behavior more complex, but I currently do not see how the system should do the seemingly complex computation required to obtain the log-likelihood ratio, which involves the statistics of the time series, not just the integral over it. Also, these nonlinear effects are not included in the model, so the authors do not seem to need them for the logic of their argument. Can the authors fully spell out their rationale regarding this point?

In my mind it is crucial for the significance of this work that the above point is made crystal clear in the manuscript (it may well be that it has indeed been explained, but that I somehow missed it). Without clarifying this point, the message of the manuscript would be reduced to a more abstract and theoretical statement like "hunchback promoters could potentially readout morphogenetic positional information in less than a minute, if they had sufficient computational ability". This statement would still be interesting, but I think then the paper would be better suited for a more specialized journal.

Reviewer #2:

In this paper Desponds and colleagues study the tradeoff between speed and accuracy of the anterior-posterior nuclei fate determination in the early fly embryo. They use hunchback regulation by the transcription factor bicoid as a case-study and using theoretical methods they study the capability of the system to decide between anterior and posterior fate sufficiently fast and with low error. They focus on bicoid binding up to 6 binding sites in the hb promoter and consider that the gene is active (ON) when a minimum number, k, binding sites are occupied. The time-evolution of ON and OFF times is then used to determine the bicoid concentration and subsequently the location of an individual nucleus within the embryo. The authors conclude that a fixed-time decision strategy would require longer times to reach this decision than that available during nuclear cycles prior to NC 14, but a variable-time strategy can do the task. Additionally, they describe how under their framework various activation rules, rates and number of binding sites influence this capability.

This work adds to a small but growing body of literature that addresses an important aspect of transcriptional regulation: time constraints during development. However, the reviewers were concerned that this work may not be accessible to a broad, biologically centered readership, namely because the text lacks a clear thread connecting the theoretical rationale to the underlying biology. The work focuses on a putative decision between high and low hunchback expression at the hunchback promoter, but does not include nor hypothesize a conceivable mechanism by which this strategy may be implemented, beyond continuing to follow the time-evolution of the system. There are also numerous contradictory statements regarding the nature of this decision throughout the text, making it difficult to discern the authors' intent. In addition, this approach applies to promoters that contain an array of binding sites for a single TF and no attempt is made to connect it to regulatory mechanisms that are more typical inside the embryo (e.g. enhancer regulation). Finally, although the Results section does a good job of contextualizing this work within recent results of other studies, there are important statements lacking references. We feel that these general concerns should be addressed before further consideration.

---

## [Author Response]

Reviewer #1:The manuscript "hunchback promoters can readout morphogenetic positional information in less than a minute" was interesting to read and definitely thought provoking. The authors develop a theoretical framework for the analysis of the time-dependent readout of the positional information contained in morphogen profiles, which is based on a statistical method for making optimal decisions while steadily accumulating information (the "Sequential Probability Ratio Test" developed by Wald).The presentation is generally quite clear (assuming a good background in information theory and/or statistical physics). However, even after reading the entire manuscript, I was left with one crucial question/concern that I would need to be clarified before I can give a recommendation on this manuscript:According to my current understanding, the presented statistical formalism is basically a theory for optimal decision making, but the notion of a "decision" in the context of interpreting a morphogen profile is left somewhat vague. The formalism takes as the information input for the decision the full statistics of the time series of transcription factor binding and unbinding events, and this information is processed by the transcription machinery into a time series of the binary variable gene ON/OFF. Up to that point, the logic seems pretty clear to me. However, then the theoretical framework seems to assume that the system can "compute" its output, i.e. whether hunchback settles into the high or the low state in a given nucleus, based on the time-dependent log-likelihood ratio, as presented in the manuscript. How should the system ("the promoter") do this? A priori, the output of the gene is just proportional to the fraction of time that the transcriptional machinery is in the ON state, since that determines the total mRNA and protein levels. I realize that in the real system there will be nonlinear effects, e.g. a direct or indirect feedback of hunchback on its own expression, which make the system behavior more complex, but I currently do not see how the system should do the seemingly complex computation required to obtain the log-likelihood ratio, which involves the statistics of the time series, not just the integral over it. Also, these nonlinear effects are not included in the model, so the authors do not seem to need them for the logic of their argument. Can the authors fully spell out their rationale regarding this point?In my mind it is crucial for the significance of this work that the above point is made crystal clear in the manuscript (it may well be that it has indeed been explained, but that I somehow missed it). Without clarifying this point, the message of the manuscript would be reduced to a more abstract and theoretical statement like "hunchback promoters could potentially readout morphogenetic positional information in less than a minute, if they had sufficient computational ability". This statement would still be interesting, but I think then the paper would be better suited for a more specialized journal.

We have added a new subsection to the Results entitled “Approximating the log-likelihood function with RNA concentrations” where we show how a simple mechanism of RNA production and degradation with delay can approximate the mRNA readouts expected if the promoter was (approximately) evaluating the log-likelihood function. This is due to the fact that the log-likelihood function has a very simple behavior: a strong decrease when the promoter is in the OFF state and a delayed but steady increase when the promoter is in the ON state. This generic behavior makes it well-suited for an approximate molecular implementation. In this model, the number of RNA molecules (or their concentration) is a good approximation for the log-likelihood, meaning that the cell could make a decision based on the RNA level alone. Alternatively, the decision could even be made upstream of the mRNA concentration and involve specific DNA architecture. To be clear, we do not claim that a full irreversible decision is made at the level of hunchback RNA as the decision that we describe is a complex process that has many ramifications in the downstream gap genes. Since exploring all these schemes requires more experimental input and would be even more exploratory, we have refrained from fully pursuing this issue. At this stage, our goal was to show that implementing the log-likelihood readout in a biochemical circuit involving this promoter is possible and relatively simple.

Reviewer #2:In this paper Desponds and colleagues study the tradeoff between speed and accuracy of the anterior-posterior nuclei fate determination in the early fly embryo. They use hunchback regulation by the transcription factor bicoid as a case-study and using theoretical methods they study the capability of the system to decide between anterior and posterior fate sufficiently fast and with low error. They focus on bicoid binding up to 6 binding sites in the hb promoter and consider that the gene is active (ON) when a minimum number, k, binding sites are occupied. The time-evolution of ON and OFF times is then used to determine the bicoid concentration and subsequently the location of an individual nucleus within the embryo. The authors conclude that a fixed-time decision strategy would require longer times to reach this decision than that available during nuclear cycles prior to NC 14, but a variable-time strategy can do the task. Additionally, they describe how under their framework various activation rules, rates and number of binding sites influence this capability.This work adds to a small but growing body of literature that addresses an important aspect of transcriptional regulation: time constraints during development. However, the reviewers were concerned that this work may not be accessible to a broad, biologically centered readership, namely because the text lacks a clear thread connecting the theoretical rationale to the underlying biology. The work focuses on a putative decision between high and low hunchback expression at the hunchback promoter, but does not include nor hypothesize a conceivable mechanism by which this strategy may be implemented, beyond continuing to follow the time-evolution of the system.

We have added a new subsection “Approximating the log-likelihood function with RNA concentrations” where we show how a simple mechanism of RNA production and degradation with delay can approximate the mRNA readouts expected if the promoter was evaluating the loglikelihood function. Our goal here is to show that the log-likelihood calculation is possible and relatively simple to implement by using a biochemical circuit that includes the hb promoter. The main elements of the biochemical approximation of the log-likelihood are a strong decrease when the promoter is in the OFF state and a delayed but steady increase when the promoter is in the ON state. We show this is possible in terms of the number of RNA molecules (or their concentration) given a delayed readout and degradation. We do not mean to claim that is the way this decision is implemented — it is one possible solution showing this kind of calculation is possible and biologically plausible.

There are also numerous contradictory statements regarding the nature of this decision throughout the text, making it difficult to discern the authors' intent.

We have added a sentence to the section the decision process of the anterior vs posterior hunchback expression to clarify our understanding of the decision process. We have also added an extra figure in the Appendix (and a reference to it in the main text) to illustrate how the times for decision-making vary across the embryo. We hope that this added material, along with the suggestion for the concrete implementation of the decision process added in subsection “Estimating the log-likelihood function with RNA concentrations”, will help make our presentation more transparent and clear.

In addition, this approach applies to promoters that contain an array of binding sites for a single TF and no attempt is made to connect it to regulatory mechanisms that are more typical inside the embryo (e.g. enhancer regulation).

We had initially decided to leave enhancers aside because adding the computational power of enhancers to that of the promoter can only increase the performance of the decision and we could already find suitable architectures within the promoter model class. To confirm this intuition and explore enhancer dynamics, we have now followed the reviewer’s suggestion and considered one example of an enhancer model that seems natural. In this model, the enhancer has two states (ON and OFF) and the gene can only be transcribed when both the enhancer and the promoter are ON. We have added a short section to the main text and a section to the Appendix with details of our parameter search and the method that we developed to compute decision times for models of joint enhancer-promoter dynamics. We did find that enhancers improve the performance of the decision, all the more when they have a larger number of binding sites.

Finally, although the Results section does a good job of contextualizing this work within recent results of other studies, there are important statements lacking references. We feel that these general concerns should be addressed before further consideration.

We have added several references throughout the text so that all claims should now supported by proper referencing to the literature.